# Integrating GPR and ice-thickness models for improved bedrock detection: the case study of Rutor temperate glacier

Andrea Vergnano[1,2], Diego Franco[1], and Alberto Godio[1]

[1]Department of Environment, Land and Infrastructure Engineering (DIATI), Politecnico di Torino, Torino, Italy
[2]Department of Earth Sciences, Università degli studi di Torino, Torino, Italy

**Correspondence:** Andrea Vergnano (andrea.vergnano@polito.it, andrea.vergnano@unito.it)

**Abstract.**

Estimating the ice volume contained in glaciers is a topic of increasing interest, because the cryosphere has rapidly evolved during the last decades of global warming. Many disciplines collaborate to study the global warming impacts on glaciers. From the perspective of a geophysical research team, we examine the advantages of integrating glaciological ice-thickness models into the workflow for geophysical data processing.

In temperate glaciers, the widespread englacial water content often challenges the analysis of Ground Penetrating Radar (GPR) data, the most commonly used geophysical technique for measuring ice thickness. In fact, recognizing the ice-bedrock interface is hard where the englacial water content generate high levels of scattering in the GPR data. A past GPR survey estimated that the Rutor glacier (European Alps) stored about 150 million $m^3$ of ice in 2008. However, this estimate proved unrealistic after analyzing the geodetic mass balance of the following decade. Therefore, we analyzed new GPR measurements on the same glacier, acknowledging how it is easy to misinterpret the data. This study builds on the idea that ice-thickness models can help the GPR data analyst to better recognize the ice-bedrock interface in the radargrams. We selected four models, OGGM, GlabTop2-Py, Original-GlabTop2 and GlaTE, that estimate bedrock topography starting from surface topography, following principles of ice flow theory, ice dynamics and mass conservation. Combined visualization of the GPR and model data in 2D and 3D helped the analyst to manually select the ice-bedrock interface. This proved useful especially where the GPR data was scattered and interpretation was uncertain. The GPR data were then used to constrain one of the models, GlaTE, to produce an ice-thickness map that is the result of both model estimates and GPR information. According to this methodology, the Rutor glacier stored about 450 million $m^3$ of ice in 2021, about three times the previous estimate. The analysis of more than one approach, and a simple sensitivity analysis on the GlaTE model, made it possible to estimate the ice-thickness uncertainty linked to the use of models and GPR. The workflow is openly available in the supplementary material and may improve future GPR surveys of temperate glaciers, especially when facing scattered data due to englacial water content or other sparse reflectors such as debris. More accurate ice-thickness estimates will improve local studies and provide better calibration data for regional studies.

# 1  Introduction

Temperate valley glaciers are a characteristic feature of mountain chains in temperate zones, such as the European Alps, and have a pivotal role in the hydrological cycle (Milner et al., 2017). Studying the glacier processes and changes linked to the current climate crisis often requires a reliable estimate of their present-day ice-thickness distribution and geometry, including bedrock topography, crevasses, cavities, debris, and englacial water (Haeberli et al., 2019). Temperate glaciers are characterized by containing temperate (or "warm") ice at the melting point. Indeed, colder ice zones often exist in temperate glaciers, but they are limited to specific areas or seasons, e.g., where the snow melts late in the warm season, or in areas where the ice is thin and provides little insulation to the geothermal heat during the cold season (Suter et al., 2001; Reinardy et al., 2019). Temperate glaciers at the pressure melting point are primed for rapid meltwater production upon small energy or heat inputs.

A high englacial water content in glaciers can challenge the interpretation of bedrock returns from Ground Penetrating Radar (GPR), one of the most used techniques for surveying ice masses (Colucci et al., 2015; Forte et al., 2015; Urbini et al., 2019). Since englacial water has very different dielectric permittivity compared to ice, it reflects and scatters the electromagnetic wave, hindering the signal from traveling to the bedrock and then being detected on its way back (Reinardy et al., 2019). Smaller-scale heterogeneities, such as small fractures or sediment grains smaller than about a quarter wavelength, generate weak or undetectable responses, but their presence influences the signals as they pass by. The heterogeneities extract energy as the electromagnetic field passes and scatter it in all directions (Jol, 2009). Langhammer et al. (2019b) studied the effects of antenna orientation on detection of the bedrock reflection. Challenges in detecting basal returns over temperate glaciers have been studied by Rutishauser et al. (2016), who analyzed a large set of GPR data acquired on Swiss glaciers and found that depending on the specific glacier, the bedrock interface could only be successfully detected in 12-69% of the data due to this scattering issue. GPR signal scattering rarely occurs in arctic cold glaciers, but when it is detected in some areas, it may be evidence of temperate conditions and englacial water (Karušs et al., 2022). Another common source of noise in GPR sections is the presence of debris at the glacier surface, as noted by Colombero et al. (2019), or at depth, due to phenomena such as adfreezing and entrainment of sediments into the basal ice layer (Weertman, 1961). Air bubbles trapped in ice cause scattering of GPR signal, which helps differentiate between various types of ice, such as firn and superimposed ice (Langley et al., 2009).

To study and address the issue of detecting bed returns obscured by scattering noise in temperate glaciers, we focus on the Rutor Glacier (RGI ID: RGI60-11.03039), the third-largest glacier in the Aosta Valley, Southern Europe Alps. The Rutor massif develops on multiple terraces, over which the Rutor Glacier has advanced or retreated during the last centuries because of the climate variations, shaping the basin morphology, as reviewed by Vergnano et al. (2023). Nowadays, the Rutor Glacier covers an area of 7.5 $km^2$ and it is situated in the municipality of La Thuile. It has retreated, since about 1860 (when it was at the Little Ice Age maximum extent), to an upper terrace, forming new proglacial lakes near each of its three tongues (western, central, and eastern). Its tongues are situated on the northern side of the glacier at about 2550 m a.s.l. (the eastern tongue) and 2650 m (the central and the western tongues). Its highest elevation is about 3440 m, near the southern margin.

In recent decades, the glacier has experienced an accelerated retreat, linked to climate warming (Corte et al., 2024; Gizzi et al., 2022). By differencing 2021 and 2008 DEMs, pixel by pixel, we observed the changes in the glacier surface topography

from 2008 to 2021. This simple analysis showed a loss of more than 20 vertical meters of ice in about 1/4 of the current glacier area (especially in the tongue area) for a total ice volume loss of about 100 million $m^3$. The 2008 DEM was retrieved from the regional cartography (Val d'Aosta Region, 2008); the 2021 DEM is available in an open repository described by Corte et al. (2024). Previous GPR surveys (1996-2006) reported an average thickness of the glacier of 17.5 m (a volume of 150 million $m^3$ over an area of 8.5 $km^2$) (Villa et al., 2008, 2007). However, this estimate does not agree with the ice lost from 2008 to 2021. If the previous ice volume estimate was correct, Rutor Glacier would have lost 2/3 of its volume in just one decade, meaning that only 50 million $m^3$ are left. However, while shrinking is evident, the remaining ice volume is likely larger given the relatively small area reduction in that decade (see Figure 2). We hypothesize that the ice volume of 150 million $m^3$ is underestimated, and that this issue may be due to problems in interpreting scattered GPR data. The analysis performed on the same glacier by Viani et al. (2020) presents similar considerations about the challenge of interpreting the correct bedrock geometry. Moreover, many examples of other alpine glaciers with similar areas consistently show greater ice thickness (Grab et al., 2021).

In this study, the Rutor Glacier is investigated with two new GPR datasets, acquired in May 2012 with a helicopter-based survey (Morra Di Cella, 2024) and in May 2022 with a ground-based survey. These new datasets reveal high scattering of the radar signal over most parts of the glacier, demonstrating the difficulty in detecting the ice-bedrock interface. The scattering zone is often located at a depth of around 20 meters, and may easily be misinterpreted as the ice-bedrock interface, possibly explaining the previous doubtful estimates of ice thickness (Villa et al., 2008).

To address this problem encountered on the Rutor Glacier, but common to other temperate glaciers, this study explores the idea that ice-thickness models may aid in the interpretation of scattered GPR sections, especially to fill the gaps in the most scattered survey sections. Those algorithms require, as input, the glacier surface topography, which can be retrieved, for example, from satellite imagery. Some models have obtained much acknowledgment in recent years and were reviewed during the ITMIX project (Farinotti et al., 2017), in which many glacier models were compared on the same set of glaciers. One of the conclusions of that research is that considering an average output from different models provides a more reliable ice-thickness estimate than finely tuning just one model.

The ice-thickness modeling algorithms employed in this work are four different models: OGGM (Maussion et al., 2019), GlaTE (Langhammer et al., 2019a), the original GlabTop2 (Frey et al., 2014), and its open-source implementation Glabtop2-Py. The ice thickness is predicted by using the four models. During the manual selection of the ice-bedrock interface in the GPR data, the results from the models are superimposed on the radargram to help identify the most likely ice-bedrock interface, often submerged by noise due to the englacial water content. Finally, a second run of the GlaTE model constrains the estimate of the ice thickness by using the ice-bedrock interface selected manually from the GPR data, to provide a final model of the bedrock topography.

## 2 Methods

We present here an overview of the workflow, enumerating the main steps of the proposed methodology. Then, the next paragraphs analyze the methodology in more detail. For reasons of space, we did not detail here all the technical steps needed

to reproduce the workflow. The interested reader is encouraged to download the supplementary material, organized with the same logic used here, which contain "readme.txt" files providing technical help.

1. Run four models (Original-GlabTop2, Glabtop2-Py, GlaTE, OGGM), which, based on the surface topography, estimate the ice thickness. We calculated the ice-thickness average and standard deviation of the four models, to have an overview of the ice-thickness uncertainty.

2. Collect and analyze new GPR datasets from two acquisitions, a helicopter-based survey made in 2012 and a ground-based survey made in 2022. The GPR data is often scattered and difficult to interpret.

3. Extract the ice thickness estimated by the models in the same locations of the GPR paths, thanks to the v.sample tool of QGIS, GRASS plugin (QGIS Development Team, 2021), using a bilinear interpolation method.

4. Visualize the GPR data and the model estimations together, in 2D with RGPR, (Huber and Hans, 2018) and 3D with Paraview (Ayachit, 2015).

5. Manually select ("pick") the bedrock reflector in the GPR data with the help of the estimations from the three models, to reduce the chance of misinterpreting bedrock reflections. For analysis and discussion purposes, we divided the pickings into "sure" pickings, "guided by model" pickings, and "probably incorrect" pickings.

6. Run the GlaTE model for a second time, constraining the estimations with the GPR data.

7. Run the GlaTE model multiple times, varying the input glaciological parameters, to perform a simple sensitivity analysis on both the unconstrained and the constrained-by-GPR version of the model.

8. Compare the results with literature products.

## 2.1 Ice thickness models

The GlabTop2, GlaTE and OGGM models used in this study require, as input, a digital elevation model (DEM) of the glacier surface, and they estimate the ice-thickness based on ice flow theory, ice dynamics and mass conservation. In this study, these models were employed for two objectives. First, to provide an initial estimate of the ice thickness distribution of the Rutor Glacier, in order to help identify the ice-bedrock interface in the GPR sections. Second, the GlaTE model was employed to provide a final estimate of the Rutor Glacier ice thickness, based on both model estimates and GPR data constraints.

The models required additional input parameters (e.g. ice density, Glen's exponent, etc.). These were checked for consistency with the Rutor Glacier study area, but unless otherwise stated, the default values from similar alpine glacier studies by the model developers were used. This is a simple choice that could introduce some errors: in fact, these parameters are not transferable between glaciers (Zekollari et al., 2022). For this reason, we performed a sensitivity analysis on the final model (see the specific paragraph).

The DEM and the glacier margin, manually drawn based on a high-resolution orthophoto, come from a topographical survey carried out in 2021 (Macelloni et al., 2022; Corte et al., 2024). With the warp/reproject tool of QGIS software, using a bilinear-based triangulation method, the DEM was undersampled to 20-m resolution, for computation-time optimization and because using a too-fine resolution was found to produce misleading structures in the resulting bedrock topography.

  The features of the three models are described below.

### 2.1.1 GlabTop2

The GlabTop2 (Glacier bed Topography 2) model assesses the distribution of ice thickness in a glacier starting from a DEM file and a mask file containing the margin of the glacier (Frey et al., 2014). It employs an algorithm first developed by Linsbauer et al. (2012), but slightly modified to avoid the laborious process of manually drawing branch lines:

$$h_f = \frac{\tau}{f \cdot \rho \cdot g \cdot sin(\alpha)} \tag{1}$$

where $h_f$ is the mean ice thickness along the central glacier flow line, $f$ is a shape factor, fixed at 0.8 (according to Haeberli and Hoelzle (1995); Frey et al. (2014)), $\tau$ the shear stress at the glacier base, calculated with an empirical formula based on the elevation range of the glacier $\Delta H$ (equation 2), and expressed in kPa, $\rho$ the ice density (900 kg/m$^3$) (Langhammer et al., 2019a), $g$ is the gravity acceleration, and $\alpha$ is the surface topography slope:

$$\tau = 0.5 + 159.81 \Delta H - 43.5 \Delta H^2 \tag{2}$$

The first processing step employed by the algorithm is an initial approximation of ice thickness in some random cells of the domain, based on the surface slope of a sufficiently large buffer zone around the cell. Then, the ice thickness of the remaining cells is estimated with a simple inverse distance weighting algorithm. The two steps are repeated for $n$ times, and the results are averaged.

  As the original GlabTop2 code is written in a closed-source environment, we also used an implementation of GlabTop2
in Python, available open-source in a public repository (Terink, 2018). Further details are provided in the appendix of Frey et al. (2014). Glabtop2-Py is a partial implementation of the original code, and we noted a poorer performance near the glacier margins. In the supplementary material, the GlabTop2-Py model code adapted for this case study is available to ensure full reproducibility. The config.cfg file contains the input parameters used in this study.

### 2.1.2 GlaTE

GlaTE (Glacier Thickness Estimation) is also based on equation 1, but with a different estimate of the shear stress $\tau$ and a different implementation algorithm, following to the work of Clarke et al. (2013). In this work, it was employed in two separate steps: first, to provide an initial estimate of the glacier thickness, together with the other three models, and second, to calculate a final estimate of the ice thickness with known ice thickness points (from GPR data) constraining the model. In fact, the strength of GlaTE is the integration between model estimates and ground-proof data derived from GPR profiles.
GlaTE performs an inversion procedure, constraining the ice thickness results such that they match, with a certain degree of

uncertainty, a series of ground-proof data, while following some smoothness requirements, respect the glacier perimeter, and the values at the glacier border are consistent with the terrain slope outside the glacier. The system of equations to be inverted can be summarized into the matrix in equation 3 :

$$
\begin{bmatrix}
\lambda_1 G \\
\lambda_2 L \\
\lambda_2 B_{gb} \\
\lambda_3 B_0 \\
\lambda_4 S
\end{bmatrix}
h_{est} =
\begin{bmatrix}
\lambda_1 h_{GPR} \\
\lambda_2 \nabla h_{Clarke} \\
\lambda_2 \nabla h_{boundary} \\
\lambda_3 \\
\lambda_4
\end{bmatrix}
\tag{3}
$$

where $h_{GPR}$ is the ground proof GPR-derived ice thickness, $h_{Clarke}$ is the ice thickness modeled according to Clarke's algorithm, $\nabla h_{bound}$ is the gradient of terrain slope at the boundary of the glacier. The operator G ensures the constraint with GPR data, $L$ with the ice thickness modeled according to Clarke's algorithm, $B_{gb}$ with the slope outside the glacier, $B_0$ with the 0 thickness at the boundary, while $S$ is a smoothing constraint. The $\lambda$ factors are weighting factors and are varied in an iterative manner, in order to give maximum weight to the Clarke model and the smoothness constraint while fitting the GPR data (Grab

et al., 2021). The ice density was set at $900 \ \mathrm{kg/m^3}$. The creep factor (or ice softness) $A$ was set to be $2.4 \cdot 10^{-24} \ \mathrm{s^{-1} Pa^{-3}}$, neglecting its temperature dependence as if the glacier was at 0°C (Cuffey and Paterson, 2010), since the Rutor Glacier shows temperate conditions (even if this could potentially not be true for the entire glacier) (Cook et al., 2020). The exponent of Glen's flow law was fixed at 3, as considered the best approximation in the absence of data about the ice fabric (Glen and Paren, 1975). The potential for debris presence, which can be added to the model, was not taken into consideration, because

both the orthophotos and visual investigation showed no evident thick debris cover in the ablation area of Rutor Glacier. For further details, see Langhammer et al. (2019a); Grab et al. (2021); Schwanghart and Scherler (2014).

    The model is open source, it runs in MATLAB environment and is available in an open repository (Maurer, 2022). In the supplementary materials, the GlaTE model code adapted for this case study is available to ensure full reproducibility. The parameters.txt file in the Matlab folder contains the input parameters used in this study.

### 2.1.3   OGGM

The OGGM (Open Global Glacier Model) is an open-source collection of algorithms written in Python that provides different insights into glaciers, for example, thickness, meltwater runoff, and future predictions based on climate variations. Its main aim is regional-scale modeling, but the code is modular and can be adapted to work with a single glacier. In this work, the OGGM version 1.6.2 was used, and only the OGGM ice thickness inversion algorithm was employed, which is based on ice

flow dynamics and mass conservation (Farinotti et al., 2009; Maussion et al., 2019). The ice flux is computed as equation 4:

$$
q = uS = \left( f_d h \tau^n + f_s \frac{\tau^n}{h} \right) S
\tag{4}
$$

where $h$ is the ice thickness, $q$ is the ice volume flux, $u$ is the ice flow velocity, $S$ is the section, which in case of a simplified parabolic section is $= 2/3 \cdot h \cdot width$, $n$ is the exponent according to Glen's law (=3), $\tau$ is the shear stress, $f_d$ is proportional

to the ice softness $A$ ($f_d = 2A/n + 2$), $f_s$ is a sliding factor, neglected for simplicity in this run of the model. The flux $q$ in a section is also equal to the mass balance (mass input - output due to precipitation and melting) integrated over the area of the glacier situated above the section considered. This model, in its latest version, works even without the equilibrium assumption, calibrating the mass balance parameters using a geodetic mass balance calibration (Hugonnet et al., 2021). The climate data are taken from the W5E5 dataset (Lange, 2019). During the inversion process, one parameter, the ice softness $A$, is calibrated against a regional consensus ice volume (Farinotti et al., 2017) and is allowed to vary, in general about one order of magnitude, from the standard value of $2.4 \cdot 10^{-24} \ \mathrm{s^{-1} Pa^{-3}}$.

Further details and the implementation in the OGGM framework are described in Maussion et al. (2019) and the software is freely available in an open repository (Maussion, 2024). In the supplementary materials, the OGGM model code adapted for this case study is available to ensure full reproducibility. Among the various folders and files, the .ipnyb Python Jupiter notebook contains a small section about the input and calibrated parameters used in this study.

## 2.2 Ground-penetrating radar (GPR)

Two different low-frequency antennas were tested to survey the thick ice of the Rutor Glacier. In the 2012 helicopter-based dataset, a GSSI single-frequency antenna with a central frequency of 70 MHz was employed. In the 2022 ground-based survey, a 40 MHz antenna, RIS ONE model with single-frequency antenna configuration, manufactured by IDS, was carried on skis by an operator. The location of the GPR profiles is shown in Figure 1.

The raw data were processed using the commercial ReflexW software (Sandmeier, 2012), with the following method:

1. Application of a background removal filter (x-direction average-trace removal over the entire profiles), to eliminate instrument noise constant in the x-direction.

2. Application of a bandpass butterworth filter, from 0 to 150 MHz, to eliminate high frequency noise.

3. Application of a make equidistant-traces filter, to plot 1 trace per meter, especially important for the 2012 survey, since the helicopter was not flying constantly at the same speed.

4. Application of a gain filter called Energy decay: a gain curve in y-direction (time) applied on the complete profile based on a mean amplitude decay curve, which is automatically determined (Sandmeier, 2012), to compensate attenuation and geometric spreading in the time-direction.

5. Conversion of the y-axis from time to depth, assuming a constant velocity of the electromagnetic signal in the ice of 0.167 m/ns (Bohleber et al., 2017). For the helicopter-based survey, we also removed the part of the signal traveling in the air. We did not notice any interference from nearby slopes or features, as in other studies (Church et al., 2018): this is because the helicopter was flying at a low altitude above ground and most of the Rutor Glacier is not surrounded by steep slopes.

6. Manual selection of the ice-bedrock interface with the guidance of model estimations.

The data were not migrated. Some attempts with simplified velocity models were made, without any significant enhancement. Given the very scattered radargrams of this survey, we investigated most of the advanced processing algorithms included in the ReflexW software, in order to find eventual filters improving the radargram clarity. However, simple processing flows, such as the one reported here, gave the most effective results.

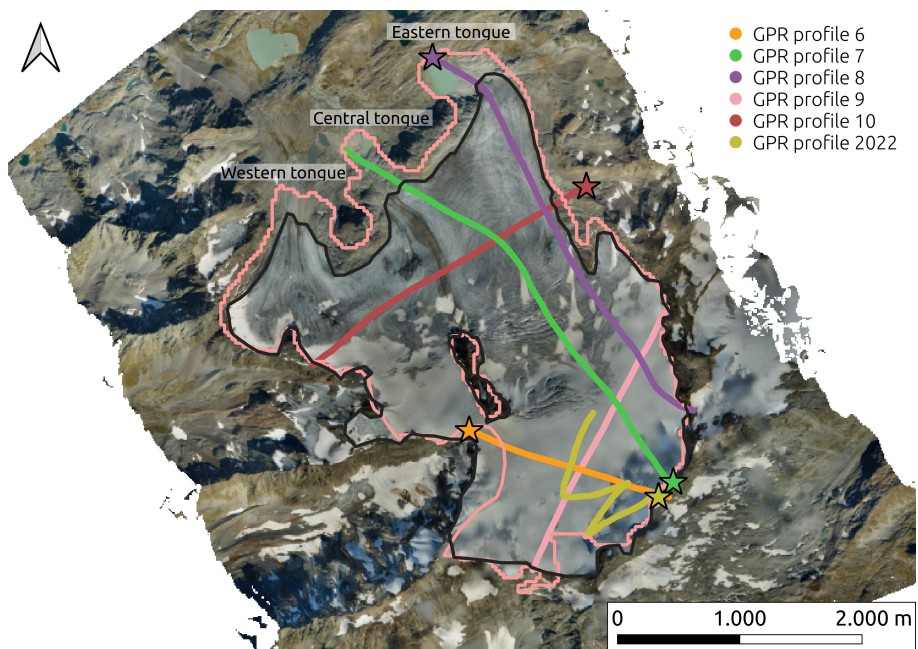

**Figure 1.** The GPR profiles of the Rutor Glacier survey. Star markers indicate the start of the profiles. Numbering from 6 to 10 is based on the numbering of the original helicopter-based dataset (Morra Di Cella, 2024). The black glacier margin is that of 2021 used in this study (Macelloni et al., 2022). The pink glacier margin is the 2003 RGI outline (RGI Consortium, 2017).

### 2.3    2D and 3D visualization of GPR and models together - selection of the bedrock interface

After having run the three unconstrained models and had an overview of the ice thickness, we needed tools to visualize them together with the processed GPR data, making it possible to perform the selection of the ice-bedrock interface with the model guidance.

The first visualization tool that we employed is RGPR, an open source software package to analyze and process GPR data (Huber and Hans, 2018). Since it runs in the R environment, it was straightforward to slightly adapt the code to plot the ice

thickness estimated by the models together with the GPR profiles. The visual overlap of GPR and models helped in picking the ice-bedrock interface.

With a second visualization tool, Paraview (Ayachit, 2015), we built a 3D reconstruction of surface topography, bedrock topography based on three models, and GPR profiles. The aim of this tool is the same as the profile visualization made in RGPR. However, notwithstanding the drawback of being a little more complex to set up, it has two advantages: first, the 3D

visualization allows one to directly see where the GPR profiles intersect, helping recognize the ice-bedrock interface as it develops along different profiles; second, visualizing the entire glacier in a 3D environment is a better support to integrate the user's local knowledge of the glacier, allowing them to better recognize the glacier shape and features and their correlation with ice thickness.

Some topographical adjustments were necessary to assist in analyzing GPR observations that span different time periods (2012 and 2022). Moreover, the DEM used as models input was acquired in another year, in 2021. In other words, the models represent the 2021 situation, and the GPR data corresponds to 2012 and 2022.

To compare the four ice-thickness models with the GPR sections of the 2012 survey, every model had to be "converted" to 2012, that is, the ice lost from 2012 to 2021 had to be accounted for in the comparison. In fact, the ice lost in that time frame, especially in the lowest parts of the glacier, was not negligible, as shown in Figure 2. Since a good 2008 DEM of the glacier surface was available from the regional cartography (Val d'Aosta Region, 2008), the ice loss from 2012 to 2021 was estimated by a simple linear interpolation between the 2008 glacier surface elevation and the 2021 glacier surface elevation, supposing that the glacier melting in those years was constant on average. The calculated ice lost from 2012 to 2021 was added to the model thickness, to allow for comparison with the 2012 GPR sections. For comparing the 2022 GPR profile to the models, no correction was performed, since the ice lost from 2021 to 2022 is negligible for the type of comparison performed, and the 2022 profile is situated in the highest portion of the glacier, where melting is minimal.

To perform these topographical adjustments and analyses, we made sure that all data were reprojected to the same Coordinate Reference System (WGS84, UTM 32 N), using the warp/reproject tool of QGIS software.

Thanks to these visualization tools, the comparison between models and GPR could be done. The ice-bedrock interface was then manually selected ("picked") in ReflexW, assigning different codes (and colors) to "sure" pickings, "guided by model" pickings, and "probably incorrect" pickings. The "probably incorrect" pickings were selected in those portions of the GPR profiles where it could seem reasonable to "see" an ice-bedrock interface, but considering the broader picture of the glacier, by observing the glacier shape estimated by the models, and based on local knowledge, it is likely a false positive. Those "probably incorrect" pickings served only to show the difficulties in interpreting scattered GPR data, and they were not used in the following analysis of the GlaTE model constrained by GPR data.

## 2.4 GlaTE model constrained by GPR and sensitivity analysis

After the manual selection of the reflectors in the GPR data, the final GlaTE model, constrained by GPR data, was run. To achieve this, the ice-bedrock interface manually selected on GPR data, based on the 2012 survey, was corrected to 2021, by subtracting the ice lost from 2012 to 2021 from the thickness. The GPR-based ice thickness was given an estimated error of 30% as GlaTE input parameter, to account for the uncertain nature of the pickings due to the scattered data.

A simple sensitivity analysis was performed on the glaciological parameters A (creep factor [$s^{-1}Pa^{-3}$]) and gsi (fraction of creep relative to basal sliding [-]). The creep factor A varied from $1.2 \cdot 10^{-24}$ to $4.8 \cdot 10^{-24}$, $2.4 \cdot 10^{-24}$ being the standard value as suggested by Cuffey and Paterson (2010), while gsi varied from 0.25 to 0.75, with 0.5 being the default value suggested by

the model developers. In QGIS, we calculated the average and standard deviation of the various ice thickness models estimated with the different parameters, for both constrained and unconstrained GlaTE models.

## 3 Results

Figure 2 presents a map of the ice-thickness changes of the Rutor glacier from 2008 to 2021. As mentioned in the Introduction section, this analysis was also one of the starting motivations of this study. Integrating the elevation variations over the glacier area, a total ice loss of 100 million $m^3$ was estimated. This estimate did not agree well with the previous total ice volume estimate of 150 million $m^3$ by Villa et al. (2008), based on a GPR survey, because it would have meant that only 50 million $m^3$ still remained in the glacier, which was improbable, given its size.

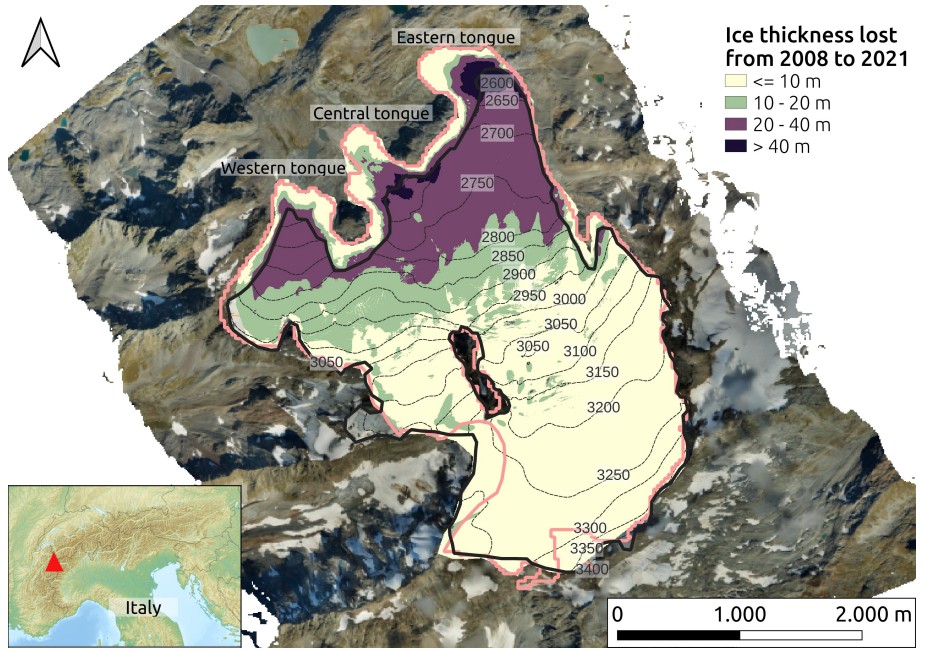

**Figure 2.** The Rutor Glacier (RGI ID: RGI60-11.03039) divided in four categories, according to changes in ice thickness (m) from 2008 to 2021 ("tokyo" color scale, according to Crameri (2021)). The black glacier margin is that of 2021 used in this study (Macelloni et al., 2022). The pink glacier margin is the 2003 RGI outline (RGI Consortium, 2017), and this choice is visible in the yellow areas near the three tongues of the glacier, which did not change in altitude from 2008 to 2021 because they were already free of ice in 2008.

Based on the above observation, we supposed that probably the previous GPR data by Villa et al. (2008) could have been misinterpreted. In fact, we noticed great difficulties in interpreting our GPR data on the temperate Rutor Glacier, because they were affected by scattering and random noise. Figure 3 shows an example section of the helicopter-based 2012 dataset. An omnipresent clutter zone is situated at 10-50 m of depth, and could be easily misinterpreted for the true bedrock, fortunately visible in the right part of the profile.

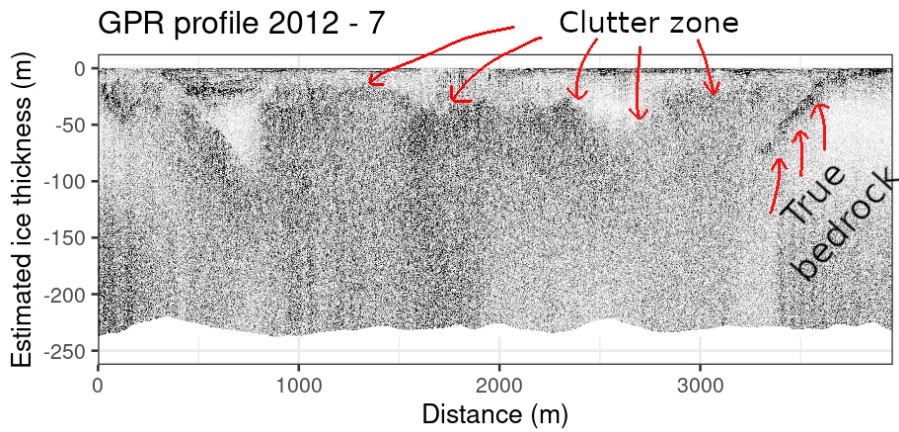

**Figure 3.** The GPR section number 7 of the helicopter-based 2012 dataset. The thickness scale was built assuming a constant velocity of the electromagnetic signal in ice of 0.167 m/ns. The white area is compact firn or ice without water. The ice-bedrock interface is well visible only on the right part of the image.

Looking at the central part of Figure 3, one could be tempted to interpret the area without reflections, pictured in white, as ice, and the first strong backscatter zone, omnipresent in the profile at 10-50 m of depth, as the bedrock. However, on the right side of the plot, the ice-bedrock interface that submerges below the backscatter zone suggests that the former backscatter area is probably not a bed return, even if it looks like it. Elsewhere, possible ice-bedrock interfaces may be spotted and somewhat followed in the GPR section, but the interpretation is far from straightforward.

Therefore, a help was needed to get a general idea of where the bedrock could be situated. This help came from the four ice-thickness models (Original-GlabTop2, Glabtop2-Py, GlaTE, and OGGM) shown in Figure 4, which estimated the bedrock position based on the surface topography, ice flow theory, ice dynamics and mass conservation.

The four raster maps in Figure 4, although produced with different models, showed a very similar reconstruction of the glacier geometry. The total ice mass estimated by the models was consistently higher than the 150 million $\mathrm{m}^3$ of the previous estimate, and it was about: OGGM = 630, GlaTE = 510, GlabTop2-Py = 580, Original-Glabtop2 = 470 million $\mathrm{m}^3$. Average and standard deviation plots are also included in Figure 4, while a main flowline profile is shown in Figure 5.

To visualize the model estimation together with the GPR data, their thickness value was bilinearly interpolated at the points corresponding to the GPR survey paths. Figure 6 shows an example ice-thickness profile extracted from the previous models, matching the location of a GPR section, used as a base to manually select more easily the reflections of the ice-bedrock interface. With the guidance of the models, the GPR analyst is more conscious that the clutter zone identified in Figure 3 is not the bedrock, but represents some other internal feature of the glacier. The manual selection of the bedrock interface was indicated in Figure 6 with different colours, to distinguish the "sure" pickings, the probably "incorrect" ones, and those guided by the models. In this way, we selected about 15% of GPR data as "sure" pickings, 26% as "guided by models", and 37% as "probably incorrect". Some of GPR data was not picked at all, where the GPR data was too scattered or the models were not

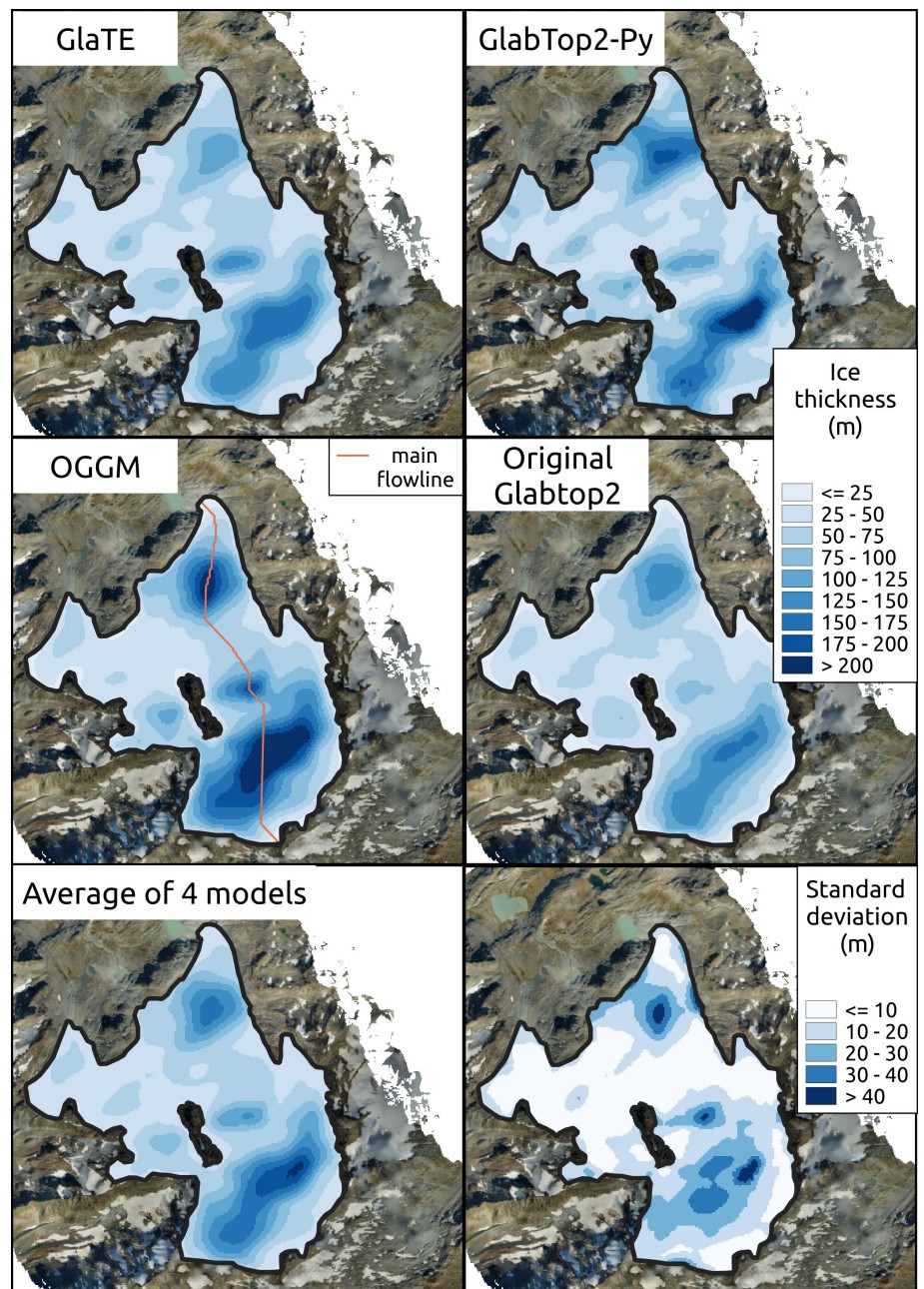

**Figure 4.** Ice thickness maps of the Rutor Glacier produced with GlaTE, GlabTop2-Py, OGGM, and Original Glabtop2 models without any constrain by ground-proof data, but only with topographic surface data as input.

consistent between each other. Moreover, a 3D environment was set up in Paraview software to better visualize the GPR profiles as they intersected between each other and the modeled bedrock. This visualization tool enhanced the visual interpretation of

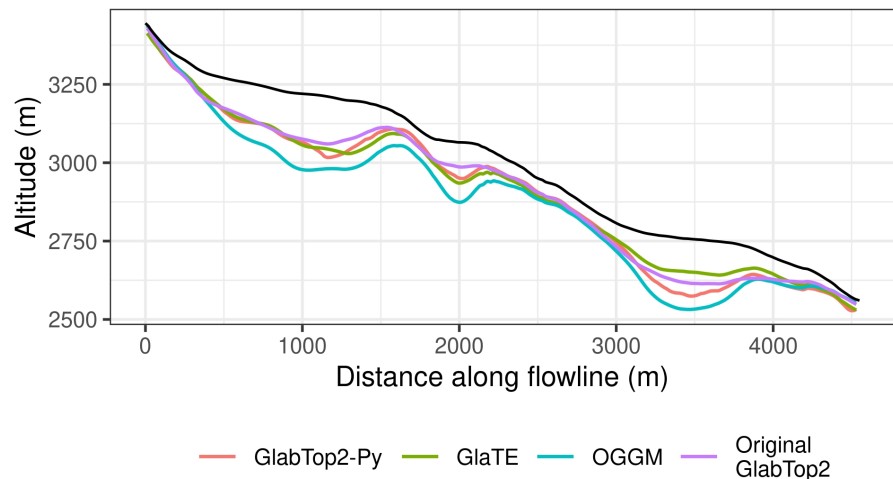

**Figure 5.** Main flowline profile. The surface topography is in black, the modelled bedrock is in various colors.

the GPR data, and an example is given in the Appendix, in Figure A7. All the other GPR sections with the model estimations are shown and commented in detail in the Appendix (Figures A1 to A6). The interested reader is strongly encouraged to take

a little time to familiarize with those GPR profiles and the model estimations, in order to follow the Discussion section more conscious of the subjective interpretation problems we faced.

The bedrock interface selected in the GPR data was then imported into the GlaTE model to perform a constrained run. Figure 7 shows the results of the last GlaTE inversion, performed constraining the model with the manually selected ice-bedrock interface of all the GPR profiles. The total ice volume calculated in this way was about 450 million $\mathrm{m}^3$.

Figure 8 shows the results of the sensitivity analysis on the final GlaTE model, in both its unconstrained and constrained version. Maps of average and standard deviation are shown, as well as a profile visualization. By varying the input glaciological parameters A and gsi, the unconstrained version of the model experienced a standard average variation of +- 75 millions $\mathrm{m}^3$ of ice, whereas the variation of the constrained model was negligible.

In Table 1 we report the ice volume estimates of the different models employed in this study, as well as ice-thickness

literature products from Farinotti et al. (2019), Millan et al. (2022) and Villa et al. (2008).

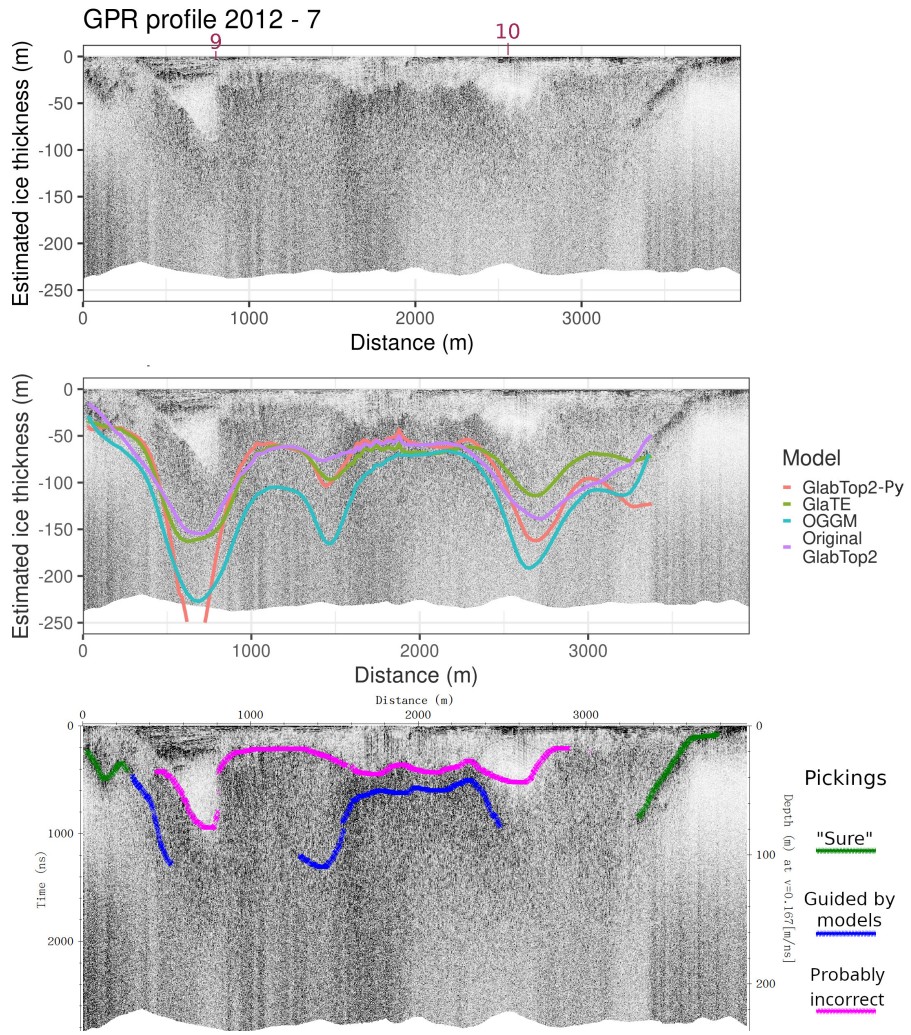

**Figure 6.** An example GPR section with the estimation from the four models overlayed: GlabTop2-Py in red, GlaTE in green, OGGM in blue, Original-GlabTop2 in purple. The manual selection of the ice-bedrock interface, which was possible by comparing the GPR profiles and the models, is displayed in different colours according to their reliability. Purple numbers indicate where this profile crosses other GPR profiles.

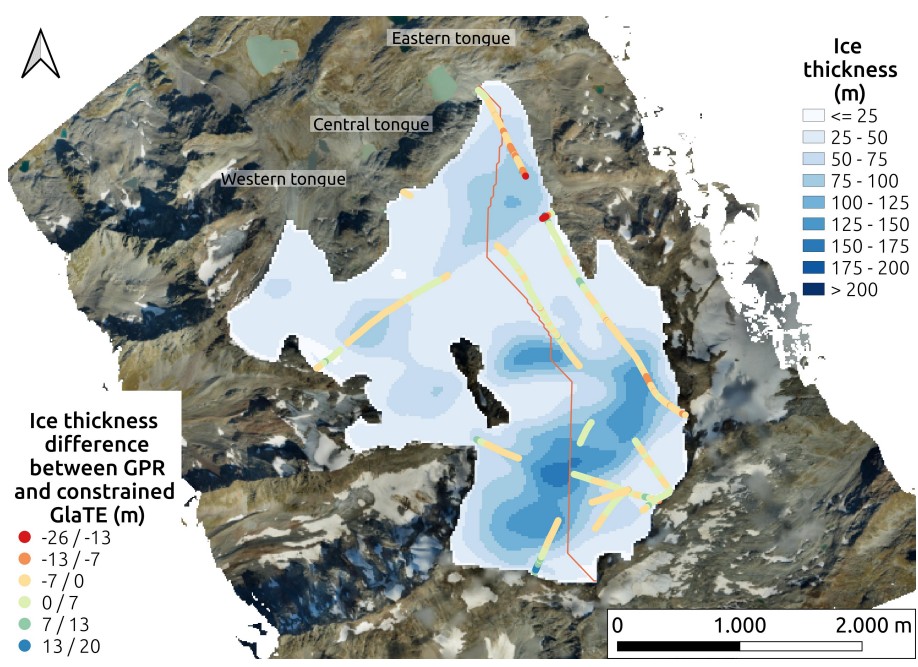

**Figure 7.** The final model of the Rutor Glacier ice thickness obtained by GlaTE model constrained with the GPR data. The pickings of GPR data are shown as differences between GPR ice thickness and GlaTE ice thickness. The final GlaTE model was fairly consistent with the GPR input data.

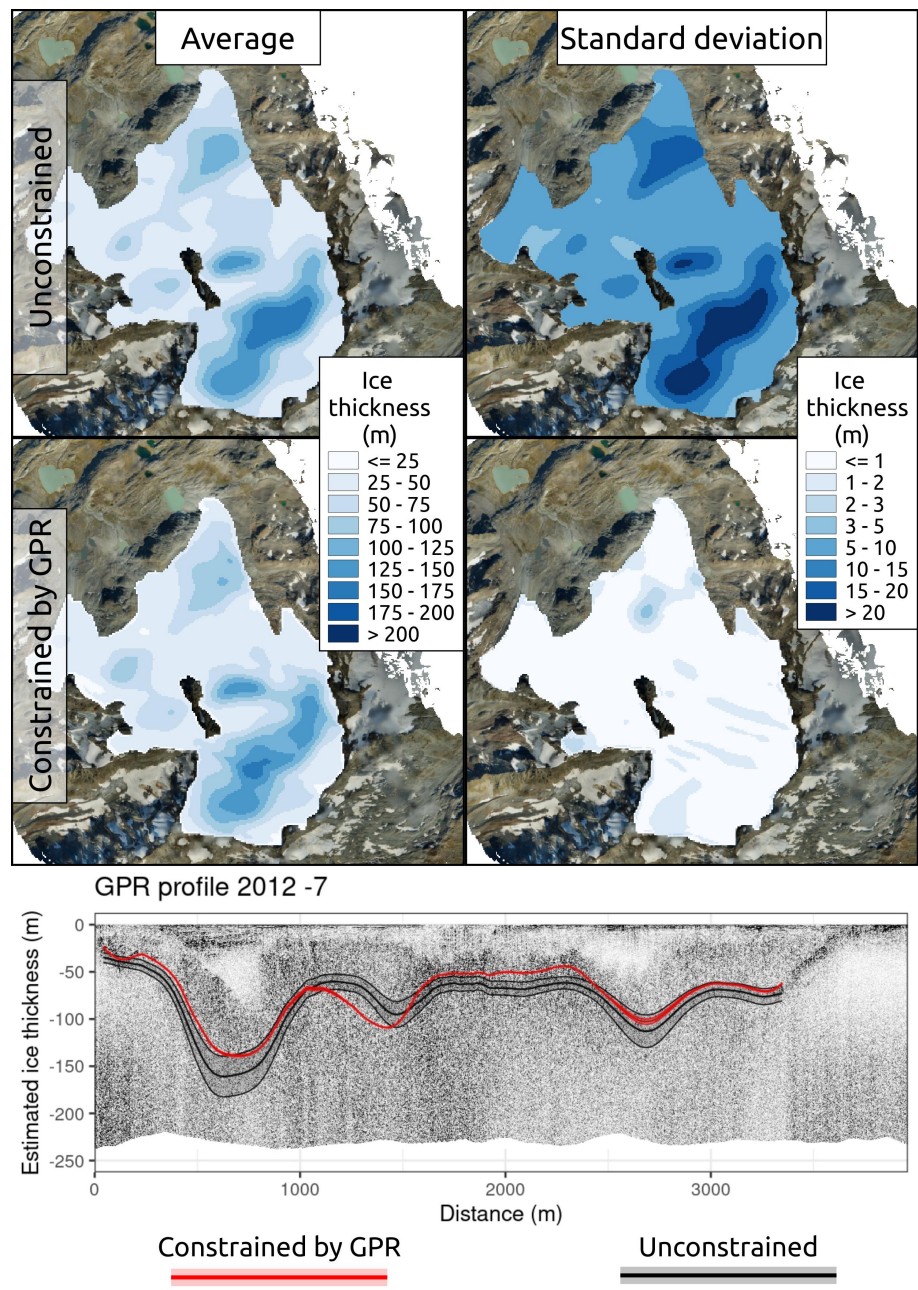

**Figure 8.** Sensitivity analysis plots on the unconstrained (top 2 plots) and constrained (middle 2 plots). Profile 7 view of the sensitivity analysis (bottom plot).

**Table 1.** Ice volume comparison between the different ice-thickness products of this study and literature.

| Model | Ice volume (million m3) |
|---|---|
| OGGM | 626 |
| GlabTop2-Py | 577 |
| GlaTE (unconstrained) | 512 |
| Original Glabtop2 | 471 |
| Average of 4 models | 543 |
| Standard deviation | 102 |
| GlaTE (constrained by GPR) | 452 |
| Consensus (Farinotti) | 631 |
| Millan | 634 |
| Villa et al. (2008) | 150 |
| Sensitivity analysis on GlaTE model | |
| GlaTe average (unconstrained) | 507 |
| GlaTe standard deviation (unconstrained) | 75 |
| GlaTe average (constrained) | 451 |
| GlaTe standard deviation (constrained) | 4 |

## 4 Discussion

We discuss a methodological approach for jointly interpreting models and sparse GPR data, with specific reference to the results obtained on the Rutor glacier, which had a poor estimate of glacier thickness due to the scattered GPR data (Villa et al., 2008). GPR has been the favorite tool to investigate glacier thickness in the last decades; however, we observed poor performance on temperate, meltwater-rich glaciers. At the same time, models that estimate glacier thickness based on surface topography and ice-flux equations are now of widespread use, and have been used to perform regional studies, for example, on the Austrian glaciers (Helfricht et al., 2019) and on Swiss glaciers (Grab et al., 2021), the latter including large-scale GPR surveys. We focus on a specific challenge of such general studies: the not uncommon situation of dealing with very scattered and unclear GPR data on temperate glaciers. We compare different ice-thickness models and literature products, and evaluate how they can help GPR analysts before and after a new GPR survey. Then, we discuss the subjectivity in interpreting GPR data, which can bring to possible human errors, especially in difficult contexts, and how models can be integrated in the interpretation process. This subjectivity is intrinsic to GPR data interpretation, but may raise concerns; therefore, we discuss the different aspects of this issue as they apply to our situation.

### 4.1 Comparison of the three ice-thickness models

The three different models, in their first run without GPR constraints, gave fairly similar estimates of ice volume (OGGM = 630; GlaTE = 510; GlabTop2-Py = 580; Original-GlabTop2 = 470 million $m^3$), with an average of about 540 million $m^3$. In particular, the glacier bedrock shape was similar. Setting the resolution of the models at 20 m avoided the exaggeration of bedrock overdeepenings that resulted from a too-fine resolution. Note that, in the Original-Glabtop2 model, the algorithm automatically resampled the data to 75m resolution to perform the processing, and the final results was resampled back to 20 m. Observing the distribution of standard deviation, the most uncertain areas are those with overdeepenings. The OGGM estimates were qualitatively similar to those of GlaTE and GlabTop2-Py (which both use the perfect plasticity method), except in some minor parts, despite their differences in the algorithms employed. The OGGM higher estimated volume is probably due to its mass balance calibration against the consensus estimate (Farinotti et al., 2019), the latter estimating a volume of about 630 millions $m^3$ (Table 1 and Figure A9). The consistency between different models, especially in terms of the general shape of the glacier, was already observed in a previous review (Farinotti et al., 2017), which demonstrated that models based on surface topography can provide fairly reliable estimations of ice thickness. They can provide gross estimates of ice volume or future position of lakes in place of glacier overdeepenings (Viani et al., 2016), with less effort in terms of input data and computational time compared to a GPR survey. However, most of the models performed poorly near the margin of the glacier: here, the thickness was generally overestimated compared to the GPR data. We suppose that this fact could be due to the simplification of the glacier geometry and interpolation procedures used by the models, which work better in elongated-shape valley glaciers, but perform worse in wider and more complex glaciers such as the Rutor glacier. The Original-GlabTop2 model was the only one that consistently showed smoother ice-thickness variations near the glacier margins (well visible, for example, in the left part of Figure A6.

## 4.2 Joint interpretation of GPR and ice-thickness models

GPR and ice-thickness models were interpreted together across all the GPR sections of the Rutor Glacier, which were both helicopter- and ground-based. The various GPR sections had a different degree of strength of the bedrock return. In some of them, the ice-bedrock interface could be followed relatively easily, such as section 2012 - 8 (Figure A3). In that case, the four models (GlaTE, GlabTop2-Py, Original-Glabtop2, OGGM) were generally consistent with the ice-bedrock interface depicted by the GPR reflections, demonstrating the overall good quality of the retrieved ice-thickness, already proven by Farinotti et al.

(2017). In most other sections the bedrock reflection was rapidly lost below about 50 m of depth. In general, the signal suffered from scattering, due to the many reflection events distributed throughout the glacier, except in some areas near the margin. This phenomenon made the interpretation of these sections challenging. In this sense, the ice-thickness models were fundamental to retrieving the bedrock topography where it could not be detected by the GPR and to avoid misinterpretations. In particular, in the parts of the profiles where all four models indicated the same ice thickness, we were more confident in picking that ice

thickness as the correct one, provided that it was a reasonable interpretation of the GPR data, although scattered. No particular improvement was seen when using a ground-based 40 MHz GPR antenna compared to the helicopter-based 70 MHz antenna. Specific comments on the interpretation of each GPR section are available in the Appendix section, which we invite the readers to look at, because they are different examples of how problematic GPR profiles were interpreted thanks to this methodology, and which kinds of mistake were avoided.

Mainly, this joint interpretation prevented the mistake of interpreting the first non-reflective layer (white in the GPR sections) as ice and the first reflective zone (scattered black) as bedrock. The deepening reflection on the right side of Figure 3 clearly shows that the ice-bedrock interface is not related to the scattered reflective zone observed at 10-50 m depth everywhere in the profile. Manually picking the ice-bedrock interface, guided by the estimates from the models, was particularly helpful, especially below 50 m, where the GPR signal was too attenuated. A possible source of uncertainty in detecting the correct

ice-bedrock interface may also be due to off-nadir reflections, i.e., from valley sidewalls. However, we do not think this issue is critical in this glacier, which is large and has no steep side valley walls.

The ice thickness estimated with the second run of the GlaTE model constrained by GPR data is about 450 million $m^3$, which is close to the GlaTE estimate without GPR data (510 million $m^3$). Partially, the estimate of the GlaTE constrained model is lower because the models tended to overestimate the thickness near the glacier margin, an issue that was mitigated by the GPR

data used in the constrained model. The final estimate of 450 million $m^3$ is almost three times larger than the 150 million $m^3$ previously calculated (Villa et al., 2008). Two literature ice-thickness products were compared to this result, the consensus estimate by Farinotti et al. (2019) and a velocity-based model by Millan et al. (2022). Those models showed consistency between them in terms of volume estimate (both about 630 million $m^3$, even though the bedrock shape was quite different (Figure A9). The higher volume estimate compared to the GlaTE model is probably due to the fact that these products still

rely on the 2003 RGI outline (RGI Consortium, 2017), which is outdated due to the rapid shrinking of the glacier. A question arises: should those ice-thickness products be used instead of running the three models, for the same goal of helping to select the correct bedrock interface in the GPR radargrams? Using these existing ice thickness model products to help interpreting

the GPR data would be simpler than running the three models as performed in this study. However, these regional scale models may have outdated glacier outlines, which are changing rapidly, and therefore thay may provide unreliable estimates in the ablation zone. In both cases of using literature products or running own models, it is important to consider the associated uncertainties.

## 4.3 Advantages and limitations of the methodology

Without the help of ice-thickness models, only a small part of the GPR profiles clearly identified the bedrock. This is not an ideal situation to draw a thickness map of the glacier, and it is argued that a similar interpretation problem could have arisen in the 2006 dataset analyzed by Villa et al. (2008), explaining the unexpectedly shallow thickness calculated there. In the GPR profiles, we highlighted in purple the possibly "incorrect" pickings, which mainly represent the clutter zone easily misinterpreted as bedrock, without the model guidance. This clutter phenomenon has been previously reviewed by Rutishauser et al. (2016), which observed that, depending on the glacier, only from 12 to 69% of the bedrock could be identified. This review was performed on Swiss glaciers, which have climatic conditions similar to the Rutor Glacier. Moreover, better results were achieved only in alpine glaciers where cold conditions are more widespread. Therefore, including modeled ice thickness to guide the GPR interpretation could represent a tool to increase the "pickable", or "selectable", regions of scattered GPR sections. This methodology can help especially in the areas where generally GPR is more difficult to interpret: where the ice is thickest, where the bedrock slope is highest, and where there is englacial water (Rutishauser et al., 2016). In the Rutor glacier, we probably have all of these three issues. Another issue of GPR surveys on glaciers that this methodology helped to solve is the limited spatial resolution. The spatial coverage of GPR surveys is limited by survey speed, time and access (e.g. crevasses), leading to discrete, limited sampling of the glacier bed. It is, therefore, possible that the maximum ice thickness remains unknown due to limited survey coverage. Spatial resolution and speed of investigation have always been a compromise. By using the constrained GlaTE model, the GPR analyst can provide a better ice-thickness estimate in those zones not surveyed by GPR, without having to rely on pure interpolation.

The integration of GPR and models seems the most viable option to provide a more reliable estimate of ice thickness than one method alone, especially in the absence of costly boreholes that intercept the bedrock at depth. This is a great achievement, because those models are open-source and require low effort to use, and their reliability and comparability with GPR data has been observed in this and previous research (Farinotti et al., 2017). They also can complement the design of a GPR survey to select the best antenna frequency, based on the expected ice depth, alongside forward modeling to produce synthetic data (MacGregor et al., 2021). Other geophysical surveys, such as Electrical or Seismic tomographies, could also be employed, but to reach more than 100 m of ice thickness, as needed in this glacier, one would need to set up very long profiles (>500m) and use powerful seismic or electrical sources. These are not as logistically convenient as the GPR.

However, some drawbacks have to be considered.

First, the difficulties in interpreting the GPR data, which represent the most significant source of possible errors and subjectivity; second, inaccuracies in the estimates provided by the different models, which are a simplification of the real system, and rely on estimates of many parameters. However, the most delicate aspect which may limit this methodology is the balance

between the reliability of GPR and that of the models. In fact, our approach is a joint interpretation of two different methodologies, in a context where data are of indirect nature and subjective choices must be made. We need the information of the models to better understand the scattered and sparse GPR data, and vice versa, in a sort of manual iterative interpretation process. We stress that, in this difficult context, we do not see the GPR as the main technique, nor the models only as a supporting information, rather, we see that they both support each other. However, the subjective interpretation is potentially prone to human error. In our opinion, the key to minimizing this potential error is to perform simulations on multiple models, display the GPR sections with multiple color scales and in both 2D and 3D environments, and acquire local knowledge about the studied glacier. In this regard, we performed a small, qualitative inter-analyst comparison experiment at a conference in Rome, September 2025 (https://www.gpritalia.it/gpritalia2025/). In this experiment, after explaining the same methodology described in this paper to the audience (GPR professionals and academics), we gave the participants a printed version of the GPR profiles, asking them to pick the bedrock of all the profiles, before and after seeing the model estimations. The 10 participants to this activity mostly picked the clutter zone as bedrock (similarly to the clutter zone indicated in Figure 3 with red arrows). With this type of picking, the total glacier volume would be greatly underestimated, as in the previous work of Villa et al. (2008). Those (unfortunately few) who also reported on their papers the bedrock picking after seeing the models showed that they recognized potential deeper bedrock reflections, closer to the models. We received positive feedback from the participants, who were concerned of the difficulty in interpreting GPR profiles on temperate glaciers, and glad that there was a complementary tool to estimate the glacier thickness. One comment pointed out the possibility of using this kind of activity as a teaching tool to stress the importance of data interpretation and local knowledge of the survey sites.

Another source of error is the glacier perimeter, which is an important input of the models, because it was retrieved with a manual observation of the aerial orthophoto. In many glaciers, the perimeter is not straightforward to draw, due to debris hiding the ice (Santin et al., 2023). Moreover, the fact that the DEM and the GPR surveys were carried out at different years required a time interpolation of the ice thickness, to take into account the progressive melting of ice, which introduced another deviation from the real value of ice thickness. The combination of multiple models and methods was key to minimizing the errors; however, the uncertainty remains a main factor in this kind of analysis and it is also difficult to estimate accurately (Aguayo et al., 2024; Shahateet et al., 2025). We observed a standard deviation of about 100 million $m^3$ of ice, which corresponds to an average ice thickness of about + - 13 meters, by comparing the four unconstrained models; we observed instead a standard deviation of about 75 million $m^3$ by comparing the unconstrained GlaTE model with varied glaciological parameters (sensitivity analysis). These values represent a reasonable estimate of the potential standard deviation in the ice thickness, but they are still biased by the choice of models used and the choice of the parameters to be varied in the sensitivity analysis. Moreover, the definition of standard deviation itself is probably not completely applicable in this case, since it would need a Gaussian error distribution for the ice thicknesses estimated by the various models, which is probably not true. Interestingly, the standard deviation of ice volume calculated during the sensitivity analysis of the constrained GlaTE model is particularly low (about 5 million $m^3$), meaning that the constrained model is quite robust to variations in the input glaciological parameters and tries to adhere to the input GPR data, even if the given GPR data were quite sparse and they were assigned an uncertainty of 30 % in the GlaTE input parameters. It is difficult to say if this very low standard deviation is indicative of real reliability

of the GlaTE constrained model; nevertheless, 450 million m$^3$ of ice, distributed as in Figure 7, represents the best estimate of the Rutor glacier thickness we could obtain with this integrated methodology.

A final and obvious limitation of the study is the absence of any borehole to confirm either the GPR and the model estimated thicknesses. Future research may consider analyzing glaciers where such borehole data is available. Nevertheless, in some decades, due to the ongoing glacier melting, we might directly see the current glacier bedrock in most parts of the glacier. These observations, together with the analysis of DEMs at different years, now possible due to the widespread satellite observations, will provide accurate geodetic mass balance information and surface topography that will deepen the knowledge about ice-flux behaviour in a non-equilibrium state and better calibrate the ice thickness models.

## 4.4   Future applications and perspectives

For different reasons, both the GPR and ice-thickness models have non-negligible accuracy limitations in reconstructing glacier geometry. The GPR suffers from scattering in temperate glaciers, while models alone have a high uncertainty based on which parameters we choose as input. This study highlights the benefit of combining the two worlds. For GPR applications, the use of models can help in designing the survey and selecting the proper instrumentation (e.g., antenna frequency) based on the estimated geometry of the glacier. Models can also provide a picture of the glacier in those zones inaccessible to GPR for logistics limitations or time constraints. For modeling studies, including GPR surveys in their workflow can improve their estimations, especially for those glaciers with complex and difficult-to-model geometries. Thus, applying the workflow presented here to other glaciers with GPR data available could help calibrate regional models more accurately, or at least raise awareness of the potential uncertainties of the models. The supplementary materials can be used to reproduce the same workflow on other glaciers.

Future integration of GPR data in glacier models could include not only ice thickness, but other features of the glacier which are detected by GPR and are very difficult to predict with models, such as the distribution of cavities and the distinction between cold and warm ice zones inside a glacier. The latter, according to Reinardy et al. (2019) and Comiti et al. (2019), plays an important role in regulating the sediment transport at the glacier base and can be retrieved from the distribution of reflections (due to englacial water) in the GPR sections. This could enhance the link between glacier geometry and hydrological output in a glacier model. The clutter itself in the radargram can be used to infer glacier features (Scanlan et al., 2020).

From a local perspective, a more robust reconstruction of the Rutor glacier compared to previous estimates will be helpful for ongoing studies on the water mass balance and sediment transport within the Rutor basin. Since the Rutor basin hosts many proglacial lakes, the bedrock topography map reveals the possible position of future lakes, in place of the overdeepenings of the bedrock topography (Figure A8). Their location is consistent with the estimates of a previous study (Viani et al., 2020). The shrinking of Rutor Glacier is speculated to occur mostly in the following decades, given its volume of about 450 million m$^3$ and its loss of 100 million m$^3$ from just 2008 to 2021. After a few decades, little ice is expected to be still stored in the Rutor Glacier.

# 5 Conclusions

Investigating glacier substructures with GPR may be challenging in temperate glaciers, where widespread water content and debris cause signal scattering, making it difficult to distinguish the ice-bedrock interface. The Rutor Glacier had already been surveyed with GPR in the past but, due to these interpretation difficulties, was estimated to have a very small ice thickness, of about 17.5 m on average. This estimate proved to be wrong after observing that, in the 2010-2020 decade, the Rutor Glacier lost more than 20 vertical meters in 1/4 of its area, while reducing its area only by a fraction.

The analysis of two new GPR datasets from 2012 (helicopter-based) and 2022 (ground-based) confirmed the difficulty of reliably detecting the ice-bedrock interface. Therefore, the Original-Glabtop2 and the open-source GlabTop2-Py, GlaTE, and OGGM models were tested, to understand how they could support the interpretation of difficult datasets acquired on temperate glaciers. First, these models were run with only the glacier surface topography as input. Then, their estimated thickness was overlapped with the GPR sections, providing substantial help in manually selecting the ice-bedrock interface. In particular, this methodology avoided the misinterpretation of englacial water-rich areas as the ice-bedrock interface. Finally, a second run of GlaTE produced an ice-thickness model of the Rutor Glacier constrained by GPR data.

Analyzing the ice-thickness variations among models and by varying the glaciological parameters of the GlaTE model, we recognized that the uncertainty in both GPR and models is a major factor and it is difficult to measure a true value of ice thickness in a temperate glacier.

A preliminary run of two or three ice-thickness models, such as the ones tested in this study, is advised before carrying out a GPR survey on a glacier. The ice-thickness models, in combination with the GPR, are the most cost-effective way to represent, with a certain degree of uncertainty, the glacier bedrock geometry. This is in line with the philosophy behind the GlaTE model, built to constrain a topographical-data-based algorithm with "ground-proof" GPR data. An effort was made to provide supplementary materials which can be used to reproduce the same workflow on other glaciers. Local studies can benefit from more accurate glacier geometry estimates, as well as regional studies, which can be calibrated more effectively.

*Code and data availability.* All the data, codes and detailed reproducible workflow is available the Supplementary materials. We organized the Supplementary materials in 9 folders, each for each step of our workflow, each with a readme.txt file to describe the folder's contents, similar to a little methodological section. We invite the reader to download them and explore the proposed methodology on other glaciers.

The 2012 GPR dataset is available on a Zenodo repository at: https://doi.org/10.5281/zenodo.8027417

The 2021 DEM used for the models, together with the ortophoto used to draw the glacier margin, is available on a Zenodo repository at: https://doi.org/10.5281/zenodo.7713299

Links at the main codes websites:

OGGM: https://oggm.org/

GlaTE: https://gitlab.com/hmaurer/glate

GlabTop2-Py: https://glabtop2-py.readthedocs.io/en/latest/index.html

*Author contributions.* Conceptualization: A.V.; Data Curation: A.V.; Formal Analysis: A.V.; Funding Acquisition: A.G. and D.F.; Investigation: D.F.; Methodology: A.V., A.G. and D.F.; Project Administration: A.G.; Resources: A.G. and D.F.; Software: A.V; Supervision: A.G.; Validation: A.V.; Visualization: A.V; Writing: A.V..

*Competing interests.* The authors declare that no competing interests are present.

*Acknowledgements.* Thanks to Umberto Morra di Cella and the regional environmental protection agency of Aosta Valley, Italy (ARPA Val d'Aosta) for the 2012 dataset. Thanks to Fabio Villa for his help in understanding the Rutor Glacier GPR and topography datasets. Thanks to Maurizio Ercoli, Emanuele Forte, Michele Freppaz and Chiara Colombero for their precious suggestions on the manuscript. Thanks to Myrta Maria Macelloni, Isabella Pisoni, Elisabetta Corte, and Alberto Cina for their help with the topographical data and for the 2021 Digital Elevation Model. Thanks to "CC-Glacier lab" of the MIUR project "Department of excellence" at the Politecnico di Torino—DIATI for

funding part of this research. Thanks to Hansruedi Maurer and Melchior Grab for their help about the GlaTE model. Thanks to Emanuel Huber for his support about the RGPR open source software. Thanks to LeldeBry, mainainer of GlabTop2-Py Github repository, for his help with the model. Thanks to Horst Machguth for sharing the original GlabTop2 model, running in IDL, as used in Frey et al. (2014). Thanks to Vincenzo Castigli for his helpful comments on the visualization of GPR profiles and his feedback about data interpretation subjectivity. Thanks to the reviewers and editors of this manuscript, who put a lot of effort and care to make this paper improve.

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

## Appendix A: Appendix - GPR Sections

This appendix is devoted to all the GPR sections analyzed in this work, to show the interested reader what the data looked like, which were the difficulties, and in which cases the three models' predictions were useful in avoiding clear misinterpretations.

In any case, the interpretation subjectivity is high, and analyzing the openly available GPR dataset with specialistic software is advised. The first 5 figures represent the 5 GPR sections of the 2012 helicopter-based survey; the last figure is the merging of all (subsequent) sections of the 2022 ground-based survey. Purple numbers on top of the figures indicate where that profile crosses other GPR profiles. See Figure 7 for the location of the GPR sections.

Additional figures at the end of the appendix show: a 3D visualization of models and GPR in Paraview (Figure A7); a map of
the bedrock and surface topography as in the constrained GlaTE model, visualized in contour lines (Figure A8); a comparison with literature ice-thickness products (Figure A9).

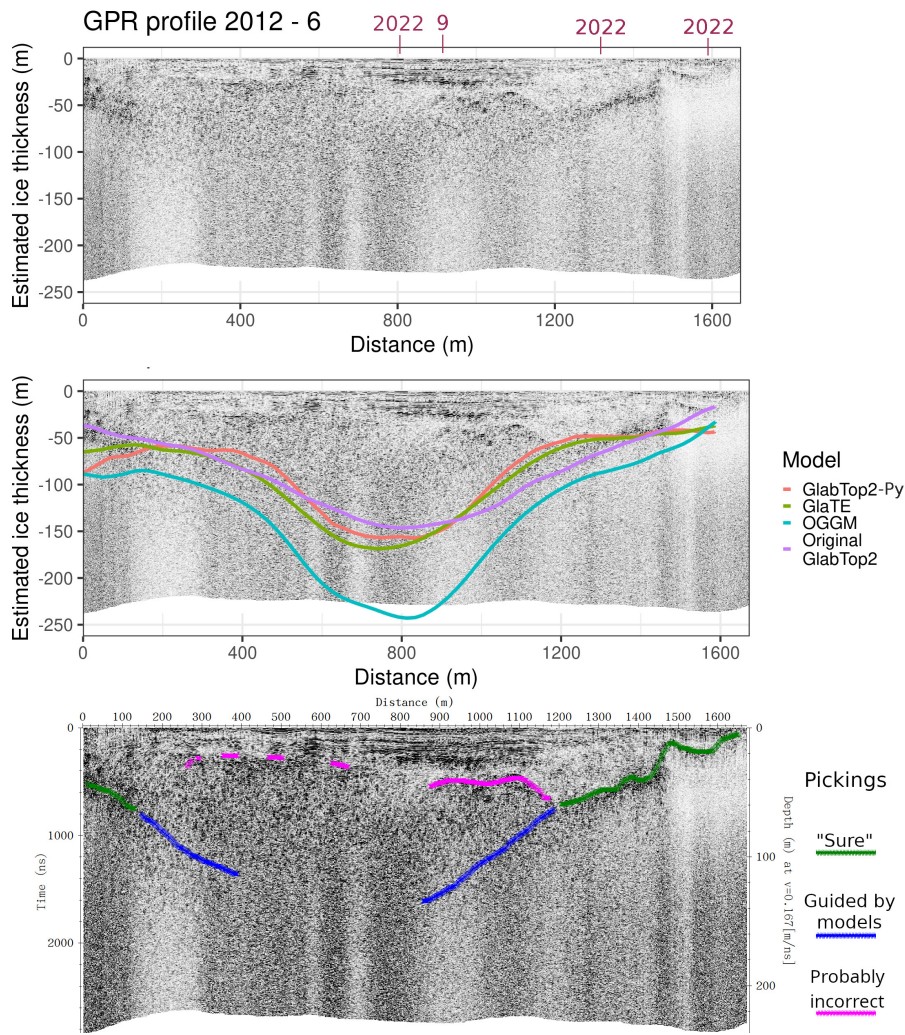

**Figure A1.** GPR section 2012 - 6. This section is situated at the top of the glacier, and in its central part (500-1000 m of Distance coordinate) the three models estimate an overdeepening of over 150 m. The GPR reflections are reasonably clear until 50-70 m of depth. In the right part of the picture, the models follow closely the GPR reflections; however, they are quite imprecise just near the margin, where they do not draw correctly the bedrock shape where the ice is very thin. Based on the estimates of the models, the analyst did not trust the sparse reflections at 25-30 m of depth as an ice-bedrock interface but acknowledged that, in the right part, the deepening reflections continue to go deeper leftward to around the center of the image.

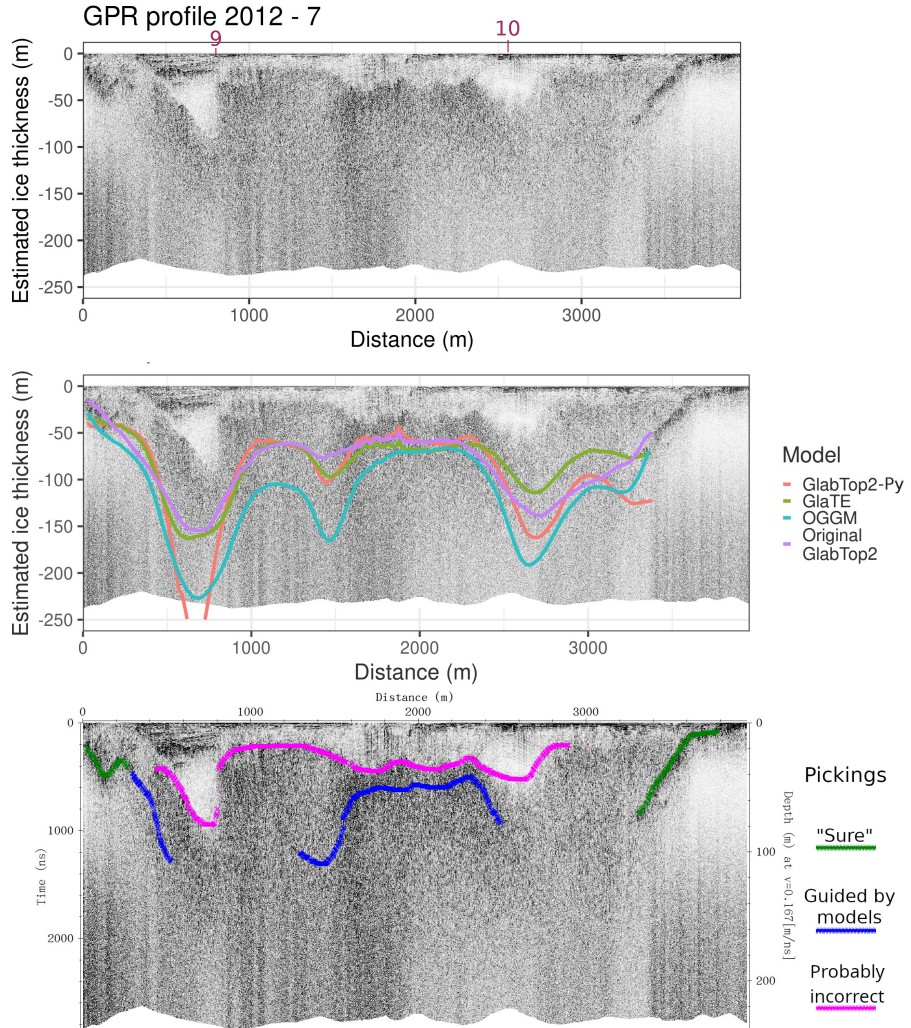

**Figure A2.** GPR section 2012 - 7. This section was already shown and discussed in Figure 6, but it is possible to add a consideration on the steep overdeepening "seen" by the GlabTop2-py model. The creation of deep overdeepenings was recognized to be an artifact of such models, especially when using a too-fine input DEM [Maurer and Grab, personal communication]. Also, the other models estimate a high ice thickness at that point, but they seem more realistic. Unfortunately, the GPR was of little help in that region.

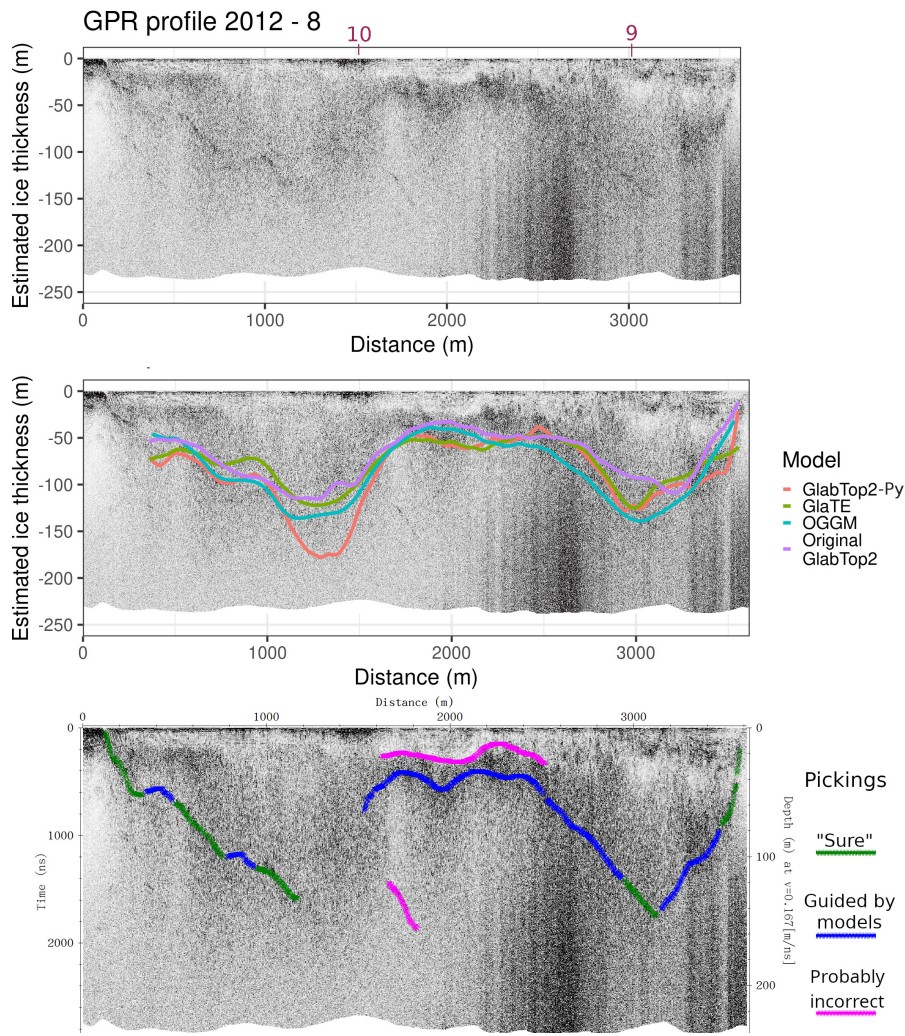

**Figure A3.** GPR section 2012 - 8. This GPR section was probably the most clear of the dataset. The ice-bedrock reflector can be followed to great depths, although the clarity is far from ideal. This section proves that the models can reasonably estimate the ice thickness of the Rutor Glacier, because they match with GPR data where the latter are clear. In the left part of the image, which corresponds to the Eastern tongue, the models are not shown because they were cut with the 2021 margin. From 2012 to 2021, the glacier experienced a notable retreat in the Eastern tongue.

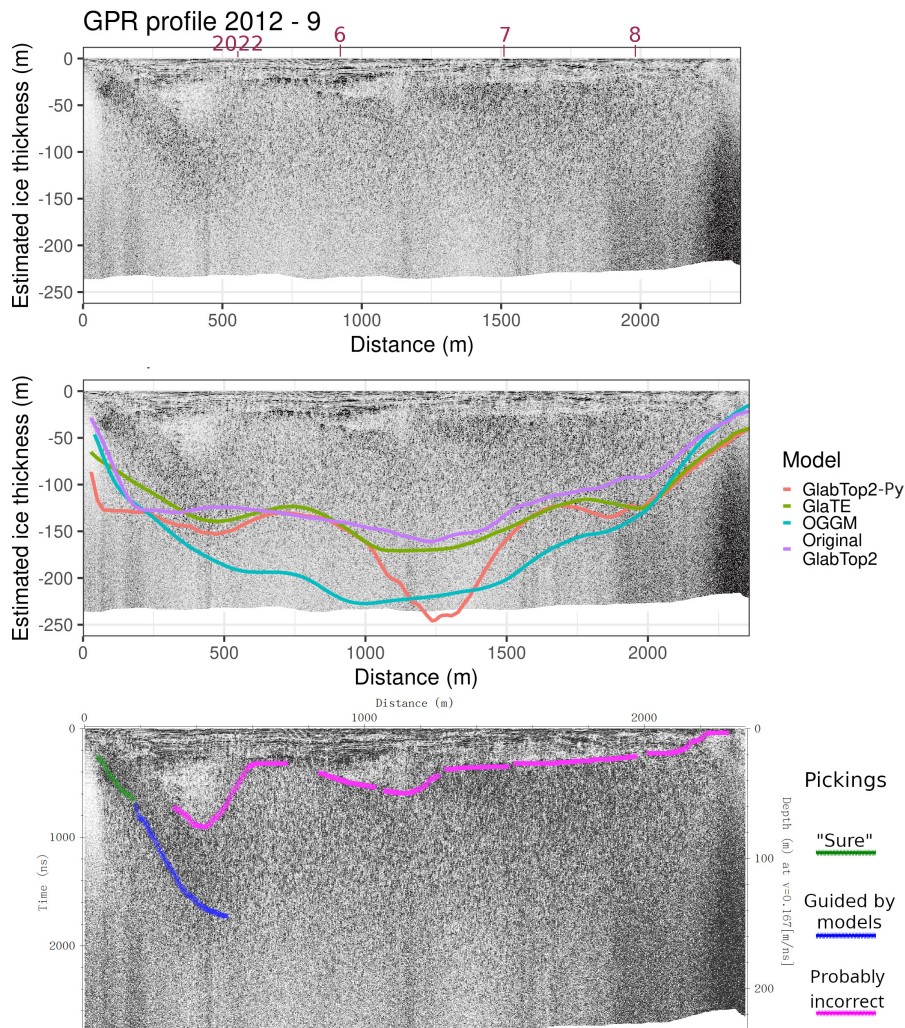

**Figure A4.** GPR section 2012 - 9. This GPR section was probably the least clear. Its path runs along the elongated overdeepening of the top of the glacier (Figure 7). In the left part of the picture, the reflections deepen very fast, and this is displayed also by the models, but after that, it was impossible to retrieve any other reflections attributable to the ice-bedrock interface. The role of the models, in this case, was very important, to avoid thinking that the darker reflections seen in the center-right part of the image at 30-50 m depth are the bedrock interface, while probably it is due to the higher englacial water content, or debris (Forte et al., 2021).

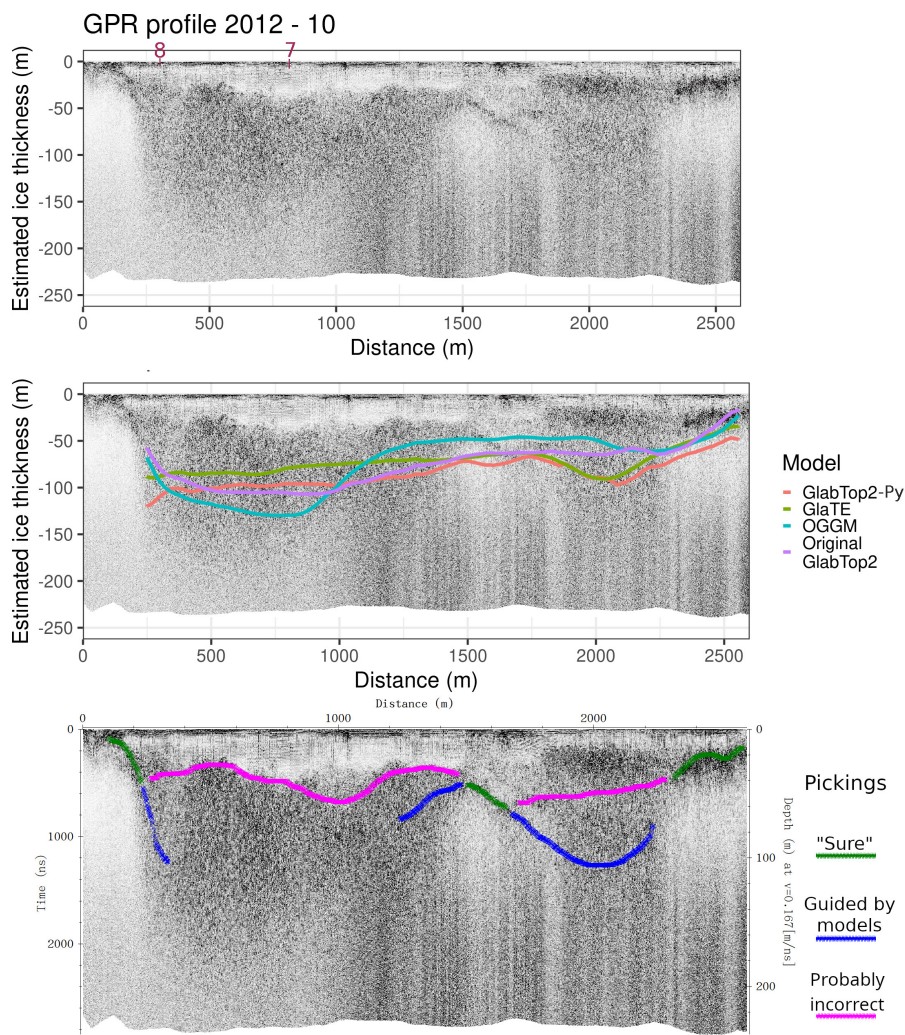

**Figure A5.** GPR section 2012 - 10. This GPR section was generally clear in the right part of the image, and also showed consistency between GPR and models. All the profile has an interface around 30-40 m interpreted as englacial water or debris (Forte et al., 2021). The left part of the picture was more problematic, because the GPR, although not clear, seems to suggest a deeper bedrock interface than the models. Probably, this is due to a peculiar location of this survey line, possibly running along a very high-sloping bedrock (Figure 7). Such high-slope bedrock areas are known to disrupt GPR measurements and they could be common near the overdeepenings of Rutor Glacier since they are also evidenced in many locations in the proglacial zone (which was formerly occupied by ice during the past ice ages).

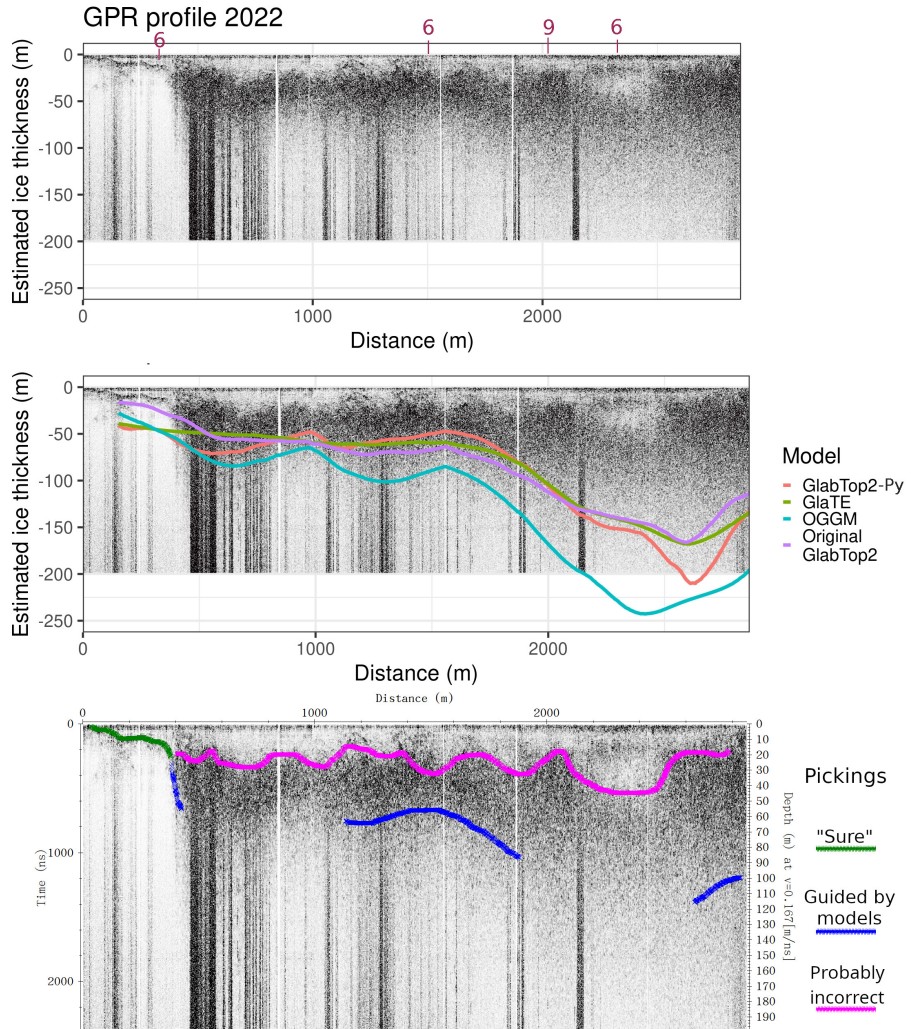

**Figure A6.** GPR section 2022. This GPR section was acquired with an antenna (40 MHz) different from that used for the previous sections shown. It also was ground-based and not helicopter-based. Notwithstanding the lower frequency and being ground-based, the GPR reflections were not particularly clearer, showing a physical limit of the technology in the presence of widespread englacial water or debris (Forte et al., 2021). The models did not perform well near the margin, with the exception of Original-Glabtop2. as observed also in the other GPR sections. However, probably the models offer a reasonable estimate of the bedrock interface in all the left-central part of the image. The right part seems more problematic because the GPR signal is completely lost, however, its greater depth is supported by the GPR signal texture: compare e.g. the image at 100 m depth at Distance = 1000 m and = 2500 m. At distance = 2500 m, the image shows a granular texture similar to where there is ice and englacial water, while at Distance = 1000 m the texture is transparent. Similar considerations were used to interpret the GPR sections by Forte et al. (2021).

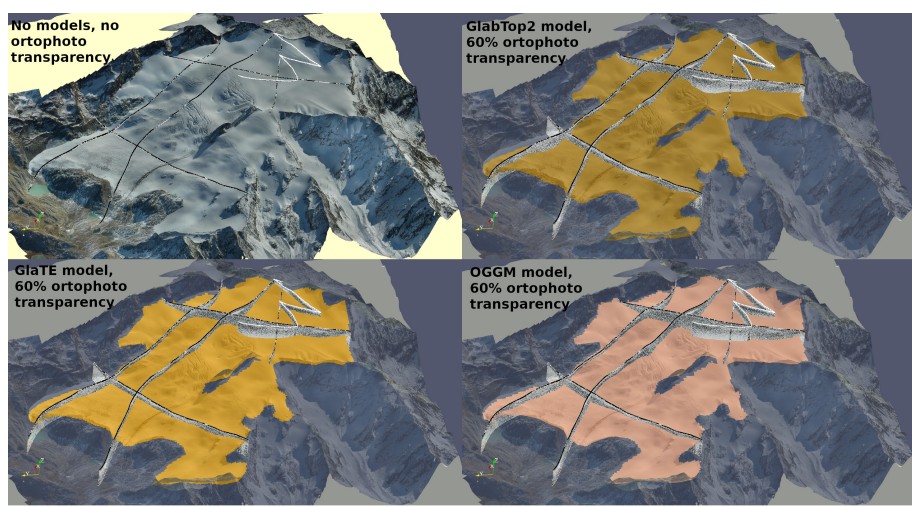

**Figure A7.** 3D visualization in Paraview of GPR profiles and bedrock as estimated by the three models. This 3D visualization, even if difficult to render in a 2D image, was helpful to see the GPR profiles at their intersections, where their bedrock interface must match.

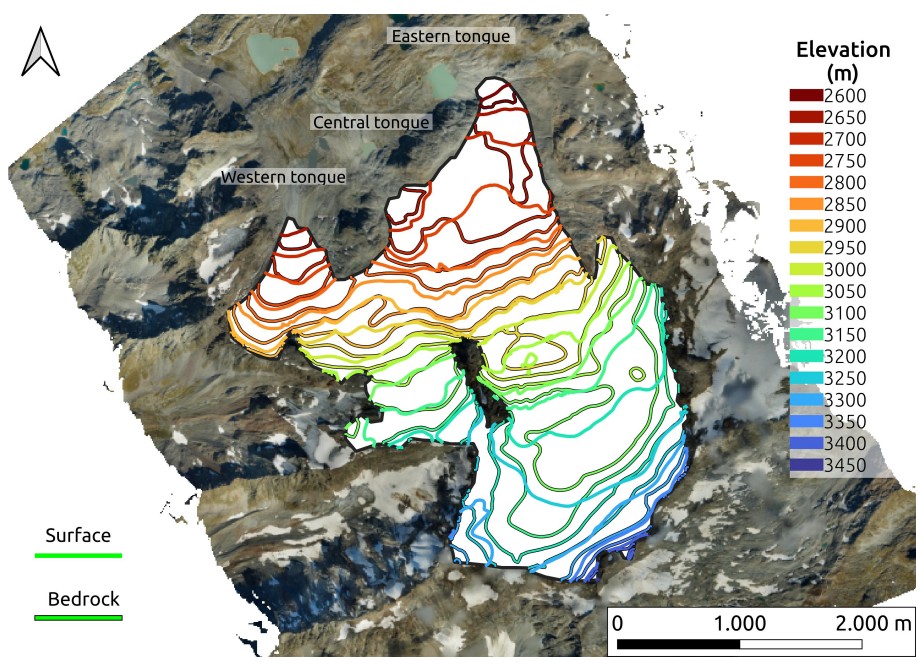

**Figure A8.** Bedrock and surface contour lines as in the constrained GlaTE model.

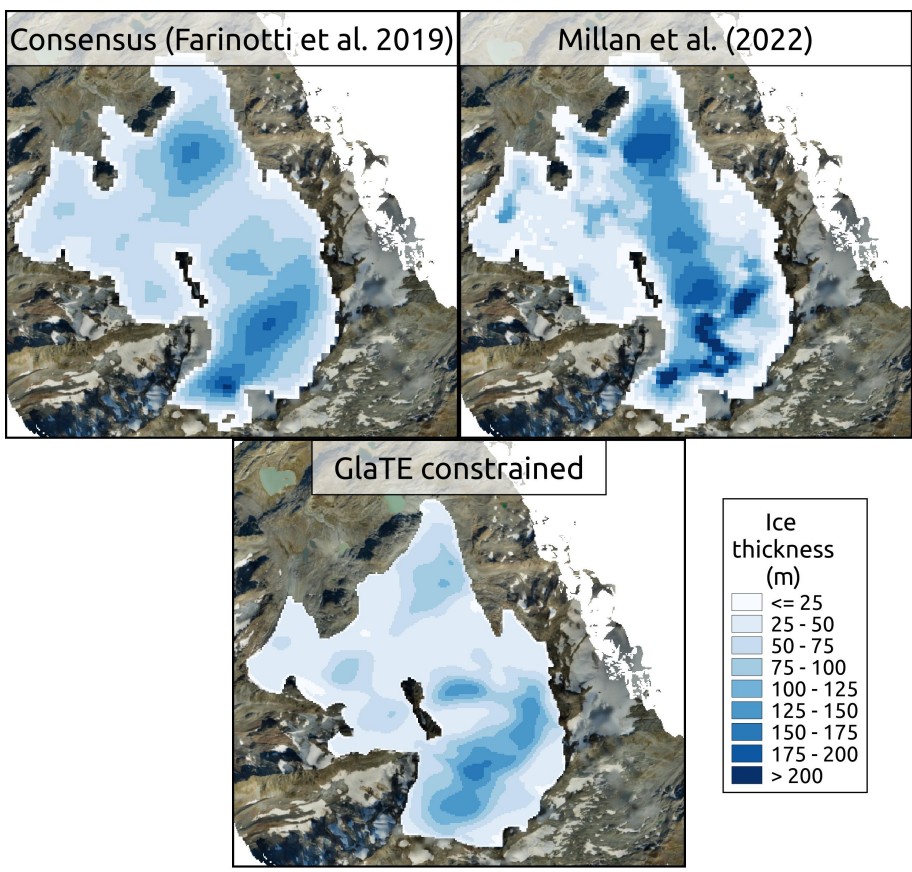

**Figure A9.** Comparison between the ice-thickness product of the constrained GlaTE model and two literature products: an ice-thickness map by Millan et al. (2022) and the consensus estimate (Farinotti et al., 2019).