# Peer review of "Integrating GPR and ice-thickness models for improved bedrock detection: the case study of Rutor temperate glacier"

_EGUsphere, 2024_

## Referee Comment (RC1)

Review of the article entitled:

**Ground penetrating radar on Rutor temperate glacier supported by ice-thickness modelling algorithms for bedrock detection**

This study addresses the challenges of measuring ice thickness in temperate glaciers, such as the Rutor Glacier in the Southern Alps (Italy), in these glaciers the ice is at or near the melting point throughout its entire mass, including both the surface and deeper layers. This means the glacier contains both ice and meltwater, which makes it sensitive to temperature changes and contributes to faster melting. Meltwater then interferes with the clarity of Ground Penetrating Radar (GPR) signals. To improve the manual selection of bedrock profiles from GPR-based ice thickness measurements in such glaciers, the researchers combined GPR data with three open-source thickness inversion algorithms (GlabTop2, GlaTE, and OGGM), which estimate ice thickness based on surface topography and mass turnover. These models guided the manual selection of unclear or scattered GPR data for the Rutor Glacier. The study analysed two new GPR datasets and produced a more accurate ice-thickness map using GlaTE (one of the algorithms, after selecting the correct bedrock profile with the aid of outputs from all three models). Authors then conclude that the glacier stored about 515 million cubic meters of ice in 2021, significantly higher than previous estimates. The authors claim that this methodology is replicable and can simplify future GPR surveys of temperate glaciers, particularly in noisy data conditions caused by meltwater.

Overall, the manuscript, methods, and results are well explained, however, I have several corrections to the current text. I find the study very creative and could have potential for the use of this type of data to validate and calibrate ice thickness inversion algorithms. However, my main concern lies with the novelty of the study and significant gaps in the methodology, such as the uncertainty quantification of model results and the use of OGGM in such a small-scale glacier-specific study, as well as not providing details of the set up used for the OGGM inversion. I could consider this study for publication, but only after the authors address my questions and make the necessary changes to the manuscript.

**Major comments:**

Novelty, Reproducibility, and Scalability

- After reading the manuscript, I find it difficult to see how this analysis effectively contributes to the broader challenge of providing ice thickness observations and distribution products that could be used to constrain, evaluate, or train model simulations, or new deep learning algorithms (e.g., The Instructed Glacier Model).
- Additionally, there is no discussion on how this method could be scaled to a regional level. Expanding the study across multiple temperate environments and numerous glaciers with GPR measurements (e.g., in Alaska, the Alps or South America) raises concerns about the efficiency of manually detecting multiple profiles. Such an expansion would likely require a more robust approach for parameter calibration, validation of ice thickness inversion by each algorithm, and the generation of a final thickness map using more than one algorithm.

  I am concerned about the scalability of this method, as the parameters used for this specific glacier may not be transferable to others. This approach heavily depends on both data quality and the accuracy of model outputs. The examples presented in the manuscript show that the three algorithms perform well on some profiles but less so on others. A sensitivity analysis of the algorithms, with varied parameters to assess their impact on the thickness profiles, would have been valuable.

- Additionally, a more detailed uncertainty estimation for the final thickness product is needed. This could have been addressed by combining the results from all three algorithms, not just GlaTE, and providing a standard deviation on the final ice thickness map, while also comparing the results to existing ice thickness inversions and volume products (e.g., Millan et al. 2022, Farinotti et al. 2019 and Cook et al 2023 – all available in OGGM).

Methodology:

Related to the data input used:

- Regarding the GPR survey conducted by helicopter, I wonder if the authors need to correct for signal reflection from the nearby mountain terrain and elevation changes - i.e., interference caused by the surrounding mountain slopes in the radargram?
- Is the outline from the Randolph Glacier Inventory (RGI)? If authors have used their own glacier outline this might significantly deviate from its RGI counterpart, which could introduce errors and the authors should have computed the calibration steps again in OGGM.
- In the introduction (L69) authors are using their own DEM to predict the ice thickness from all models. In that case, the authors should have re-processed the GIS task of OGGM. A detail on how they use OGGM is missing (see below).
- All data inputs and as well as the model's thickness inversion (glacier initial state) represent different timespans. Why not use the same DEM and glacier outline across all models? You can input your own DEM and glacier outline into OGGM and recompute all steps until the inversion. See the following tutorials.
- Using your own outline in OGGM
- Create local topography maps from different DEM sources with OGGM
- Step-by-Step guide to building preprocessed directories from scratch

Algorithms:

- Choice of input parameters in ice thickness inversion models: The authors should clarify that these parameters are not transferable between glaciers (see Zekollari et al. 2022). Additionally, a sensitivity study on the model parameters should have been conducted to assess the impact of parameter variation on ice thickness profiles computed by the models.
- How did the authors calibrate surface mass balance and ice thickness inversion in OGGM? or Did they use pre-processed directories? A specific workflow of the steps followed with OGGM is missing. The actual code repository of this study is not shared, thus is not possible to verify.

Results:

- It would be interesting to see a comparison of ice thickness differences between GlaTE and the other two models, along with a more in-depth discussion of the reasons behind these differences.
- The findings of the paper would be strengthened by comparing the resulting ice thickness map from GlaTE to existing ice thickness inversion and volume products (e.g., Millan et al. 2022, Farinotti et al. 2019, and Cook et al. 2023, all available in OGGM).

Discussion:

- I would encourage the authors to provide a stronger justification for how this methodology could be scaled to other glaciers and applied to existing GPR surveys in temperate glaciers. Additionally, it would be helpful to explain how this study could address the under-sampling

problem of ice thickness in temperate regions (e.g., the Andes). However, caution is needed, as once models are used to improve observations, they are no longer pure observations and here there is a "human error" element also in place with this method. The authors could emphasise that GPR measurements provide a better representation than models, especially in areas like valley walls where models may struggle due to the simplification of glacier geometry (e.g., elevation flowlines and bed geometry assumptions in the case of OGGM).

● There is little mention about debris cover which is likely not accounted for in the ice thickness inversion algorithms.

Conclusions:

● While the study is well-detailed and clearly explained, it could benefit from a stronger emphasis on its contribution to the broader challenge of ice thickness observations in temperate glaciers. The results, though valuable for this specific glacier, do not provide new insights beyond the updated GPR surveys and improved ice thickness map. To enhance the overall impact, the authors could explore a more quantitative interpretation of the results and better highlight how their findings address larger-scale issues in future research.

Minor corrections and suggestions:

**Abstract**

L1: Add an example of where temperate glaciers are located (i.e. not at the poles).

L10: Authors should explicitly state that they used model output to manually select the best bedrock profile from the GPR data in problematic survey sections, clarifying that the model output is used to fill gaps in the GPR observations along those profiles. This should be stated early on in the abstract and the introduction, to improve the objective of the manuscript.

**Introduction**

L29-30; replace "inner composition." with "present day ice thickness distribution and geometry"

L42; EM to Electro Magnetic.

L43; Suggestion: Rutishauser et al. (2016) analysed a large set of GPR data acquire on Swiss glaciers and found that depending on the specific glacier, the bedrock interface could only be successfully detected in 12-69% of the GPR data due to this scattering issue.

L48 Remove "Also". Suggestion: "Air bubbles trapped in ice cause additional scattering of the GPR signal, which helps differentiate between various types of ice…"

L51: replace "are reported in the Study site paragraph" by "are summarised in section X"

L56: replace "paper" by "study".

L58: point to a figure to direct the reader to a GPR profile to indicate the issue.

L62: replace "help the analysist" with "aid".

L64: I will just call it glacier models or ice dynamical models (they all are ice thickness inversion algorithms of some sort). Authors should pick a single definition throughout.

L67: Please cite the version of OGGM used in the study. See https://docs.oggm.org/en/latest/citing-oggm.html#software-doi

L77: statements like "it should be" introduces doubt on the results, try to avoid this type of language and quantify how much the ice thickness product improved via statistics.

**Study site**

This section is too long and I don't see how past geomorphological events are relevant to this particular study. I would start by describing the site (from L97) and georeferenced so the readers know where the glacier is geographically.

L102: Add citation of DEM's used to compute ice thickness losses.

**Methods**

L111-123: Remove all "to".

L113: Remove "and updated".

L119: Replace "to perform the manual picking" with "Manually select reflexion events"

L123: Replace "to draw a final result…" with Produce a map of the glacier ice thickness (Figure 6)

L124-127: Suggestion:

"Some topographical adjustments were necessary to assist in analysing GPR observations that span different time periods (2012 and 2022). A 2021 DEM of the glacier surface was used for the GlaTE and GlabTop2 algorithms, while the 2000 DEM was used for the OGGM algorithm. In other words, the GlaTE and GlabTop2 models represent the 2021 situation, OGGM represents 2000, and the GPR data corresponds to 2012."

What about the glacier outline date?

L131: Why do the authors not use the same DEM (or the best DEM) for all models? See above.

L124-145: This text seems a bit misplaced, I would divide the text into sections for (i) pre-processing of input data for models and (ii) post processing of model output and the describe (ii) after describing the algorithms.

Sect 3.1 Explain if the GPR data collected from a helicopter needed to be corrected for altitude changes in the survey and the scattering effects caused by the nearby terrain. See Church, G. et al. (2018).

L154: Add - The GPR data was processed by the following method:

L170-175: These lines contain irrelevant text. The increase in usage and citations of a tool or model (e.g., OGGM) does not necessarily indicate it is the best tool for a particular study. A more thorough justification for the choice of tools should be provided here. The OGGM documentation clearly states that it is designed for large-scale or regional glacier modelling. Caution is advised when using OGGM for single glacier studies, and a detailed workflow for producing the thickness inversion should be included in such cases.

L177: Remove "ice flux mechanics" and replace: ice flow theory and mass conservation.

L179: Replace all "picking" with manual selection.

Suggestion replace L181 – L184 with

"The thickness inversion models required specific input parameters to run. These were reviewed for consistency with the physical characteristics of the study area, but unless stated otherwise, default values from similar alpine glacier studies by the algorithm developers were used. See below a summary of all models".

Here, authors should state that these parameters are not transferable glacier to glacier and a sensitivity study on one profile at least should have been carried out on model parameters to see the impact of parameter uncertainty in ice thickness distribution.

Sect. 3.2

This section would benefit from a table comparing the parameters (and their values) used in all models, providing a quick overview of each model's setup, along with a column citing the publications from which these parameters were sourced.

L225: Cite OGGM version used.

L227-228: Replace "OGGM bed topography inversion algorithm" with "OGGM ice thickness inversion algorithm" … "which is based in ice flow dynamics and mass conservation (Farinotti, et al. 2009 and Maussion et. al 2019). The ice flux is computed as: … "

L234: "under the simple assumption of equilibrium". This is not correct in the case that the latest version 1.6.0 of OGGM was used. Please, note the version used in this study and how the ice thickness inversion was calculated. Do authors calibrate surface mass balance and ice dynamical parameters? Note that it is possible to calibrate OGGM to match geodetic mass balance data which removes the equilibrium assumption. In the latest version is possible to calibrate the glacier mass balance and ice dynamics parameters at the same time using a "dynamic spin-up" see Appendix A and Aguayo et al. (2023) for details and the following tutorials:

https://tutorials.oggm.org/master/notebooks/tutorials/observed_thickness_with_dynamic_spinup.html#dynamic-model-initialization-using-observed-thickness-data

**Discussion**

L276: "ice thickness"? do you mean ice volume (why is this not just stated in $Km^3$)

Section 5.1 Here ideally authors should have done a better analysis on the difference between the thickness maps computed by the different models and also show a flowline profile view. Also compare the resultant volume with previous studies and estimates (see references).

A lot of this section could be removed if the authors use the same DEM and there is no need to correct ice thickness changes over time.

L302-307: Suggestion

"This joint interpretation prevented the mistake of interpreting the first non-reflective layer (white in the GPR sections) as ice and the first reflective zone (scattered black) as bedrock. The deepening reflection on the right side of Figure 3 clearly shows that the ice-bedrock interface is not related to the scattered reflective zone observed at 20-40 m depth. Manually picking the ice-bedrock interface, guided by the estimates from the algorithms, was particularly helpful, especially below 50 m where the GPR signal was too attenuated."

L309 "This is not far from estimates without GPR data" Quantify such differences.

L329. "previous research" add citations.

**(more comments below)**

**Figures**

**Figure 1.**

This figure needs a map of the alps with the location of Rutor glacier. Add RGIID or GLIMS ID.

Replace "how many meters it has subsided in the past decade (from 2008 to 2021)" with changes in ice thickness (m) from 2008 to 2021.

Add RGI outline as well as the outlines used in this study with different colours. Add citations.

**Figure 2.**

Dotted survey lines could be thicker.

**Figure 3.**

I would add a point of first guess from authors of where the bedrock might be if they didn't know from the thickness inversion algorithms.

**Figure 4.**

Another panel could be added to this figure looking into thickness profiles from models along the main flowline and two more figures showing the ice thickness differences between GlaTe and the other two models.

**Figure 5. (and similar)**

Add to the bottom panel the part of the profile that is taken or selected using the ice thickness inversion algorithm (i.e., fill the gap in the profile via another colour)

**Figure 6.**

Instead of displaying the GPR data on top of the thickness map, display thickness differences between GlaTE and the GPR. Or plot differences in profiles.

**Appendix.**

Authors should also compare their resultant thickness map with other estimates. See comments above, this could go in the appendix.

**References**

Aguayo, R., Maussion, F., Schuster, L., Schaefer, M., Caro, A., Schmitt, P., Mackay, J., Ultee, L., Leon-Muñoz, J., and Aguayo, M.: Assessing the glacier projection uncertainties in the Patagonian Andes (40–56° S) from a catchment perspective, EGUsphere [preprint], https://doi.org/10.5194/egusphere-2023-2325, 2023.

Church, G. & Bauder, A. & Grab, Melchior & Hellmann, Sebastian & Maurer, H.. (2018). High-resolution helicopter-borne ground penetrating radar survey to determine glacier base topography and the outlook of a proglacial lake. 1-4. 10.1109/ICGPR.2018.8441598.

Cook, S. J., Jouvet, G., Millan, R., Rabatel, A., Zekollari, H., & Dussaillant, I. (2023). Committed ice loss in the European Alps until 2050 using a deep-learning-aided 3D ice-flow model with data assimilation. Geophysical Research Letters, 50, e2023GL105029. https://doi.org/10.1029/2023GL105029

Millan, R., Mouginot, J., Rabatel, A. et al. Ice velocity and thickness of the world's glaciers. Nat. Geosci. 15, 124–129 (2022). https://doi.org/10.1038/s41561-021-00885-z

Farinotti, D., Huss, M., Fürst, J.J. et al. A consensus estimate for the ice thickness distribution of all glaciers on Earth. Nat. Geosci. 12, 168–173 (2019). https://doi.org/10.1038/s41561-019-0300-3

Zekollari, H., Huss, M., Farinotti, D.,& Lhermitte, S. (2022). Ice-dynamicalglacier evolution modeling—Areview. Reviews of Geophysics,60, e2021RG000754. https://doi.org/10.1029/2021RG000754.

---

## Referee Comment (RC2)

**Review of "Ground penetrating radar on Rutor temperate glacier supported by ice-thickness modeling algorithms for bedrock detection" by Andrea Vergnano et al. (2024)**

The manuscript presents airborne and ground-based GPR data collected in 2012 and 2022 over Rutor Glacier, a temperate glacier, which are known for challenges posed by high signal scattering and absorption. The study's novel approach combines three models (GlabTop2, GlaTE, OGGM) to help with the identification of the ice-bed interface, improving upon prior estimates that likely underestimated ice thickness due to misinterpreted scattering zones near the surface. The study concludes that incorporating the models improves the GPR interpretation in terms of ice thickness. Finally, a new ice thickness map is generated with the new GPR interpretations constraining the GlaTE model.

I think this study presents a creative approach to improve the interpretation of challenging GPR data over temperate glaciers. Overall, the paper fits the scope of the journal and has potential, but in my opinion, several major issues need to be addressed before publication. These include the need for clearer methodological explanations, particularly concerning the use of DEMs in the models. The introduction should more clearly highlight the true novelty of using models to improve GPR interpretation. Furthermore, a deeper analysis of model-assisted picking, including statistical comparisons between model-guided and unguided picks, is necessary to fully support the claim that the models "provided substantial help in manually picking the ice-bed interface". Finally, the manuscript requires substantial English language revision to improve clarity, as many sentences are awkwardly phrased or repetitive. I hope the authors find my comments useful and that they can help to improve the manuscript.

**Major Issues**

- **Language:** The manuscript would benefit from significant English editing. Many phrases are unclear or awkward, and the text could be more concise. Paragraphs often repeat information unnecessarily. I have made specific suggestions in the line-by-line comments.

- **Research focus**: The main purpose of this research as stated in the introduction is to "investigate the Rutor glacier thickness with two new GPR datasets" (L56). However, I believe that the manuscript could better highlight the key goal/innovation – using models to assist in identifying the glacier bed in GPR data. This is underemphasized in the introduction, results, and discussion sections.

- **Abstract**: I find the abstract quite lengthy, and the primary goal and key findings are not clearly conveyed. I recommend revising the abstract after the manuscript has been edited to ensure the message is concise and focused on the main points.

- **Methods:**
    - **Ice thickness change:** It is unclear whether the ice thickness change (Figure 1) from the DEM differencing is original to this study or based on previous work. If new, the method should be explained

- o **DEM use:** The rationale for using different DEMs for different models is unclear, especially why a 2000 DEM was used for OGM. I am not familiar with the models, but is it not possible to run the OGGM model with the 2021 surface topography? Additionally, why was a 2008 and 2021 DEM used for the GlaTE and GlabTop2?
- o **Ice thickness vs bedrock topography:** I understand that the models output ice thicknesses, but why not compute a bedrock DEM instead? The bedrock topography is not expected to change over the study period, and could directly be compared to the GPR data from any survey time (i.e. 2012 and 2022). Ice thicknesses can still be extracted (subtracting the bed DEM from the surface DEM). This could reduce all the ice thickness corrections that currently need to be applied.
- **Results:**
  - o **Ice loss map:** As the ice loss map supports the hypothesis of underestimated thickness, it should be included in the results.
  - o **Statistical analysis:** A more in-depth quantitative analysis is needed to assess how much the models aid in picking the ice-bed interface. This could include comparisons of ice thickness picks with and without the models, as well as how each individual model was used (e.g. for future recommendation, is there one model that stands out, instead of having to run all three?) The discussion includes some statistics (e.g. "20% of the GPR lines clearly identified the bedrock"), but it is unclear how these were calculated, and they are not included in the results.
  - o **Radargram interpretation:** The manuscript could more strongly emphasize how weak reflectors, identified with low confidence, are validated through model agreement, increasing confidence in identifying the ice-bed interface.

    That being said, I also think there are instances where the selected reflectors appear questionable, which may raise concerns about potential bias in the manual picking process when influenced by model outputs (e.g. picking noise). For example, I have difficulties identifying a reflector that was picked on
    - Profile 2012-7 between ~200-500 m
    - Profile 2012-8 between ~1500-1900 m
    - Profile 2012-9 between 500-1000 m
    - Profile 2012-10 between 300-1500 m

    This risk should be discussed explicitly (in the discussion section), as it is important to acknowledge the possibility of seeing patterns in noise when guided by models.

- **Figures**
  - o A study area overview map to see where in the Alps Rutor glacier is would be useful (e.g. integrated in Figure 1 or 2)
  - o Consider increasing the font size in all figures and remove color scale name information in the figure caption.
  - o *Figure 1*: Add elevation contour lines (or on Figure 2) and a reference to the source of the glacier outline. Also consider labelling the glacier tongues as described in the text.

- *Figure 2*: Increase line width, and consider using markers instead of "start" and "end" labels to reduce text and improve readability.
- *Figure 3*: I suggest adding arrows to indicate the "clutter zone" and "true bedrock" so the reader can follow what is meant in the text (L256-259). Also, consider removing Figure 3 as it is repeated in Figure 5, or replace it with another example (e.g. Profile 2012-8).
- *Figure 5*: I suggest using different colors instead of line-styles to better distinguish the models.
- *Figure 6*: I suggest using the same colormap for the GPR and model ice thickness for easier comparison. The GPR data can be surrounded with a white outline for contrast.
- *Appendix Figures:* I think that some of the description should be moved into the main results/discussion sections.

**Minor Issues/Line-by-line comments**

L3: I suggest removing the sentence with cold ice, it is irrelevant here.

L8-9: I suggest removing the sentence "Besides, GPR…."

L31, L36, etc.: Consider replacing "meltwater" with "englacial water content" or "water", to avoid confusion with surface meltwater generation/runoff, englacial water may also result from rain.

L32: I believe it is "pressure-melting point", not "temperature-pressure melting point".

L35-36: This also reads a bit awkward, e.g. we wouldn't expect a sudden change in geothermal heat flux. I suggest rewording to "Temperate glaciers at the pressure melting point are primed for rapid meltwater production upon small energy or heat inputs…"

L37: Specify that while high-quality GPR surveys are possible (e.g. for snow/firn near surface studies),challenges lie in detecting the bed returns. Reword to "…can challenge the interpretation of bedrock returns from Ground Penetrating Radar (GPR) surveys."

L40: Clarify "smaller-scale heterogeneities", e.g. small fractures or sediment grains, smaller than the wavelength (or quarter wavelengths/range resolution)?

L42-44: Reword to clarify what was studied, e.g. "Challenges in detecting basal returns over temperate glaciers have been studied …" Additionally, I think it would be good to mention the studies on effects of antenna orientation on detection of the bedrock reflection e.g. (Langhammer et al., 2019).

L47-48: Rephrase to clarify that englacial debris may also originate from surface material, not just freeze-on at the bed.

L49-52: I think this sentence could benefit from directly referencing some of these studies. Also consider integrating the study site description here.

L52-55: Replace "resolution" with "spatial resolution". I suggest reformulating to "The spatial coverage of GPR surveys is limited by survey speed, time and access (e.g. crevasses), leading

to discrete, limited sampling of the glacier bed. It is therefore possible that the maximum ice thickness remains unknown due to limited survey coverage."

L57-L59: Re-word for clarity, e.g. "These new datasets reveal high scattering of the radar signal over most parts of the glacier, demonstrating the difficulty in detecting the ice-bedrock interface."

L60: Include a reference for the "previous doubtful estimates of ice thickness".

L68: I believe the correct reference is (Langhammer et al, 2019a), verify other instances.

L68-70: "Thanks to … are extracted." I suggest rewording to "The ice thickness is predicted using the three models." (i.e. the DEM part belongs in the methods section).

L71-72: Rephrase to "… superimposed on the radargram to help identify the most likely ice-bedrock interface…"

L74: Replace "inner geometry of the glacier" with "bedrock topography"

L93-96: Instead of just mentioning multidisciplinary aspects/different perspectives, provide examples (e.g. glaciology, geomorphology, ecology, hydrology …?).

L96: misspelling of "multidisciplinary"

L97: reword to "… the Rutor glacier covers an area of 7.5 km$^2$ …"

L100 and others: Replace "outline" with "margin"

L101-108: Moving the ice thickness change discussion to the methods/results sections, or reference to original source if from another study.

L107: replace "extension" with "area".

L108-109: Move this sentence to the introduction for better context.

L116: Reword to "The results of this step are show in Figure 4.", or remove this sentence.

L119: Replace "reflection events" to "reflectors"

L120: Replace "limit…" with "reduce the chance of mis-interpreted bedrock reflections"

L121: Be more specific: "… surface topography and the GPR-derived bedrock topography."

L123: Step 6 does not contribute to the "overcome the difficulties in interpreting the GPR data…" as stated at the beginning of the methods section. I suggest removing this step.

L124-145: Address comments above and consider moving this section to 3.2. Clarify the glacier outline source (e.g. mentioned in L186)?

L150-151: This is repeated in Step 4, I suggest removing it here.

L154: What was the bandpass filter of for the ground-based survey? I assume it was lower than this.

L156: Clarify "correct max phase", e.g. is it a dewowing process? Also, avoid non-scientific language like "suggested by Reflexw"

L178: Replace "drive…" with "help identify the ice-bedrock interface during manual picking…"

L181-182: Reword to: "The modeling algorithms required additional input parameters (e.g. xxx). These were checked for consistency with the Rutor glacier study area, …"

L202: Remove double citation.

L207: Clarify that known ice thickness/bedrock points, not GPR data itself, are used as input. Similarly, further down, I assume $h_{GPR}$ is the GPR-derived ice thickness, not the GPR data.

L212: I suggest removing "outside"

L213: Clarify "gradient of outside terrain slope", i.e. is it the slope outside the glacier?

L226: I believe this should be "meltwater runoff"

L228: remove the "is" before "equation"

L233: precipitations (remove s)

L232-235: If a mass balance was used to estimate q, include the details on how this was determined for Rutor glacier and the value used.

L245-253: Instead of listing the figures at the start of the results section, I suggest integrating them into the text to improve the flow of the text.

L257: Replace "black reflection zone" with "strong backscatter zone" or "high amplitude zone".

L257-258: "However, on the right side of the plot, the clearly submerging ice-bedrock interface shows…" I suggest rewording the interpretation of the submerging ice-bedrcok interface to make it less definitive and more interpretative (e.g. the contrast dipping towards the center on to the left also looks like a bed return, but is not picked as such).

L260-263: Move the comparison with other studies to the discussion section. Also, the Villa et al. (2008) study used GPR data from 2006, not 2008.

L270-274: There is a lot of repetition of methods within this section. I suggest focusing on results here only.

L277-282: This section is mostly a repetition of the methods part. Move any methods to the methods section and focus the discussion on e.g. how resolution affects the result (e.g. over-deepening being an effect of fine-resolution DEMs?)

L288: Explain how the ice thickness near the glacier margin was overestimated, e.g. was it compared to the GPR data?

L291: Replace "readability" with "…degree of visibility" or "strength of the ice-bedrock return."

L298: "… more confidence was given…", it is not clear how this was implemented. E.g., do the picks come with a confidence level?

L303-305: I suggest including a discussion on the possibility of off-nadir returns (e.g. from valley side walls).

L326: What about seismic surveys?

L329: I suggest adding this citation here (MacGregor et al., 2021) (relation between frequency and ice thickness)

L337: Can we quantify "reasonably comparable models" in the results section, e.g. what is the mean, maximum, standard deviation in the differences in ice thickness predictions?

L339: misspelling of minimizing

L340: It is unclear where these uncertainty estimates come from

L343-L360: This section mainly focuses on how the GPR data could be used in the future. However, I think there should be more focus on future applications of this methodology, including whether these models could assist in interpreting GPR data from other glacier surveys.

L376: "… one can choose a lower frequency antenna…", This conclusion is not supported by this study, as the 40 MHz data also did not show improvement regarding ice-bed returns.

**References**

Langhammer, L., Rabenstein, L., Schmid, L., Bauder, A., Grab, M., Schaer, P., and Maurer, H.: Glacier bed surveying with helicopter-borne dual-polarization ground-penetrating radar, J. Glaciol., 65, 123–135, https://doi.org/10.1017/jog.2018.99, 2019.

MacGregor, J. A., Studinger, M., Arnold, E., Leuschen, C. J., Rodríguez-Morales, F., and Paden, J. D.: Brief communication: An empirical relation between center frequency and measured thickness for radar sounding of temperate glaciers, The Cryosphere, 15, 2569–2574, https://doi.org/10.5194/tc-15-2569-2021, 2021.

Scanlan, K. M., Rutishauser, A., Young, D. A., and Blankenship, D. D.: Interferometric discrimination of cross-track bed clutter in ice-penetrating radar sounding data, Ann. Glaciol., 61, 68–73, https://doi.org/10.1017/aog.2020.20, 2020.

---

## Referee Comment (RC3)

**Review of "Ground penetrating radar on Rutor temperate glacier supported by ice-thickness modeling algorithms for bedrock detection"**

November 2024

**1 General**

The authors demonstrate a model-driven technique for picking points in radargrams corresponding to the glacier bed. They first ran three different models based on surface features, later used to guide the manual picking of Ground-Penetrating Radar (GPR) radargrams of 2012 and 2022. They then estimated the ice thickness in regions without GPR measurements by running the GlaTE model constrained by the GPR measurements.

The manuscript is well-written, and the subject of the work, the difficulty of retrieving the glacier bed, is a hot topic in glaciology, which deserves all the attention of the community. It is one of the most important sources of uncertainty for estimates of the future contribution to sea level rise. Dynamical glacier models are based on reconstructions of bed topography, which are themselves based on *in situ* measurements such as GPR and boreholes. The latter are reliable data, but they are not practical for surveying large areas. In this sense, GPR measurements are the foundation for glaciological studies. For this reason, the manuscript "Ground penetrating radar on Rutor temperate glacier supported by ice-thickness modeling algorithms for bedrock detection" from Vergnano et al. is very important.

However, it is important not to turn the logic around. Since GPR measurements are an important source of *in situ* information, reversing the process and leaking the modeling data into the GPR measurements can be delusive. This is the most important comment I have for this work, and I would like to see it discussed further in the manuscript.

**2 Major comments**

As mentioned previously, my main concern is related to the leakage of modeling data into measurement data. When inversion modeling is performed, it is crucial to have reliable data to constrain the model and evaluate its quality (see, for example, [Shahateet et al., 2023], where they show the impact of using different thickness maps for ice-discharge calculation and [Shahateet et al., 2024] where they show the importance of reliable thickness measurements). If the measurements are biased toward a specific model, it can highly impact everything that comes after, such as the inversion of the bed and the dynamical models that will use the inversion map.

The methodology is valuable, but the main point is to what extent you can use the picking drove by modeling estimations without data leakage. By analyzing Figure 5 and the appendix, I think you introduced too much bias. Some of the picks are not seen in the radargrams, only through the models.

I think that instead of having the model to then do the picks, the best approach would be to do several different picks and compare them to the models you have. In this case, you have less data leakage and more reliable measurements. In case where you have no reliable pick, leaving it without value is better than filling with model information, since in the future you do the inversion modeling of the ice thickness to cover all the domain. In this way, all the measurements you have are trustful and can be used broadly.

In this case, since you use models to support the picking of GPR measurements, you need independent data to validate your method. For this reason, it is desirable to use your method in another glacier with borehole measurements. In this way, you can have an independent validation method. I know it is easy to say and hard to do, but I think it is something important to keep in mind.

In chapter 5.1 (comparison of the three ice-thickness modeling algorithms), you stated that the GlaTE and GlabTop2 had similar results, proving the consistency between the different algorithms. First, you do not provide an overall analysis of their agreement, except by the total volume. To say that, you at least need to show an overall metric to conclude that. Second, it is no surprise that they agree well, since they use the same perfect plasticity method. In my opinion, their agreement is not a proof of the consistency of the method.

L287-288 is a warning that something may not be right. Why is the thickness overestimated near the outline of the glaciers? This is the region where you have reliable information from the GPRs, which shows that the measurements do not agree well with the models. Furthermore, L316 stated that 20% of the GPR data was used. Does it mean that 80% of the other points were taken from the models? In this case, it is no surprise that the total mass calculation of your method agrees well with the other models.

**3 Minor comments**

- The description of the homogenization of the different data sources is confusing and hard to follow with so many different years. Consider clarification and reduction of information.

- Why do you use the GlaTE as your final model? You never gave a complete reason for that. See my comment on L170-L175.

- In the Methods chapter, the figures are not presented in order. Furthermore, Figure 2 is not mentioned in the text except in the first enumeration of the Methods chapter. In the text, you mentioned Figure 1, and the next Figure to be mentioned is Figure 5. In general, I think it is important to improve the way you make references to the figures.

- Where do the other inputs from the OGGM model come from? You did not describe all the inputs.

- You don't need the enumeration from L245-253. This information is contained in the legend of the figures. Also, it is better to start talking about the figures before showing them out of the blue.

- You cite a personal communication twice. If you do not have a regular citation for that, rephrase it. For example, changing the word "considered" in L129 to "...showed to be..." avoids the need for a citation of a personal communication.

- For the OGGM model, you assumed that the glacier was in equilibrium to infer the ice volume flux ($q$), which according to the section of the study site is wrong. You can easily use a geodetic mass balance (the one you mentioned) to account for this mass change.

- Several times, you should change to "Rutor Glacier", with capital "G".

**4 Specific comments**

- L42: The acronym EM is not defined, and it is the first and only time that you use it. So, it is not needed.

- L52: Change "paragraph" to "section".

- L96: Change "multiisciplinary" to "multidisciplinary".

- L118: I think mentioning the v.sample tool in this overview of the methodology is not necessary and can distract from the main point.

- L123: The information in this line is not needed.

- L151: The sentence "according to the following steps," made me get lost. It looks like you are going to explain the steps, but you start to talk about the software. Only in the next page you are actually explaining the steps. Consider passing the sentence to the end of the paragraph: "The raw data were processed using the commercial... open source software (Huber and Hans, 2018), according to the following steps:".

- L170-L175: I think this paragraph is not necessary. It seems to me that you try to give a reason to use them because of their popularity. I would try to address this question with a more objective reasoning.

- L177: "Ice dynamics" instead of "ice flux mechanics".

- L183: Change "writers" to "authors".

- L188-L190: How do you avoid the glacier flow line computation? Furthermore, in L190 you say that $h_f$ is the mean ice thickness along the central glacier flow line. So do you actually not avoid it?

- L190: You say what is $f$, but no further explanation is given. What value did you use? It highly impacts the final result, since it accounts for lateral drag. I presume that in an alpine glacier, this value is important to discuss.

- L198-199: You can exclude this line and pass only "Further details are provided in the appendix of Frey et al. (2014)" and a good reference of the code (see my next comment).

- L199: several times you wrote the URL link as reference. I think it is not the right way of referencing a webpage.

- L202: "Clarke et al., 2013" should be "Clarke et al. (2013)" and "(Clarke et al., 2013)" is duplicated.

- L223: Same as L199.

- L228: "is" should be removed.

- L238: Same as L199.

- L239: The data is not well cited. It is (NASA JPL, 2020). Also, the reference is wrong in L465.

- L240: Better "(based on DEM differencing)".

- L280: Same as L239.

- 311: How the bias can not be considered significant? You said that the interpretation of GPR measurements below 50 m was difficult and only 20% of the GPR data was clearly identified (presumably in shallow regions, considering the previous statement). It means that in 80% of the time you used ice thickness from the models, or at least driven by it (it is not clear to me when it is driven and when you simply used the same thickness), especially in the regions where it accounts more to the total volume (deep regions). For me, this bias is the major concern regarding the methodology used, and need to be addressed in more details.

- L337: "from" is duplicated in "... from starting from..."

- L339: Change "miimizing" to "minimizing".

- L357-360: It is a conclusion.

- L365: 17.5 m "on average".

**4.1 Figures**

- Figure 1: In the legend, change "areas" to "categories". Furthermore, remove the parentheses from "(Crameri, 2021)".

- Figure 2: The legend is confusing. Why not numbered from 1 to 5? Also, it is better to number at the end also (e.g.: end-1).

- Figure 4: Same comment as in Figure 1 regarding "(Crameri, 2021)".

- Figure 5: The legends of GlaTE and OGGM are indistinguishable. It would be clearer if you used different colors for the different models.

- Figure 6: Same as in Figure 4.

- All the Appendix Figures: Same as in Figure 5.

- Is Figure A2 the same as Figure 5. If so, no need to show it again.

**References**

[Shahateet et al., 2024] Shahateet, K., Fürst, J. J., Navarro, F., Seehaus, T., Farinotti, D., and Braun, M. (2024). A reconstruction of the ice thickness of the antarctic peninsula ice sheet north of 70°s. EGUsphere, 2024:1–29.

[Shahateet et al., 2023] Shahateet, K., Navarro, F., Seehaus, T., Fürst, J. J., and Braun, M. (2023). Estimating ice discharge of the antarctic peninsula using different ice-thickness datasets. Annals of Glaciology.

---

## Author Comment (AC1)

Dear Editor and Reviewers,

Thank you for your helpful and in-depth revision of our manuscript. Below, you find our answers (in *italics*) to your raised issues.

Best regards
the Authors.

**Reviewer 1**

Review of the article entitled:

**Ground penetrating radar on Rutor temperate glacier supported by ice-thickness modelling algorithms for bedrock detection**

This study addresses the challenges of measuring ice thickness in temperate glaciers, such as the Rutor Glacier in the Southern Alps (Italy), in these glaciers the ice is at or near the melting point throughout its entire mass, including both the surface and deeper layers. This means the glacier contains both ice and meltwater, which makes it sensitive to temperature changes and contributes to faster melting. Meltwater then interferes with the clarity of Ground Penetrating Radar (GPR) signals. To improve the manual selection of bedrock profiles from GPR-based ice thickness measurements in such glaciers, the researchers combined GPR data with three open-source thickness inversion algorithms (GlabTop2, GlaTE, and OGGM), which estimate ice thickness based on surface topography and mass turnover. These models guided the manual selection of unclear or scattered GPR data for the Rutor Glacier. The study analyzed two new GPR datasets and produced a more accurate ice-thickness map using GlaTE (one of the algorithms, after selecting the correct bedrock profile with the aid of outputs from all three models). Authors then conclude that the glacier stored about 515 million cubic meters of ice in 2021, significantly higher than previous estimates. The authors claim that this methodology is replicable and can simplify future GPR surveys of temperate glaciers, particularly in noisy data conditions caused by meltwater.

Overall, the manuscript, methods, and results are well explained, however, I have several corrections to the current text. I find the study very creative and could have potential for the use of this type of data to validate and calibrate ice thickness inversion algorithms. However, my main concern lies with the novelty of the study and significant gaps in the methodology, such as the uncertainty quantification of model results and the use of OGGM in such a small-scale glacier-specific study, as well as not providing details of the set up used for the OGGM inversion. I could consider this study for publication, but only after the authors address my questions and make the necessary changes to the manuscript.

**Major comments:**
Novelty, Reproducibility, and Scalability

- After reading the manuscript, I find it difficult to see how this analysis effectively contributes to the broader challenge of providing ice thickness observations and distribution products that could be used to constrain, evaluate, or train model simulations, or new deep learning algorithms (e.g., The Instructed Glacier Model).
- Additionally, there is no discussion on how this method could be scaled to a regional level. Expanding the study across multiple temperate environments and numerous glaciers with GPR measurements (e.g., in Alaska, the Alps or South America) raises concerns about the efficiency of manually detecting multiple profiles. Such an expansion would likely require a more robust approach for parameter calibration, validation of ice thickness inversion by each algorithm, and the generation of a final thickness map using more than one algorithm.

I am concerned about the scalability of this method, as the parameters used for this specific glacier may not be transferable to others. This approach heavily depends on both data quality and the accuracy of model outputs. The examples presented in the manuscript show that the three algorithms perform well on some profiles but less so on others. A sensitivity analysis of the algorithms, with varied parameters to assess their impact on the thickness profiles, would have been valuable. Additionally, a more detailed uncertainty estimation for the final thickness product is needed. This could have been addressed by combining the results from all three algorithms, not just GlaTE, and providing a standard deviation on the final ice thickness map, while also comparing the results to existing ice thickness inversions and volume products (e.g., Millan et al. 2022, Farinotti et al. 2019 and Cook et al 2023 – all available in OGGM).

*Thank you for this valuable analysis.*
*Regarding scalability, it depends on the meaning of scalability at a regional level.*
*If this means using this methodology with the same set of parameters and similar uncertainty on a bulk of glaciers, we do not think it could work effectively. The main problems are mainly related to the great variability in the GPR profiles available on different glaciers. The data quality of the GPR profiles depends on many intrinsic factors of the temperate glacier; it happens that many times the signal is not very clear, as reviewed by Rutishauser et al., 2016. Moreover, the spatial resolution (lateral) is often constrained by the spatial distribution of GPR profiles along the glacier because of logistical or economical constraints. The manual processing and picking of GPR profiles is a need, which often requires local knowledge of the specific glacier and its recent history.*
*In Alpine regions, it could happen that the local environmental authorities could be more focused on some specific needs of specific glaciers, more than regional studies.*

*The scalability at a regional level of this methodology, therefore, it may surely be possible, but every glacier should be accounted for and analyzed manually and separately.*
*The strength of this methodology is to allow us to analyze a single glacier in a more effective way, and while scalability is certainly possible, it cannot be fully automated.*

*Regarding the sensitivity analysis, we fully agree with you. We will improve the study by including the analysis of a reasonable range in which the main parameters can change and make a sensitivity analysis.*

Methodology:
Related to the data input used:

- Regarding the GPR survey conducted by helicopter, I wonder if the authors need to correct for signal reflection from the nearby mountain terrain and elevation changes - i.e., interference caused by the surrounding mountain slopes in the radargram?
- Is the outline from the Randolph Glacier Inventory (RGI)? If authors have used their own glacier outline this might significantly deviate from its RGI counterpart, which could introduce errors and the authors should have computed the calibration steps again in OGGM.
- In the introduction (L69) authors are using their own DEM to predict the ice thickness from all models. In that case, the authors should have re-processed the GIS task of OGGM. A detail on how they use OGGM is missing (see below).
- All data inputs and as well as the model's thickness inversion (glacier initial state) represent different timespans. Why not use the same DEM and glacier outline across all models? You can input your own DEM and glacier outline into OGGM and recompute all steps until the inversion. See the following tutorials.
  - Using your own outline in OGGM
  - Create local topography maps from different DEM sources with OGGM
  - Step-by-Step guide to building preprocessed directories from scratch

*Thank you for your comments. Regarding the GPR survey by helicopter, we considered the local morphology and the helicopter altitude above ground, but we did not notice interference caused by surrounding mountain slopes. We think that this is because the glacier sits on the top and only a few slopes are higher than it, as you can see by this photo we took recently.*

[Figure]

*For the other comments, we used our own outline and DEM, provided by a recent geomatic survey (in 2021). For the OGGM model only, we used the RGI (Randolph glacier inventory) and another DEM provided by OGGM (and the results were corrected by the differences between the two DEMs). This choice was because we were curious about the impact of different DEMs on the outcomes; however, we should have tested OGGM with both DEMs. We can do it as part of our sensitivity analysis.*

*Thank you for the links to the tutorials :)*

Algorithms:

- Choice of input parameters in ice thickness inversion models: The authors should clarify that these parameters are not transferable between glaciers (see Zekollari et al. 2022). Additionally, a sensitivity study on the model parameters should have been conducted to assess the impact of parameter variation on ice thickness profiles computed by the models.
- How did the authors calibrate surface mass balance and ice thickness inversion in OGGM? or Did they use pre-processed directories? A specific workflow of the steps followed with OGGM is missing. The actual code repository of this study is not shared, thus is not possible to verify.

Results:

- It would be interesting to see a comparison of ice thickness differences between GlaTE and the other two models, along with a more in-depth discussion of the reasons behind these differences.
- The findings of the paper would be strengthened by comparing the resulting ice thickness map from GlaTE to existing ice thickness inversion and volume products (e.g., Millan et al. 2022, Farinotti et al. 2019, and Cook et al. 2023, all available in OGGM).

*Thank you, they are two good ideas that we can implement for sure.*

Discussion:

- I would encourage the authors to provide a stronger justification for how this methodology could be scaled to other glaciers and applied to existing GPR surveys in temperate glaciers. Additionally, it would be helpful to explain how this study could address the under-sampling problem of ice thickness in temperate regions (e.g., the Andes). However, caution is needed, as once models are used to improve observations, they are no longer pure observations and here there is a "human error" element also in place with this method. The authors could emphasize that GPR measurements provide a better representation than models, especially in areas like valley walls where models may struggle due to the simplification of glacier geometry (e.g., elevation flowlines and bed geometry assumptions in the case of OGGM).
- There is little mention about debris cover which is likely not accounted for in the ice thickness inversion algorithms.

*Thank you. I agree with the need to improve the discussion you suggest here. For the issue of scaling to other glaciers, we would formulate some critical evaluation, starting from the previous comments (see the Novelty, Reproducibility section).*

*We did not consider debris cover, which is not crucial in our glacier (see the photo above, taken at the end of the ablation season). In the GlaTE model there is the possibility to account for it with a simple parameter, but we are confident that this could be skipped in the sensitivity analysis.*

Conclusions:

- While the study is well-detailed and clearly explained, it could benefit from a stronger emphasis on its contribution to the broader challenge of ice thickness observations in temperate glaciers. The results, though valuable for this specific glacier, do not provide new insights beyond the updated GPR surveys and improved ice thickness map. To enhance the overall impact, the authors could explore a more quantitative

interpretation of the results and better highlight how their findings address larger-scale issues in future research.

*Thank you. A more quantitative approach will be our effort during this revision. For the larger scale issue, refer to our previous discussion. This methodology does not have a straightforward scalability in terms of automatic processing of many glaciers, but can surely be applied to other glaciers one by one. We could highlight here the main weaknesses and strengths of the methodology in this sense.*

Minor corrections and suggestions:

*Thank you for all your very detailed corrections and suggestions.*
*We do not reply one by one, but we will try to include everything in the manuscript, since we agree with each of them. In the event we fail to include some suggestions in the next revision, we will highlight them in the next revision comments.*

**Abstract**
L1: Add an example of where temperate glaciers are located (i.e. not at the poles).

L10: Authors should explicitly state that they used model output to manually select the best bedrock profile from the GPR data in problematic survey sections, clarifying that the model output is used to fill gaps in the GPR observations along those profiles. This should be stated early on in the abstract and the introduction, to improve the objective of the manuscript.

**Introduction**
L29-30; replace "inner composition." with "present day ice thickness distribution and geometry"

L42; EM to Electro Magnetic.

L43; Suggestion: Rutishauser et al. (2016) analyzed a large set of GPR data acquire on Swiss glaciers and found that depending on the specific glacier, the bedrock interface could only be successfully detected in 12-69% of the GPR data due to this scattering issue.

L48 Remove "Also". Suggestion: "Air bubbles trapped in ice cause additional scattering of the GPR
signal, which helps differentiate between various types of ice…"
L51: replace "are reported in the Study site paragraph" by "are

summarised in section X" L56: replace "paper" by "study".

L58: point to a figure to direct the reader to a GPR profile to indicate the issue.

L62: replace "help the analysist" with "aid".

L64: I will just call it glacier models or ice dynamical models (they all are ice thickness inversion algorithms of some sort). Authors should pick a single definition throughout.

L67: Please cite the version of OGGM used in the study. See https://docs.oggm.org/en/latest/citing- oggm.html#software-doi

L77: statements like "it should be" introduces doubt on the results, try to avoid this type of language and quantify how much the ice thickness product improved via statistics.

**Study site**
This section is too long and I don't see how past geomorphological events are relevant to this particular study. I would start by describing the site (from L97) and georeferenced so the readers know where the glacier is geographically.

L102: Add citation of DEM's used to compute ice thickness losses.

**Methods**
L111-123: Remove all

"to". L113: Remove "and

updated".

L119: Replace "to perform the manual picking" with "Manually select reflexion events"

L123: Replace "to draw a final result…" with Produce a map of the glacier ice thickness (Figure 6)

L124-127: Suggestion:

"Some topographical adjustments were necessary to assist in analyzing GPR observations that span different time periods (2012 and 2022). A 2021 DEM of the glacier surface was used for the GlaTE and GlabTop2 algorithms, while the 2000 DEM was used for the OGGM algorithm. In other words, the GlaTE and GlabTop2 models represent the 2021 situation, OGGM represents 2000, and the GPR data corresponds to 2012."

What about the glacier outline date?

L131: Why do the authors not use the same DEM (or the best DEM) for all models? See above.

L124-145: This text seems a bit misplaced, I would divide the text into sections for (i) pre-processing of input data for models and (ii) post processing of model output and the describe (ii) after describing the algorithms.

Sect 3.1 Explain if the GPR data collected from a helicopter needed to be corrected for altitude changes in the survey and the scattering effects caused

by the nearby terrain. See Church, G. et al. (2018).

L154: Add - The GPR data was processed by the following method:

L170-175: These lines contain irrelevant text. The increase in usage and citations of a tool or model (e.g., OGGM) does not necessarily indicate it is the best tool for a particular study. A more thorough justification for the choice of tools should be provided here. The OGGM documentation clearly states that it is designed for large-scale or regional glacier modelling. Caution is advised when using OGGM for single glacier studies, and a detailed workflow for producing the thickness inversion should be included in such cases.

L177: Remove "ice flux mechanics" and replace: ice flow theory and

mass conservation.

L179: Replace all "picking" with manual selection.

Suggestion replace L181 – L184 with

"The thickness inversion models required specific input parameters to run. These were reviewed for consistency with the physical characteristics of the study area, but unless stated otherwise, default values from similar alpine glacier studies by the algorithm developers were used. See below a summary of all models".

Here, authors should state that these parameters are not transferable glacier to glacier and a sensitivity study on one profile at least should have been carried out on model parameters to see the impact of parameter uncertainty in ice thickness distribution.

Sect. 3.2
This section would benefit from a table comparing the parameters (and their values) used in all models, providing a quick overview of each model's setup, along with a column citing the publications from which these parameters were sourced.

L225: Cite OGGM version used.

L227-228: Replace "OGGM bed topography inversion algorithm" with "OGGM ice thickness inversion algorithm" … "which is based in ice flow dynamics and mass conservation (Farinotti, et al. 2009 and Maussion et. al 2019). The ice flux is computed as: … "

L234: "under the simple assumption of equilibrium". This is not correct in the case that the latest version 1.6.0 of OGGM was used. Please, note the version used in this study and how the ice thickness inversion was calculated. Do authors calibrate surface mass balance and ice dynamical parameters? Note that it is possible to calibrate OGGM to match geodetic mass balance data which removes the equilibrium assumption. In the latest version is possible to calibrate the glacier mass balance and ice dynamics parameters at the same time using a "dynamic spin-up" see Appendix A and Aguayo et al. (2023) for

details and the following tutorials:

**Discussion**
L276: "ice thickness"? do you mean ice volume (why is this not just stated in Km$^3$)

Section 5.1 Here ideally authors should have done a better analysis on the difference between the thickness maps computed by the different models and also show a flowline profile view. Also compare the resultant volume with previous studies and estimates (see references).

A lot of this section could be removed if the authors use the same DEM and there is no need to correct ice thickness changes over time.

L302-307: Suggestion

"This joint interpretation prevented the mistake of interpreting the first non-reflective layer (white in the GPR sections) as ice and the first reflective zone (scattered black) as bedrock. The deepening reflection on the right side of Figure 3 clearly shows that the ice-bedrock interface is not related to the scattered reflective zone observed at 20-40 m depth. Manually picking the ice-bedrock interface, guided by the estimates from the algorithms, was particularly helpful, especially below 50 m where the GPR signal was too attenuated."

L309 "This is not far from estimates without GPR data" Quantify such differences.

L329. "previous research" add citations.

**(more comments below)**

**Figures**

**Figure 1.**

This figure needs a map of the alps with the location of Rutor glacier. Add RGIID or GLIMS ID.

Replace "how many meters it has subsided in the past decade (from 2008 to 2021)" with changes in
ice thickness (m) from 2008 to 2021.
Add RGI outline as well as the outlines used in this study with different colours.
Add citations.

**Figure 2.**
Dotted survey lines could be thicker.

**Figure 3.**
I would add a point of first guess from authors of where the bedrock might be if they didn't know from

the thickness inversion algorithms.

**Figure 4.**
Another panel could be added to this figure looking into thickness profiles from models along the main flowline and two more figures showing the ice thickness differences between GlaTe and the other two models.

**Figure 5. (and similar)**
Add to the bottom panel the part of the profile that is taken or selected using the ice thickness inversion algorithm (i.e., fill the gap in the profile via another colour)

**Figure 6.**
Instead of displaying the GPR data on top of the thickness map, display thickness differences between GlaTE and the GPR. Or plot differences in profiles.

**Appendix.**
Authors should also compare their resultant thickness map with other estimates. See comments above, this could go in the appendix.

**References**
Aguayo, R., Maussion, F., Schuster, L., Schaefer, M., Caro, A., Schmitt, P., Mackay, J., Ultee, L., Leon- Muñoz, J., and Aguayo, M.: Assessing the glacier projection uncertainties in the Patagonian Andes (40–56° S) from a catchment perspective, EGUsphere [preprint], https://doi.org/10.5194/egusphere- 2023-2325, 2023.

Church, G. & Bauder, A. & Grab, Melchior & Hellmann, Sebastian & Maurer, H.. (2018). High- resolution helicopter-borne ground penetrating radar survey to determine glacier base topography and the outlook of a proglacial lake. 1-4. 10.1109/ICGPR.2018.8441598.

Cook, S. J., Jouvet, G., Millan, R., Rabatel, A., Zekollari, H., & Dussaillant, I. (2023). Committed ice loss in the European Alps until 2050 using a deep-learning-aided 3D ice-flow model with data assimilation. Geophysical Research Letters, 50, e2023GL105029. https://doi.org/10.1029/2023GL105029

Millan, R., Mouginot, J., Rabatel, A. et al. Ice velocity and thickness of the world's glaciers. Nat. Geosci. 15, 124–129 (2022). https://doi.org/10.1038/s41561-021-00885-z

Farinotti, D., Huss, M., Fürst, J.J. et al. A consensus estimate for the ice thickness distribution of all glaciers on Earth. Nat. Geosci. 12, 168–173 (2019). https://doi.org/10.1038/s41561-019-0300-3

Zekollari, H., Huss, M., Farinotti, D.,& Lhermitte, S. (2022). Ice-dynamicalglacier evolution modeling—Areview. Reviews of Geophysics,60, e2021RG000754. https://doi.org/10.1029/2021RG000754.

**Reviewer 2**

Review of "Ground penetrating radar on Rutor temperate glacier supported by ice-thickness modeling algorithms for bedrock detection" by Andrea Vergnano et al. (2024)

The manuscript presents airborne and ground-based GPR data collected in 2012 and 2022 over Rutor Glacier, a temperate glacier, which are known for challenges posed by high signal scattering and absorption. The study's novel approach combines three models (GlabTop2, GlaTE, OGGM) to help with the identification of the ice-bed interface, improving upon prior estimates that likely underestimated ice thickness due to misinterpreted scattering zones near the surface. The study concludes that incorporating the models improves the GPR interpretation in terms of ice thickness. Finally, a new ice thickness map is generated with the new GPR interpretations constraining the GlaTE model.

I think this study presents a creative approach to improve the interpretation of challenging GPR data over temperate glaciers. Overall, the paper fits the scope of the journal and has potential, but in my opinion, several major issues need to be addressed before publication. These include the need for clearer methodological explanations, particularly concerning the use of DEMs in the models. The introduction should more clearly highlight the true novelty of using models to improve GPR interpretation. Furthermore, a deeper analysis of model-assisted picking, including statistical comparisons between model-guided and

unguided picks, is necessary to fully support the claim that the models "provided substantial help in manually picking the ice- bed interface". Finally, the manuscript requires substantial English language revision to improve clarity, as many sentences are awkwardly phrased or repetitive. I hope the authors find my comments useful and that they can help to improve the manuscript.

**Major Issues**

- **Language:** The manuscript would benefit from significant English editing. Many phrases are unclear or awkward, and the text could be more concise. Paragraphs often repeat information unnecessarily. I have made specific suggestions in the line-by-line comments.

*Thank you for this comment. We will accept with thanks your revision about language and those of other reviewers, who also highlighted some unclear phrases.*

- **Research focus**: The main purpose of this research as stated in the introduction is to "investigate the Rutor glacier thickness with two new GPR datasets" (L56). However, I believe that the manuscript could better highlight the key goal/innovation – using models to assist in identifying the glacier bed in GPR data. This is underemphasized in the introduction, results, and discussion sections.

*Thank you. We will emphasize more the key goal and innovation, also taking into account the lack of discussion about regional scaling of this methodology, highlighted by Reviewer 1.*

- **Abstract**: I find the abstract quite lengthy, and the primary goal and key findings are not clearly conveyed. I recommend revising the abstract after the manuscript has been edited to ensure the message is concise and focused on the main points.

*Thank you! We will do it as the last thing after all the revisions.*

- **Methods:**
    - **Ice thickness change:** It is unclear whether the ice thickness change (Figure 1) from the DEM differencing is original to this study or based on previous work. If new, the method should be explained

*This is original of this work. It was a standard procedure, therefore, we did not emphasize it, but we could explain it in detail.*

- **DEM use:** The rationale for using different DEMs for different models is unclear, especially why a 2000 DEM was used for OGM. I am not familiar with the models, but is it not possible to run the OGGM model with the 2021 surface topography? Additionally, why was a 2008 and 2021 DEM used for the GlaTE and GlabTop2?

*Thank you, this is correct, it is possible to run the OGGM with the 2021 surface topography, and we will do it in the revision. Also Reviewer 1 is of the same opinion. We will also assess eventual differences in the outcomes due to the use of different DEMs, which can be interesting in this study.*

*The 2008 and 2021 DEMs were used because some of the GPR data were collected in 2012 and some in 2022, and we want to provide a picture of the ice thickness in 2021. Therefore, the 2012 data have to be corrected for the loss of ice between 2012 and 2021. To do so, we calculated an average annual ice loss by subtracting the 2021 and the 2008 DEMs. Maybe this process was not clear as explained in the methods sections, we think about how to clarify it.*

- **Ice thickness vs bedrock topography:** I understand that the models output ice thicknesses, but why not compute a bedrock DEM instead? The bedrock topography is not expected to change over the study period, and could directly be compared to the GPR data from any survey time (i.e. 2012 and 2022). Ice thicknesses can still be extracted (subtracting the bed DEM from the surface DEM). This could reduce all the ice thickness corrections that currently need to be applied.

*This comment sounds particularly interesting, indeed, one nice idea for future works is to test these models on a glacier in which we have the DEM in several years to check the robustness in the detection of the bedrock. It would be interesting to build a model that takes into account multiple surface DEMs collected in multiple years, and is forced to calculate always the same bedrock. It could be done on this Rutor glacier using the 2000, 2008 and 2021 DEMs, but this would require an important modification of the code. We believe it is out of the scope of this specific paper, as it may take time to assess it in a rigorous way and change the model code.*
*Regarding the GPR surveys, we do not think that the 2012 and 2022 surveys can be compared effectively in this sense, because they overlap only in the top part of the glacier, which remained almost unchanged in the last decades, since it is the coldest part.*
*I would report these points in the discussion section, prompting future research.*

- **Results:**
    - **Ice loss map:** As the ice loss map supports the hypothesis of underestimated thickness, it should be included in the results.

*I understand your point of view, We put it before because it seemed to me a good reading flow. We will reorganize the paper according to your comment.*

    - **Statistical analysis:** A more in-depth quantitative analysis is needed to assess how much the models aid in picking the ice-bed interface. This could include comparisons of ice thickness picks with and without the models, as well as how each individual model was used (e.g. for future recommendation, is there one model that stands out, instead of having to run all three?) The discussion includes some statistics (e.g. "20% of the GPR lines clearly identified the bedrock"), but it is unclear how these were calculated, and they are not included in the results.

*Thank you for these interesting ideas on how to present the results more quantitatively. Reviewer 1 asked for a sensitivity analysis, for example. So we see the need to improve in this sense, and we find your suggestions very helpful, and we will discuss them all. Thanks.*

    - **Radargram interpretation:** The manuscript could more strongly emphasize how weak reflectors, identified with low confidence, are validated through model agreement, increasing confidence in identifying the ice-bed interface.

        That being said, I also think there are instances where the selected reflectors appear questionable, which may raise concerns about potential bias in the manual picking process when influenced by model outputs (e.g. picking noise). For example, I have difficulties identifying a reflector that was picked on
        - Profile 2012-7 between ~200-500 m
        - Profile 2012-8 between ~1500-1900 m
        - Profile 2012-9 between 500-1000 m
        - Profile 2012-10 between 300-1500 m

        This risk should be discussed explicitly (in the discussion section), as it is important to acknowledge the possibility of seeing patterns in noise when guided by models.

*You are perfectly right, we want to discuss the subjectivity of the methodology*

*in a stronger way.*

- **Figures**

*For this comment section and the minor comments, I reply just here, saying that I appreciate your revision and I will do my best to include all of your comments in the manuscript. Thank you again!*

- A study area overview map to see where in the Alps Rutor glacier is would be useful (e.g. integrated in Figure 1 or 2)
- Consider increasing the font size in all figures and remove color scale name information in the figure caption.
- *Figure 1*: Add elevation contour lines (or on Figure 2) and a reference to the source of the glacier outline. Also consider labelling the glacier tongues as described in the text.
- *Figure 2*: Increase line width, and consider using markers instead of "start" and "end" labels to reduce text and improve readability.
- *Figure 3*: I suggest adding arrows to indicate the "clutter zone" and "true bedrock" so the reader can follow what is meant in the text (L256-259). Also, consider removing Figure 3 as it is repeated in Figure 5, or replace it with another example (e.g. Profile 2012-8).
- *Figure 5*: I suggest using different colors instead of line-styles to better distinguish the models.
- *Figure 6*: I suggest using the same colormap for the GPR and model ice thickness for easier comparison. The GPR data can be surrounded with a white outline for contrast.
- *Appendix Figures:* I think that some of the description should be moved into the main results/discussion sections.

**Minor Issues/Line-by-line comments**

L3: I suggest removing the sentence with cold ice, it is

irrelevant here. L8-9: I suggest removing the sentence

"Besides, GPR…."

L31, L36, etc.: Consider replacing "meltwater" with "englacial water content" or "water", to avoid confusion with surface meltwater generation/runoff, englacial water may also result from rain.

L32: I believe it is "pressure-melting point", not "temperature-pressure melting point".

L35-36: This also reads a bit awkward, e.g. we wouldn't expect a sudden change in geothermal heat flux. I suggest rewording to "Temperate glaciers at the pressure melting point are primed for rapid meltwater production upon

small energy or heat inputs…"

L37: Specify that while high-quality GPR surveys are possible (e.g. for snow/firn near surface studies),challenges lie in detecting the bed returns. Reword to "…can challenge the interpretation of bedrock returns from Ground Penetrating Radar (GPR) surveys."

L40: Clarify "smaller-scale heterogeneities", e.g. small fractures or sediment grains, smaller than the wavelength (or quarter wavelengths/range resolution)?

L42-44: Reword to clarify what was studied, e.g. "Challenges in detecting basal returns over temperate glaciers have been studied …" Additionally, I think it would be good to mention the studies on effects of antenna orientation on detection of the bedrock reflection e.g. (Langhammer et al., 2019).

L47-48: Rephrase to clarify that englacial debris may also originate from surface material, not just freeze-on at the bed.

L49-52: I think this sentence could benefit from directly referencing some of these studies. Also consider integrating the study site description here.

L52-55: Replace "resolution" with "spatial resolution". I suggest reformulating to "The spatial coverage of GPR surveys is limited by survey speed, time and access (e.g. crevasses), leading to discrete, limited sampling of the glacier bed. It is therefore possible that the maximum ice thickness remains unknown due to limited survey coverage."

L57-L59: Re-word for clarity, e.g. "These new datasets reveal high scattering of the radar signal over most parts of the glacier, demonstrating the difficulty in detecting the ice-bedrock interface."

L60: Include a reference for the "previous doubtful estimates of ice thickness".

L68: I believe the correct reference is (Langhammer et al, 2019a), verify other instances.

L68-70: "Thanks to … are extracted." I suggest rewording to "The ice thickness is predicted using the three models." (i.e. the DEM part belongs in the methods section).

L71-72: Rephrase to "… superimposed on the radargram to help identify the most likely ice- bedrock interface…"

L74: Replace "inner geometry of the glacier" with "bedrock topography"

L93-96: Instead of just mentioning multidisciplinary aspects/different perspectives, provide examples (e.g. glaciology, geomorphology, ecology, hydrology …?).

L96: misspelling of "multidisciplinary"

L97: reword to "… the Rutor glacier covers an area of 7.5 km$^2$ …" L100 and others: Replace "outline" with "margin"

L101-108: Moving the ice thickness change discussion to the methods/results sections, or reference to original source if from another study. L107: replace "extension" with "area".

L108-109: Move this sentence to the introduction for better context.

L116: Reword to "The results of this step are show in Figure 4.", or remove this sentence. L119: Replace "reflection events" to "reflectors"

L120: Replace "limit…" with "reduce the chance of mis-interpreted bedrock reflections" L121: Be more specific: "… surface topography and the GPR-derived bedrock topography."

L123: Step 6 does not contribute to the "overcome the difficulties in interpreting the GPR data…" as stated at the beginning of the methods section. I suggest removing this step.

L124-145: Address comments above and consider moving this section to 3.2. Clarify the glacier outline source (e.g. mentioned in L186)?

L150-151: This is repeated in Step 4, I suggest removing it here.

L154: What was the bandpass filter of for the ground-based survey? I assume it was lower than this.

L156: Clarify "correct max phase", e.g. is it a dewowing process? Also, avoid non-scientific language like "suggested by Reflexw".

L178: Replace "drive…" with "help identify the ice-bedrock interface during manual picking…"

L181-182: Reword to: "The modeling algorithms required additional input parameters (e.g.

xxx). These were checked for consistency with the Rutor glacier study area, …" L202: Remove double citation.

L207: Clarify that known ice thickness/bedrock points, not GPR data itself, are used as input. Similarly, further down, I assume h$_{GPR}$ is the GPR-derived ice thickness, not the GPR data.

L212: I suggest removing "outside"

L213: Clarify "gradient of outside terrain slope", i.e. is it the slope outside the glacier? L226: I believe this should be "meltwater runoff"

L228: remove the "is" before "equation" L233: precipitations

(remove s)

L232-235: If a mass balance was used to estimate q, include the details on how this was determined for Rutor glacier and the value used.

L245-253: Instead of listing the figures at the start of the results section, I suggest integrating them into the text to improve the flow of the text.
L257: Replace "black reflection zone" with "strong backscatter zone" or "high amplitude zone".

L257-258: "However, on the right side of the plot, the clearly submerging ice-bedrock interface shows…" I suggest rewording the interpretation of the submerging ice-bedrcok interface to make it less definitive and more interpretative (e.g. the contrast dipping towards the center on to the left also looks like a bed return, but is not picked as such).

L260-263: Move the comparison with other studies to the discussion section. Also, the Villa et al. (2008) study used GPR data from 2006, not 2008.
L270-274: There is a lot of repetition of methods within this section. I suggest focusing on results here only.

L277-282: This section is mostly a repetition of the methods part. Move any methods to the methods section and focus the discussion on e.g. how resolution affects the result (e.g. over- deepening being an effect of fine-resolution DEMs?)
L288: Explain how the ice thickness near the glacier margin was overestimated, e.g. was it compared to the GPR data?

L291: Replace "readability" with "…degree of visibility" or "strength of the ice-bedrock return."
L298: "… more confidence was given…", it is not clear how this was implemented. E.g., do the picks come with a confidence level?

L303-305: I suggest including a discussion on the possibility of off-nadir returns (e.g. from valley side walls).

L326: What about seismic surveys?

L329: I suggest adding this citation here (MacGregor et al., 2021) (relation between frequency and ice thickness)

L337: Can we quantify "reasonably comparable models" in the results section, e.g. what is the mean, maximum, standard deviation in the differences in ice thickness predictions?
L339: misspelling of minimizing

L340: It is unclear where these uncertainty estimates come from

L343-L360: This section mainly focuses on how the GPR data could be used in the future. However, I think there should be more focus on future

applications of this methodology, including whether these models could assist in interpreting GPR data from other glacier surveys.

L376: "… one can choose a lower frequency antenna…", This conclusion is not supported by this study, as the 40 MHz data also did not show improvement regarding ice-bed returns.

**References**

Langhammer, L., Rabenstein, L., Schmid, L., Bauder, A., Grab, M., Schaer, P., and Maurer, H.: Glacier bed surveying with helicopter-borne dual-polarization ground-penetrating radar, J. Glaciol., 65, 123–135, https://doi.org/10.1017/jog.2018.99, 2019.

MacGregor, J. A., Studinger, M., Arnold, E., Leuschen, C. J., Rodríguez-Morales, F., and Paden, J. D.: Brief communication: An empirical relation between center frequency and measured thickness for radar sounding of temperate glaciers, The Cryosphere, 15, 2569–2574, https://doi.org/10.5194/tc-15-2569-2021, 2021.

Scanlan, K. M., Rutishauser, A., Young, D. A., and Blankenship, D. D.: Interferometric discrimination of cross-track bed clutter in ice-penetrating radar sounding data, Ann. Glaciol., 61, 68–73, https://doi.org/10.1017/aog.2020.20, 2020.

**Reviewer 3**

Review of "Ground penetrating radar on Rutor temperateglacier supported by ice-thickness modeling algorithms forbedrock detection"

November 2024

- **General**

The authors demonstrate a model-driven technique for picking points in radargrams corresponding to the glacier bed. They first ran three different models based on surface features, later used to guide the manual picking of Ground-Penetrating Radar (GPR) radargrams of 2012 and 2022. They then estimated the ice thickness in regions without GPR measurements by running the GlaTE model constrained by the GPR measurements.

The manuscript is well-written, and the subject of the work, the

difficulty of retrieving the glacier bed, is a hot topic in glaciology, which deserves all the attention of the community. It is one of the most important sources of uncertainty for estimates of the future contribution to sea level rise. Dynamical glacier models are based on reconstructions of bed topography, which are themselves based on *in situ* measurements such as GPR and boreholes. The latter are reliable data, but they are not practical for surveying large areas. In this sense, GPR measurements are the foundation for glaciological studies. For this reason, the manuscript "Ground penetrating radar on Rutor temperate glacier supported by ice-thickness modeling algorithms for bedrock detection" from Vergnano et al. is very important.

However, it is important not to turn the logic around. Since GPR measurements are an important source of *in situ* information, reversing the process and leaking the modeling data into the GPR measurements can be delusive. This is the most important comment I have for this work, and I would like to see it discussed further in the manuscript.

- **Major comments**

As mentioned previously, my main concern is related to the leakage of modeling data into measurement data. When inversion modeling is performed, it is crucial to have reliable data to constrain the model and evaluate its quality (see, for example, [Shahateet et al., 2023], where they show the impact of using different thickness maps for ice-discharge calculation and [Shahateet et al., 2024] where they show the importance of reliable thickness measurements). If the measurements are biased toward a specific model, it can highly impact everything that comes after, such as the inversion of the bed and the dynamical models that will use the inversion map.

The methodology is valuable, but the main point is to what extent you can use the picking drove by modeling estimations without data leakage. By analyzing Figure 5 and the appendix, I think you introduced too much bias. Some of the picks are not seen in the radargrams, only through the models.

I think that instead of having the model to then do the picks, the best approach would be to do several different picks and compare them to the models you have. In this case, you have less data leakage and more reliable measurements. In case where you have no reliable pick, leaving it without value is better than filling with model information, since in the future you do the inversion modeling of the ice thickness to cover all the domain. In this way, all the measurements you have are trustful and can be used broadly.

*Thank you for your insight. We think this approach that you suggest, about making several different picks and comparing them to the models, can be done in our manuscript, and presented accordingly. Also yes, we could leave without value those with no reliable pick: Reviewer 2 raised a similar issue.*

In this case, since you use models to support the picking of GPR measurements, you need independent data to validate your method. For this reason, it is desirable to use your method in another glacier with borehole measurements. In this way, you can have an independent validation method. I know it is easy to say and hard to do, but I think it is something important to keep in mind.

*We would like to have a borehole on this glacier, but it is hard to do! We can discuss it better in the discussion section as a limitation of the study. Unfortunately, the goal of the paper is mainly related to this glacier and the effort of analyzing another glacier during the revision process, with little local knowledge about it, is very hard. We could think of it for further steps of research, where the ground-truth (ice-truth…) calibration is available.*

In chapter 5.1 (comparison of the three ice-thickness modeling algorithms), you stated that the GlaTE and GlabTop2 had similar results, proving the consistency between the different algorithms. First, you do not provide an overall analysis of their agreement, except by the total volume. To say that, you at least need to show an overall metric to conclude that. Second, it is no surprise that they agree well, since they use the same perfect plasticity method. In my opinion, their agreement is not a proof of the consistency of the method.

*Thank you. I understand what you mean, and probably this requires to include also OGGM with the same DEM as a comparison. We will also add your statement about the perfect plasticity method in the discussion section.*

L287-288 is a warning that something may not be right. Why is the thickness overestimated near the outline of the glaciers? This is the region where you have reliable information from the GPRs, which shows that the measurements do not agree well with the models. Furthermore, L316 stated that 20% of the GPR data was used. Does it mean that 80% of the other points were taken from the models? In this case, it is no surprise that the total mass calculation of your method agrees well with the other models.

*Thank you. We generally accepted that near the outline it is more difficult for the models to retrieve the correct thickness, as it is expected for them to provide a general shape of the glacier and not be very accurate in distinguishing if ice is thick 5 or 10 meters at a certain point near the outline. Anyway, We have to investigate this issue further during the revision process, playing with the model parameters.*

- **Minor comments**

*Thank you for all these minor comments and specific comments. We do not*

*reply one by one because we generally agree with all of them. If we fail to take some of them into account, we will highlight them in the revision comments.*

- The description of the homogenization of the different data sources is confusing and hard to follow with so many different years. Consider clarification and reduction of information.

- Why do you use the GlaTE as your final model? You never gave a complete reason for that. See my comment on L170-L175.

- In the Methods chapter, the figures are not presented in order. Furthermore, Figure 2 is not mentioned in the text except in the first enumeration of the Methods chapter. In the text, you mentioned Figure 1, and the next Figure to be mentioned is Figure 5. In general, I think it is important to improve the way you make references to the figures.

- Where do the other inputs from the OGGM model come from? You did not describe all the inputs.

- You don't need the enumeration from L245-253. This information is contained in the legend of the figures. Also, it is better to start talking about the figures before showing them out of the blue.

- You cite a personal communication twice. If you do not have a regular citation for that, rephrase it. For example, changing the word "considered" in L129 to "...showed to be..." avoids the need for a citation of a personal communication.

- For the OGGM model, you assumed that the glacier was in equilibrium to infer the ice volume flux ($q$), which according to the section of the study site is wrong. You can easily use a geodetic mass balance (the one you mentioned) to account for this mass change.

- Several times, you should change to "Rutor Glacier", with capital "G".

- **Specific comments**

  - L42: The acronym EM is not defined, and it is the first and only time that you use it. So, it is not needed.

  - L52: Change "paragraph" to "section".

  - L96: Change "multiisciplinary" to "multidisciplinary".

  - L118: I think mentioning the v.sample tool in this overview of the methodology is not necessary and can distract from the main point.

  - L123: The information in this line is not needed.

  - L151: The sentence "according to the following steps," made me get lost. It looks like you are going to explain the steps, but you start to talk about the software. Only in the next page you are actually explaining the steps. Consider passing the sentence to the end of the paragraph: "The raw data were processed using the commercial. . . open source software (Huber and Hans, 2018), according to the following steps:".

  - L170-L175: I think this paragraph is not necessary. It seems to me that you try to give a reason to use them because of their popularity.

I would try to address this question with a more objective reasoning.

- L177: "Ice dynamics" instead of "ice flux mechanics".

- L183: Change "writers" to "authors".

- L188-L190: How do you avoid the glacier flow line computation? Furthermore, in L190 you say that $h_f$ is the mean ice thickness along the central glacier flow line. So do you actually not avoid it?

- L190: You say what is $f$ , but no further explanation is given. What value did you use? It highly impacts the final result, since it accounts for lateral drag. I presume that in an alpine glacier, this value is important to discuss.

- L198-199: You can exclude this line and pass only "Further details are provided in the appendix of Frey et al. (2014)" and a good reference of the code (see my next comment).

- L199: several times you wrote the URL link as reference. I think it is not the right way of referencing a webpage.

- L202: "Clarke et al., 2013" should be "Clarke et al. (2013)" and "(Clarke et al., 2013)" is duplicated.

- L223: Same as L199.

- L228: "is" should be removed.

- L238: Same as L199.

- L239: The data is not well cited. It is (NASA JPL, 2020). Also, the reference is wrong in L465.

- L240: Better "(based on DEM differencing)".

- L280: Same as L239.

- 311: How the bias can not be considered significant? You said that the interpretation of GPR measurements below 50 m was difficult and only 20% of the GPR data was clearly identified (presumably in shallow regions, considering the previous statement). It means that in 80% of the time you used ice thickness from the models, or at least driven by it (it is not clear to me when it is driven and when you simply used the same thickness), especially in the regions
where it accounts more to the total volume (deep regions). For me, this bias is the major concern regarding the methodology used, and need to be addressed in more details.

- L337: "from" is duplicated in ". . . from starting from. . . "

- L339: Change "miimizing" to "minimizing".

- L357-360: It is a conclusion.

- L365: 17.5 m "on average".

- **Figures**
  - Figure 1: In the legend, change "areas" to "categories". Furthermore, remove the parentheses from "(Crameri, 2021)".
  - Figure 2: The legend is confusing. Why not numbered from 1 to 5? Also, it is better to number at the end also (e.g.: end-1).
  - Figure 4: Same comment as in Figure 1 regarding "(Crameri, 2021)".
  - Figure 5: The legends of GlaTE and OGGM are indistinguishable. It would be clearer if you used different colors for the different models.
  - Figure 6: Same as in Figure 4.
  - All the Appendix Figures: Same as in Figure 5.
  - Is Figure A2 the same as Figure 5. If so, no need to show it again.

**References**

[Shahateet et al., 2024] Shahateet, K., Fürst, J. J., Navarro, F., Seehaus, T., Farinotti, D., and Braun, M. (2024). A reconstruction of the ice thickness of the antarctic peninsula ice sheet north of 70°s. EGUsphere, 2024:1–29.

[Shahateet et al., 2023] Shahateet, K., Navarro, F., Seehaus, T., Fürst, J. J., and Braun, M. (2023). Estimating ice discharge of the antarctic peninsula using different ice-thickness datasets. Annals of Glaciology.

---

## Referee Report (RR1)

**Re-review of "Integrating GPR and ice-thickness models for improved bedrock detection: the case study of Rutor temperate glacier" by Andrea Vergnano et al. (2025)**

I am very excited to see this revised version, and commend the authors for the considerable effort they have invested in revising this manuscript. The new manuscript addresses almost all of my earlier concerns, and it is now written much more clear and well structured. Specifically, the research goals, hypotheses and scientific contribution are now stated clearly throughout the manuscript, the manuscript reads well, and the updated figures are much easier to read. I also really appreciate the inclusion of the comprehensive supplementary material, making these codes and the processing flow available is a valuable service to the community.

Lastly, I really appreciate the addition of pick classifications ("sure", "guided by models", "wrong"), as it reveals how manual and model-assisted interpretations interact. I have one suggestion that could further highlight the impact of the model inclusion when interpreting GPR data: I suggest adding a brief quantitative summary that exploits the pick categories, e.g. reporting the percentages of "wrong", "guided" and "sure picks" relative to the total GPR data. These could simply be mentioned in brackets throughout the text that is already there. I believe that the necessary data for this analysis now exists, and this should be straightforward.

Apart from this minor suggestion and a few trivial language suggestions below, I believe the manuscript is ready for publication and recommend acceptance after these have been addressed.

**Line-by-line comments**

L9-11: I suggest rewording to: "Combined visualization of the GPR and model data in 2D and 3D…, especially where the GPR data contained scattering noise and interpretation was uncertain."

L15: Remove "whole", and maybe write in present tense, e.g. is made openly available…

L44: I suggest slightly re-wording to: "To study and address the issue in detecting bed returns obscured by scattering noise in temperate glaciers…"

L59-60: I suggest clarifying this a bit, e.g. the statement in the Results section is much clearer. "However, … 2008 to 2021. If the previous ice volume estimate was correct, Rutor Glacier would have lost 2/3 of its volume meaning that only 50 million $m^3$ still remain. However, while shrinking is evident, the remaining ice volume is likely larger given the large glacier area size." (please check if this is what you mean)

L61: I suggest specifying that the ice volume was underestimated in your hypothesis, e.g. "… ice volume estimate of 150 million $m^3$ is underestimated, …"

L114: I suggest removing "really"

L137: remove "little"

L218: replace "whole" with "entire"

L238: Replace "whole" with "broader"

L239: I suggest replacing "probably" with "likely"

L257-258: Could remove that last sentence from the paragraph, as it is a bit of repetition.

L334: I suggest changing to: "… which is close to the GlaTE…"

L335: You could replace glacier "outline" with "margin"

L340: Replace "quite" with "which is"

L341-345: I suggest reformulating this paragraph a bit to make it easier to read and make the message a bit more clear, for example something along: "Using these existing ice thickness model products to help interpreting the GPR data would be simpler than running the three models as performed in this study. However, these regional scale models may have outdated glacier outlines (which are changing rapidly)." (Please check if this is the message you wanted to convey. In terms of model uncertainty, I would expect that also the regional scale existing models have some sort uncertainty provided, but I am not familiar with them).

L353: I suggest being more specific and write: "Therefore, including modeled ice thickness to guide the GPR interpretations could …."

L356: I suggest changing to "…recognized to cause less clear GPR sections …"

L368-359: Instead of only explaining that you assigned "wrong" picks, I suggest giving a percentage of them (e.g. compared to the total GPR profile length) to further highlight how the models can help guide the GPR interpretation.

L373: I suggest rewording to "However, some drawbacks have to be considered."

L375-377: Could you clarify these two sentences a bit? What is meant by the "balance between GPR and models", and "leaking the models into the observations"? Do you mean the possibility of incorrect modeled ice thicknesses then leading to incorrect GPR interpretations (e.g. following a reflector that coincides with the model but is not the true bed)? I think this just needs some re-wording and being a bit more specific.

L407-408: I suggest rewording to: "This study highlights the benefit of combining the two worlds."

L408-410: This is just my personal preference, but I suggest instead of saying "GPR practitioners" and "modelers", rewording to e.g. "GPR applications" and "modeling studies".

L411-414: I suggest rewording to something along: "Thus, applying the workflow presented here to other glaciers with GPR data available could help calibrate regional models more accurately, …"

L423: Specify which map, I assume the bedrock topography?

---

## Author Response (AR2)

Dear Editorial team,
thank you for handling our manuscript with kind attention and care. We appreciated your dedication to it. You can find our replies to your comments below, divided by Editor, Reviewer 1 and Reviewer 2. Your text will be highlighted in *italics.*

We mainly:

- improved results and discussion about data interpretation subjectivity, as well as focus and meaning of the study.
- improved GPR profiles, by indicating where they intersect between each other, and by improving figures' resolution.
- added the original GlabTop2 model coded in IDL, alongside the python implementation of GlabTop2 that we have previously used. The original one performs better near glacier margins (see profile 2022, for example, in the left part).
-performed a general English revision.

Sincerely,
Andrea Vergnano and the Authors

**Editor**

*I have taken over this manuscript from the previous editor because the latter has unexpected urgent work obligations. I have read the manuscript, the two recent reviews and the three initial reviews.*
*At this point, there are two reviews of which one (#2) recommends minor revision and is generally very positive while the other (#1) is very critical and recommends rejection. I understand the critique raised by #1 and acknowledge that they consider the required revision as too substantial for recommending major revisions. However, I also acknowledge that there are four other reviews who recommend revisions (major or minor). In this challenging situation I take the following decision:*
*- I ask the authors to revise the manuscript according to both reviews. Review #1 needs to be fully addressed and the authors need to reply to each point raised by the reviewer, explaining how they will address the critique.*
*- I find the circular reasoning pointed out by #1 particularly concerning. I ask the authors to clearly address this issue. I noticed that Reviewer 3 from the first round of reviews had already commented on that issue ("However, it is important not to turn the logic around [...] reversing the process and leaking the modeling data into the GPR measurements can be delusive. This is the most important comment I have for this work, and I would like to see it discussed further in the manuscript.") It appears to me that the authors have not answered this critique which has now been repeated by reviewer #1 from the second round of reviews.*

Dear Editor,
thank you for the time you spent carefully reading our manuscript and previous reviews. We understand the difficult situation, and we are certainly satisfied of this rigorous peer-review process.

It is difficult for us to completely address the critique raised by reviewer 1, because we agree with them: there is leakage of model information into GPR data. Compared to the

first submission, in the last major revision, we had clarified this issue better (maybe not better enough), without denying it, addressing the first-round-reviewer#3 comments.

What we can try to do now, is to explain why we think that our methodology does not indulge in faulty "circular reasoning" thinking, but instead, it is simply a way to interpret the GPR and the models together.

In our opinion, every joint interpretation of two different methodologies, in a context where measurements are of indirect nature and subjective choices must be made, necessarily involves looking at the information of one to understand the information of the other, or, using Reviewer 3 words, leaking data from one to another.

Yes, we use model data to interpret GPR, and GPR to calibrate model data in a sort of basic, manual iterative process. The GPR and model techniques are on the same level in this difficult context: we are not proposing something like "a main technique and a secondary one supporting the interpretation of the main one". They both support each other, in our view, and the proposed methodology is just one way to merge them together.

Finally, we think that, thanks to the "open review" model, this debate can be valuable for the interested readers, to form their personal opinion about the validity of the proposed method. We decided to highlight these main points into the discussion section, in which we would like to express a humble and multilateral perspective about the proposed methodology.

In this regard, we added a little extra part in this revision, to address the lack of inter-analyst comparison raised by Reviewer 1.
We thought a lot about this, and we came up with this idea: in mid September 2025 we went to a scientific conference about GPR (https://www.gpritalia.it/gpritalia2025/), and we made a collective activity in which we asked the listeners to pick the bedrock on our GPR sections, before and after looking at the model estimations. We included a paragraph about this activity, as a qualitative inter-analyst comparison into the paper.

*Furthermore, a few editorial comments:*

*Lines 135 to 138: The text gives the impression that the authors are using the original GlabTop2 code. However, this is not the case as the original code was written in IDL. This code has been presented in the study by Frey et al. (2014) and used extensively in e.g. Farinotti et al. (2019). The authors must make clear that here a code is used which is based on the equations presented in Frey et al. (2014), not the original code. The code used here appears to be an incomplete implementation of the equations and description in Frey et al. (2014). The code appears to simulate large ice thickness directly at the glacier margin, as visible in Fig. 4. The original code contains algorithms to avoid such unrealistic sudden transitions from thick ice to no ice. Being the author of the original code, I would be happy to share the original code with the authors, however, its major limitation is that it is written in IDL and I fully understand if the authors prefer using a more modern Python code.*

Thanks for the proposal. We were indeed unsatisfied about the problem at the glacier margins. Unfortunately, we did not manage to run the original code in IDL, due to its closed-source nature, but we thank you to run it for us. We now present the paper as a

four-model comparison, instead of three. We stated in the methodology section that the Glabtop-py model is an open-source, partial implementation of the Original-Glabtop2 model.

*Lines 335 – 336: While I do not know how the original GlabTop2 code would have performed on Rutor glacier, this issue would likely have been reduced.*
*Lines 376 - 377: This appears to be the only mentioning of circular reasoning, although in a single sentence which is kept vague and whose message on human error is unclear to me. To be clear, any circular reasoning needs to be removed from this study for it to become acceptable.*

Please, refer to the lines above about what we think about the "circular reasoning". We acknowledge that this issue must be discussed more in depth. Now we included in the main text a more comprehensive discussion about it, mostly taking from the answer to your previous point, above.

**Reviewer #1**

*Dear authors,*
*Thank you for submitting a manuscript to The Cryosphere. I regret that I have to recommend the rejection of the manuscript after you have already completed a major revision, but it still contains significant weaknesses, particularly in the overall study design. Please find the detailed review below.*

*Summary*
*This manuscript presents a case study that integrates Ground Penetrating Radar (GPR) data with three ice-thickness inversion models (GlabTop2, GlaTE, OGGM) to improve the identification of the ice-bedrock interface in the temperate Rutor Glacier in the Italian Alps. The aim is to reconcile discrepancies between past GPR-based ice volume estimates and more recent geodetic observations by visually incorporating model-derived information into the manual interpretation of GPR profiles. The GlaTE model is then re-run using these interpretations as constraints.*

*The approach is well described, and the manuscript is generally clear in its presentation. However, the study lacks methodological rigor, as it does not sufficiently validate or critically assess the techniques employed. Additionally, the work offers limited originality, with a considerable proportion of the analysis closely resembling existing studies without introducing significant advancements to the field.*

Thank you for your clear and constructive feedback. We think that the issues you raised could be discussed and hopefully solved in a reasonable amount of time, therefore, we performed a major revision of our article, which we submit now to the editor and reviewers' attention. Please, see our answers below.

*Major Concerns*

*1. Conceptual Circularity and Confirmation Bias*
*The study's core methodology—using model outputs to guide the interpretation of GPR data and subsequently constraining one of those same models with those interpretations —is inherently circular. This undermines the objectivity of the interpretation process and risks reinforcing existing biases. Although the authors briefly acknowledge this issue, they do not provide any quantitative means of mitigating it. The absence of an independent validation dataset (e.g., boreholes) impairs this problem, leaving the model outputs effectively validated by themselves.*

We cannot object to your observation, all what you said is true. The point that we want to discuss is that we conducted our work with a perspective different from the classical "finding a method (in this case GPR) that can validate models".
Our global point of view is more like "finding an efficient way to merge two indirect, sparse and sometimes subjective methodologies".

In our opinion, every joint interpretation of two different methodologies, in a context where measurements are of indirect nature and subjective choices must be made, necessarily involves looking at the information of one to understand the information of the other, or, using Reviewer 3 words, leaking data from one to another.

Summarizing, yes, we use model data to interpret GPR, and GPR to calibrate model data in a sort of basic iterative process. In our workflow, the GPR and the models are on the same level: we did not want to present "a main technique and a secondary one supporting the interpretation of the main one". They both support each other, in our view, and the proposed methodology is just one way to merge them together.  This point of view is not valid for all cases, in our opinion: sometimes you have good GPR data and in these cases you take GPR as the main technique, and others as supporting techniques.

So, in our opinion, we are not in a situation of faulty "circular reasoning", considering the problem from the point of view elaborated above. It's like having 2 indirect clues of a unsolved murder and trying to find the killer: you look at the first clue, and look at the second clue with the first clue in your mind, and vice versa. Then, you get closer to the solution.

After this more "philosophical" answer, we would like to point out some more practical considerations:

About boreholes, it is generally accepted that many glacier studies exist without always showing borehole data, due to their very high economic and logistical costs. We obviously would like to have borehole data supporting our study, but it was economically unfeasible. About the risk reinforcing existing biases, we tried to improve the discussion section in this sense in the "Advantages and limitations of the methodology" paragraph. We think that, however, one result of our work is that, even if the final model could be wrong in some parts, at least it avoided blatant errors that in the past were made, like interpreting reflections due to internal glacier features as bedrock. This kind of mistakes possibly happened also on other temperate glaciers. It seems to us that the GPR interpretation on temperate glaciers can be hard, and similar ideas of interpreting together models and GPR are relevant for scientific community and professionals. We also highlighted in the abstract that we specifically present a point of view of applied geophysicists that have to find ways to handle difficult to interpret GPR data.

*2. Lack of Methodological Rigor in GPR Survey Design*
*The manual classification of radargram features into categories such as "sure," "model-guided," and "wrong" pickings is fundamentally subjective. The manuscript does not present any reproducibility metrics, inter-analyst comparisons, or uncertainty assessments based on signal characteristics. Given that these interpreted points are subsequently used to constrain a model and inform conclusions, this lack of rigor critically undermines the study's credibility.*
*The authors acknowledge widespread scattering and signal attenuation in the GPR data but do not appear to have made any effort to adjust the survey design in response. There is no discussion of antenna frequency selection, signal processing alternatives, or survey geometry optimization. Furthermore, the limited spatial coverage of the GPR lines in high-uncertainty areas is not addressed, nor is any attempt made to densify or extend profiles to improve data quality.*

We fear that we are unable to publish quantitative reproducibility metrics on a subjective data interpretation issue. This subjectivity of data interpretation is unfortunately spread across many geophysical techniques, and we have to accept that any GPR data analysis that includes manual selection of interfaces or signal reflections is subjective. We are not willing to publish *the perfect, most statistically correct, subjective data interpretation,* which

actually sounds to us as a strange concept. We are proposing an idea to *reduce subjectivity thanks to model estimations.* Our results show that in this test site this idea is better than just GPR, because only GPR, as performed in the past (Villa et al., 2008), did not pass the geodetic mass balance proof.

About the inter-analyst comparison, we came up with this idea: in mid September 2025 we went to a scientific conference about GPR, and made a collective activity in which we asked the listeners to pick the bedrock on our GPR sections, before and after looking at the model estimations. This experiment was qualitative, but we observed that all the participants (about 10), before seeing the models, picked the internal reflections of the glacier as bedrock, mistakenly. This kind of picking would greatly underestimate the total ice volume of the glacier. After seeing the models, they recognized deeper reflections as the bedrock (unfortunately, only 3 of them. The others did not complete this second part of the exercise). We received positive feedback from the audience, who were particularly concerned that temperate glaciers could be so hard to interpret with GPR, and operators can easily make interpretation mistakes. We included a short report on this in the discussion section, in the "advantages and limitations" paragraph.

About the antenna, we performed the 2022 survey with a lower frequency antenna compared to the 2012 survey (40 vs 70 MHz),and was ground based vs helicopter-based, exactly for this purpose of critically evaluate antenna and survey selection. We noticed no improvement, unfortunately. This was already reported in the methods sections, but we highlighted better that we tested two different antennas.

About signal processing, this was one of the main Ph.D. topics of one of the authors, therefore, we had plenty of time and determination to try every (literally) processing step included in ReflexW software, in various combinations, in order to improve the radargrams. Simple processing as the one shown resulted in being the most effective. We specified better this in the methods section, just after the processing workflow of the GPR.

*3. Exaggerated Claims About Accuracy and Reproducibility*
*The manuscript makes strong claims regarding the accuracy and robustness of its results, particularly the final GlaTE-constrained volume estimate of 450 million m³. However, these claims are undermined by the methodological circularity described above and the limited scope of the GPR data. The substantial discrepancy between this estimate and previous work (e.g., 150 million m³) is not subjected to critical scrutiny, nor is it convincingly reconciled.*

The presence of a substantial discrepancy was calculated based on very simple DEM pixel-by-pixel differencing, clearly suggesting that the previous 150 million $m^3$ estimation is wrong. There is ample space in the introduction where this is discussed (we made a reference to Figure 1, in the introduction, that seems clear to us). In the paper, we investigate and we discover that probably bad GPR data and consequently bad interpretation were the issues. Our 450 million $m^3$ estimate is certainly closer to the actual value, based indeed on DEM observation, but also by comparing critically this results with other available datasets (e.g. consensus estimate from Farinotti) - which provides an estimate of more than 450 million $m^3$, but this is justified because they used a 2003 outline, much bigger than that of 2021.

We do not agree that we did a *strong claim regarding the accuracy*. In the discussion, we wrote "It is difficult to say if this very low standard deviation is indicative of real reliability of the GlaTE constrained model; nevertheless, 450 million m3 of ice, distributed as in Figure 6, represents the best estimate of the Rutor glacier thickness we could obtain with this methodology."

This does not seems a *strong claim,* to us at least. We actually paid a lot of attention, given the previous and these reviews, to ensure the reader understands the limitations of the study. We are ready to change this and similar phrases if the reviewer thinks differently.

*4. Limited Geographical Scope and Sparse Dataset*
*The study is based entirely on a single glacier and relies on a limited amount of GPR data, collected along a relatively small number of profiles (relative to the glacier surface area). The narrow spatial focus and sparse dataset therefore constrain the strength of the conclusions. As such, the robustness of the proposed workflow remains difficult to assess.*

According to us, the main result of this study should not be linked to the accuracy of the Rutor glacier bedrock model obtained, which, we agree with you, would be quite limited in geographical scope. We reworked the beginning of the discussion to clarify that this work should be viewed more as a methodological tool, than a study about the Rutor glacier. The GPR dataset is sparse, as it was collected from previous available data and we had limited capacity to carry out new surveys. However, we again think that this is not the main point of the paper. Conversely, being able to process difficult-to-interpret GPR data, and notwithstanding their sparse nature (which can be frequent in other surveys on glaciers, due to the often limited budget and logistical constrains), is exactly the scope of the methodology proposed.

5. Originality
*The Cryosphere requires research of exceptional originality and novelty. Submissions should therefore "represent substantial progress beyond current scientific understanding." However, there appears to be another manuscript by Strallo et al. (2025) that presents a very similar methodological approach, albeit applied to a different Italian glacier. Despite this change in study site and the integration of other complementary datasets, the overarching design of the study remains largely unchanged. Notably, the Strallo et al. manuscript was submitted on 3 September 2024 — just two weeks after Vergnano et al. (2024) submitted their initial version to TC on 14 August 2024. Given that A. Godio is a co-author on both manuscripts and that A. Vergnano is explicitly acknowledged in the Strallo et al. paper, it is evident that there was awareness of the overlapping work. This raises concerns regarding adherence to established norms of good scientific practice. Furthermore, previous studies from recent years have already combined GPR data with ice thickness models to derive glacier-wide ice thickness distributions over substantially larger and more topographically complex areas (e.g., Grab et al.(2021), Helfricht et al. (2019)). While the current study applies similar principles, these earlier works provide a broader context and may serve as valuable benchmarks in terms of methodological scope.*

We are concerned about your first comment about the other regional-scale studies. In fact, it is difficult to compete with such excellent studies. Fortunately, our focus is different: having to deal with a single glacier, (and thanks to the reviewers concerns that made us elaborate better in this direction) we explored much more in depth the subjectivity issue, the intercomparison between models, and we present a point of view of a GPR analyst that have to deal with scattered data, a situation in which many readers of TC may encounter. Let's say that we explore a side-aspect of such more general studies, and given the implications that it had on the Rutor glacier, we think that it's worthy of reading, because similar issues frequently arise on local temperate glaciers.

To better state the literature context in which our paper is situated, we added a short paragraph at the beginning of the discussion, to introduce the aims of our discussion:

*"We discuss a methodological approach of interpreting together models and sparse GPR data, with specific reference to the results obtained on the Rutor glacier, which experienced a wrong estimation of glacier thickness due to the scattered GPR data (Villa et al., 2008). GPR has been the favorite tool to investigate glacier thickness in the last decades, however, we observed poor performance on temperate, meltwater-rich glaciers. At the same time, models that estimate glacier thickness based on surface topography and ice-flux equations are now of widespread use, and have been used to perform regional studies, for example on the Austrian glaciers Helfricht et al., 2019) and on Swiss glaciers (Grab et al, 2021) the latter including large-scale GPR surveys.*

*We focus on a side-aspect of such general studies: the specific, but not uncommon, situation of dealing with very scattered and unclear GPR data on temperate glaciers. We compare different ice-thickness models and literature products, and evaluate how they can help GPR analysts before and after a new GPR survey. Then, we discuss the subjectivity in interpreting GPR data, which can bring to possible human errors, especially in difficult contexts, and how models can be integrated in the interpretation process. This subjectivity is intrinsic in GPR data interpretation, but may raise concerns, therefore, we try to discuss the different aspects of this issue, applied to our situation. "*

A few comments about Strallo et al. Methodologically, our and their papers are different, in our opinion, because this paper explicitly focus on the problem of subjective interpretation of "bad" GPR data on temperate glaciers, which is not the main topic of Strallo et al. The present paper focuses on the Rutor Glacier and specifically addresses the challenges of interpreting GPR data in temperate glaciers with high englacial water content. It compares four ice-thickness models (OGGM, GlabTop2-Py, Original-Glabtop2, GlaTE) and emphasizes the synergy between model outputs and GPR data to improve the identification of the ice-bedrock interface. The study also includes a sensitivity analysis and provides an open workflow. It highlights the importance of combining multiple approaches to reduce uncertainty and improve accuracy under challenging conditions. It revises previous Rutor Glacier ice volume estimates upward by integrating model estimates with GPR constraints.

Instead, in the paper by Strallo et al. (2025), the authors adopted an integrated approach for the Indren Glacier, combining historical data, GPR measurements, remote sensing, temperature analysis, and modelling (GlaTE algorithm) to assess glacier thickness changes over two decades. Their results enhance the understanding of glacier dynamics and inform water resource management and hazard mitigation.

The methodologies may be similar, but using established methodologies such as GPR and models does not necessarily mean that the paper is outdated. In fact, we see GPR studies continue to publish, probably because they explore different aspects of the methodologies or new focuses or new applications. For us, the important is that the focus of our and Strallo's papers are completely different. We suppose that the readers may be interested in both discussions, since they are two different focuses and two different points of views.

*Minor Concerns*
*Language and Style: The manuscript requires careful editing for grammar, syntax, and clarity. Specific issues include incorrect sentence constructions, and inconsistent punctuation (e.g., inconsistent use of the Oxford comma). The authors are encouraged to*

*consult with a native English speaker or use AI-assisted editing tools to improve readability.*

Thank you. We carried out a general editing for grammar, syntax and clarity as you suggested.

*Recommendation: Reject*

*This manuscript, in its current form, does not meet the high scientific standards for publication in The Cryosphere. A complete methodological redesign would be required to render the work suitable for reconsideration. Additionally, including a few additional study sites with varying glaciological characteristics would greatly strengthen the credibility and generalizability of the proposed approach. Moreover, glacier thickness models—while valuable—should not be used as a substitute for reliable observational data or to compensate for poor GPR data quality.*

After this revision, we hope you could appreciate a little our perspective, which may not be a classical one, but may represent a reasonable point of view to consider when interpreting difficult datasets. Our point of view is that of a geophysicist that have to handle with difficult data to interpret, not of a glaciologist that aims to calculate a regional ice-thickness model.

*References:*
*Grab, M., Mattea, E., Bauder, A., Huss, M., Rabenstein, L., Hodel, E., … Maurer, H. (2021). Ice thickness distribution of all Swiss glaciers based on extended ground-penetrating radar data and glaciological modeling. Journal of Glaciology, 67(266), 1074–1092. doi:10.1017/jog.2021.55*
*Helfricht et al. (2019): Calibrated Ice Thickness Estimate for All Glaciers in Austria. Front. Earth Sci., 12 April 2019, https://doi.org/10.3389/feart.2019.00068*
*Strallo, V., Colombero, C., Troilo, F., Mondardini, L., & Godio, A. (2025). Glacier thickness modelling and monitoring with geophysical data constraints: A case study on the Indren Glacier (NW Italy). Earth Surface Processes and Landforms, 50(1), e6068.* *https://doi.org/10.1002/esp.6068*

**Reviewer #2**

*Re-review of "Integrating GPR and ice-thickness models for improved bedrock detection: the case study of Rutor temperate glacier" by Andrea Vergnano et al. (2025)*

*I am very excited to see this revised version, and commend the authors for the considerable effort they have invested in revising this manuscript. The new manuscript addresses almost all of my earlier concerns, and it is now written much more clear and well structured. Specifically, the research goals, hypotheses and scientific contribution are now stated clearly throughout the manuscript, the manuscript reads well, and the updated figures are much easier to read. I also really appreciate the inclusion of the comprehensive supplementary material, making these codes and the processing flow available is a valuable service to the community.*

*Lastly, I really appreciate the addition of pick classifications ("sure", "guided by models", "wrong"), as it reveals how manual and model-assisted interpretations interact. I have one suggestion that could further highlight the impact of the model inclusion when interpreting GPR data: I suggest adding a brief quantitative summary that exploits the pick categories, e.g. reporting the percentages of "wrong", "guided" and "sure picks" relative to the total GPR data. These could simply be mentioned in brackets throughout the text that is already there. I believe that the necessary data for this analysis now exists, and this should be straightforward.*

Thank you very much for your positive and constructive feedback, yes, we put a lot of effort in revising this manuscript, as also the reviewers did. We included your suggestion about the pick categories into the text.

*Apart from this minor suggestion and a few trivial language suggestions below, I believe the manuscript is ready for publication and recommend acceptance after these have been addressed.*

*Line-by-line comments*

Thank you very much for your kind line-by-line comments, for improving our phrases. We appreciated it a lot.

*L9-11: I suggest rewording to: "Combined visualization of the GPR and model data in 2D and 3D…, especially where the GPR data contained scattering noise and interpretation was uncertain."*

Thank you, done.

*L15: Remove "whole", and maybe write in present tense, e.g. is made openly available…*

Thank you, done.

*L44: I suggest slightly re-wording to: "To study and address the issue in detecting bed returns obscured by scattering noise in temperate glaciers…"*

Thank you, done.

*L59-60: I suggest clarifying this a bit, e.g. the statement in the Results section is much clearer. "However, … 2008 to 2021. If the previous ice volume estimate was correct, Rutor Glacier would have lost 2/3 of its volume meaning that only 50 million m3 still remain. However, while shrinking is evident, the remaining ice volume is likely larger given the large glacier area size." (please check if this is what you mean)*
Thank you, we changed the text as you suggested.

*L61: I suggest specifying that the ice volume was underestimated in your hypothesis, e.g. "… ice volume estimate of 150 million m3 is underestimated, …"*

Thank you, corrected.

*L114: I suggest removing "really" L137: remove "little"*

Thank you, done.

*L218: replace "whole" with "entire"*

Thank you, replaced.

*L238: Replace "whole" with "broader"*

Thank you, replaced.

*L239: I suggest replacing "probably" with "likely"*

Thank you, replaced.

*L257-258: Could remove that last sentence from the paragraph, as it is a bit of repetition.*

Thank you, removed.

*L334: I suggest changing to: "… which is close to the GlaTE…"*

Thank you, done.

*L335: You could replace glacier "outline" with "margin"*

Thank you, replaced.

*L340: Replace "quite" with "which is"*

Thank you, replaced.

*L341-345: I suggest reformulating this paragraph a bit to make it easier to read and make the message a bit more clear, for example something along: "Using these existing ice thickness model products to help interpreting the GPR data would be simpler than running the three models as performed in this study. However, these regional scale models may have outdated glacier outlines (which are changing rapidly)." (Please check if this is the message you wanted to convey. In terms of model uncertainty, I would expect that also the regional scale existing models have some sort uncertainty provided, but I am not familiar with them).*

Thank you for the suggestion, and we think you are right about the regional-scale uncertainty. We rephrased it to tell that in both cases (literature products or self-run models) it is important to check the model uncertainty.

*L353: I suggest being more specific and write: "Therefore, including modeled ice thickness to guide the GPR interpretations could …."*

Thank you, done.

*L356: I suggest changing to "…recognized to cause less clear GPR sections …"*

Thank you, changed.

*L368-359: Instead of only explaining that you assigned "wrong" picks, I suggest giving a percentage of them (e.g. compared to the total GPR profile length) to further highlight how the models can help guide the GPR interpretation.*
Thank you, we added these percentages and a simple calculation spreadsheet in the supplementary materials, folder number 8.

*L373: I suggest rewording to "However, some drawbacks have to be considered."*

Thank you, done.

*L375-377: Could you clarify these two sentences a bit? What is meant by the "balance between GPR and models", and "leaking the models into the observations"? Do you mean the possibility of incorrect modeled ice thicknesses then leading to incorrect GPR interpretations (e.g. following a reflector that coincides with the model but is not the true bed)? I think this just needs some re-wording and being a bit more specific.*

Thank you, it was pointed out also by the Editor and the other reviewer. We reworded and discussed more in depth this phrase. Now it reads:

"
However, the most delicate aspect which may limit this methodology is the balance between the reliability of GPR and that of the models. In fact, our approach is a joint interpretation of two different methodologies, in a context where data are of indirect nature and subjective choices must be made. We need the information of the models to better understand the scattered and sparse GPR data, and vice versa, in a sort of manual iterative interpretation process. We stress that, in this difficult context, we do not see the GPR as the main technique, and the models as a supporting information, but we see that they both support each other. However, the subjective interpretation is potentially prone to human error. In our opinion, the key to minimizing this potential error is to perform simulations on multiple models, display the GPR sections with multiple color scales and in both 2D and 3D environments, and acquire local knowledge about the studied glacier.
"

*L407-408: I suggest rewording to: "This study highlights the benefit of combining the two worlds."*

Thank you, done.

*L408-410: This is just my personal preference, but I suggest instead of saying "GPR practitioners" and "modelers", rewording to e.g. "GPR applications" and "modeling studies".*

We agree, changed.

*L411-414: I suggest rewording to something along: "Thus, applying the workflow presented here to other glaciers with GPR data available could help calibrate regional models more accurately, …"*

Thank you, changed.

*L423: Specify which map, I assume the bedrock topography?*

Yes, thank you. We specified it.

---

## Author Response (AR3)

Dear Editor and Reviewers,

thank you for your careful assessment of our manuscript once again. We appreciate your care in improving our manuscript and making our central message more explicit.
We accept and we performed all the minor corrections that you suggested, and reviewed the clarity and conciseness of the entire text, trying to sharpen the central message. In particular, we would like to comment here in more detail about this issue:

"L337: This makes me wonder why the two estimates are different, when it is supposed to be the same model? It would be good to add a sentence explaining this."
We thank you for the insightful comment. Fortunately the Editor, being the author of Original-Glabtop2, gave us access to the source code of the algorithm, and we could compare the two Glabtop2 codes. We understood that probably it is due to the different interpolation method, different number of random cells selected to calculate ice thickness, different buffer zone to assign fixed thickness at the glacier margins. The core method and parameters are the same, but the implementation is slightly different in the interpolation, providing, actually, quite different results. So... it's not only the algorithm itself, but also side aspects of it, such as the interpolation to the glacier margins, that influence the results, and in a non negligible way! We added this explanation to the discussion.

We report Reviewer #2 comments in the following:

I appreciate the detailed responses to the reviewers. I previously recommended publication, and I continue to believe that the manuscript is suitable for publication after a few minor edits in the newly added text (see comments below).

Regarding the concern raised by the other reviewer about potential circularity in the method, I agree that this is an important point to clarify, but in my view the issue is not as severe as suggested. The authors use model outputs from well-established, state-of-the-art models to aid the interpretation of the GPR data. These model results convincingly support the interpretation that the shallow scattering horizons are unlikely to represent the true bed. Without this modelling context, the GPR data could be misinterpreted, and the resulting GPR-derived ice-thickness estimates might then be used to constrain a subsequent model in a way that would introduce significant errors. In this case, using the model estimates prior to GPR interpretation helps to avoid such misinterpretation and, in my opinion, is a defensible and useful approach.

That said, I think the authors could sharpen how this argument is presented. Some paragraphs are quite lengthy, and the central message (modelling is used to avoid misinterpreting shallow reflectors as bed) does not always stand out clearly. I suggest tightening the relevant sections and making this key point even more explicit.
Overall, the language of the manuscript could still be improved (e.g. for clarity and making it more concise).

In summary, I recommend acceptance subject to some minor revisions (see comments below + potential language editing to improve clarity).

Line-by-line comments
L11: re-word to e.g. "… on the same glacier, which highlights the difficulty in identifying the true bed reflection."
L266: remove "probably"
L276: I suggest rewording to e.g. "Therefore, additional information of the bedrock topography is helpful. This information came from…"

L294-L296: Reword to e.g. "The reader is strongly encouraged to look at those GPR profiles and model estimations, …"

L312-318: There is quite some repetition in this paragraph, and without much additional information than what is already in the discussion. I suggest rewording to: "We focus on the specific challenge of scattered and unclear GPR data on temperate glaciers and evaluate how ice-thickness models can help interpreting such datasets."

L337: This makes me wonder why the two estimates are different, when it is supposed to be the same model? It would be good to add a sentence explaining this.

L389: Rephrase to "… On the Rutor glacier, it is likely the combination of all these that challenge the visibility of the bedrock reflection".

L393: I suggest rewording to: "By using the constrained GlaTE model rather than only interpolation, better informed ice-thickness estimates can be made in areas not surveyed by GPR."

L421-425: I'm not sure that this belongs in a scientific paper, and would consider removing it.